# BEHAVIOR-GUIDED REINFORCEMENT LEARNING

## ABSTRACT

We introduce a new approach for comparing reinforcement learning policies, using Wasserstein distances (WDs) in a newly defined latent behavioral space. We show that by utilizing the dual formulation of the WD, we can learn score functions over trajectories that can be in turn used to lead policy optimization towards (or away from) (un)desired behaviors. Combined with smoothed WDs, the dual formulation allows us to devise efficient algorithms that take stochastic gradient descent steps through WD regularizers. We incorporate these regularizers into two novel on-policy algorithms, Behavior-Guided Policy Gradient and Behavior-Guided Evolution Strategies, which we demonstrate can outperform existing methods in a variety of challenging environments. We also provide an open source demo[1].

## 1 INTRODUCTION

One of the key challenges in reinforcement learning (RL) is to efficiently incorporate the behavioral characteristics of learned policies into optimization algorithms (Lee & Popovic, 2010; Meyerson et al., 2016; Conti et al., 2018). The fundamental question we aim to shed light on in this paper is:

*What is the right measure of similarity between two policies acting on the same underlying MDP and how can we devise algorithms to leverage this information for reinforcement learning?*

In simple terms, the main thesis motivating the methods we propose is that:

*Two policies may perform similar actions at a local level but result in very different global behaviors.*

We propose to define *behaviors* via so-called Behavioral Embedding Maps (BEMs), which are functions mapping trajectories (realizations of policies) into latent behavioral spaces representing trajectories in a compact way. BEMs enable us to identify policies with their Probabilistic Policy Embeddings (PPEs), which we define as the pushforward distributions over trajectory embeddings as a result of applying a BEM to a policy's trajectories. Importantly, two policies with distinct distributions over trajectories may result in the same probabilistic embedding. PPEs provide us a way to rigorously define dissimilarity between policies. We do this by equipping them with metrics defined on the manifold of probabilistic measures, namely a class of Wasserstein distances (WDs, Villani (2008)). There are several reasons for choosing WDs:

- **Flexibility**. We can use any cost function between embeddings of trajectories, allowing the distance between PPEs to arise organically from an interpretable distance between embedding points.

- **Non-injective BEMs**. Different trajectories may be mapped to the same embedding point (for example in the case of the last-state embedding). This precludes the use of likelihood-based distances such as the KL divergence (Kullback & Leibler, 1951), which we discuss in Section 6.

- **Behavioral Test Functions**. Solving the dual formulation of the WD objective yields a pair of test functions over the space of embeddings that can be used to score trajectories.

The behavioral test functions underpin all our algorithms, directing optimization towards desired behaviors. To learn them it suffices to define the BEM and the cost function between points in the PPE space. To mitigate the computational burden of computing WDs, we rely on their entropy-regularized formulations. This allows us to update the learned test functions in a computationally efficient manner

---

[1] Available at `https://github.com/behaviorguidedRL/BGRL`. We emphasize this is not an exact replica of the code from our experiments, but a demo to build intuition and clarify our methods.

via stochastic gradient descent (SGD) on a Reproducing Kernel Hilbert Space (RKHS). We develop a novel method for stochastic optimal transport based on random feature maps (Rahimi & Recht, 2008) to produce compact and memory-efficient representations of learned behavioral test functions. Finally, having laid the groundwork for comparing trajectories via behavior-driven trajectory scores, we address our core question by introducing two new on-policy RL algorithms:

- **Behavior Guided Policy Gradients (BGPG)**: We propose to replace the KL-based trust region from Schulman et al. (2015) with a WD-based trust region in the PPE space.

- **Behavior Guided Evolution Strategies (BGES)**: Inspired by the NSR-ES algorithm from Conti et al. (2018), BGES jointly optimizes for reward and *novelty* using the WD in the PPE space.

In addition, we also demonstrate a way to harness our methodology for imitation learning (Section 7.3) and repulsion learning (Section 9.4), and we believe there may be many more potential applications in the future.

## 2    MOTIVATING BEHAVIOR-GUIDED REINFORCEMENT LEARNING

Throughout this paper we prompt the reader to think of a policy as a distribution over its trajectories, induced by the policy's (possibly stochastic) map from state to actions and the unknown environment dynamics. We care about summarizing (or embedding) trajectories into succinct representations that can be compared with each other (via a cost/metric). These comparisons arise naturally when answering questions such as: Has a given trajectory achieved a certain level of reward? Has it visited a certain part of the state space? We think of these summaries or embeddings as characterizing the behavior of the trajectory. We formalize these notions in Section 3.

We show that by identifying policies with the embedding distributions that result of applying the embedding function (summary) to their trajectories, and combining this with the provided cost metric, we can induce a topology over the space of policies given by the WD over their embedding distributions. The methods we propose can be thought of as ways to leverage this "behavior" geometry for a variety of downstream applications such as policy optimization and imitation learning.

This topology emerges naturally from the sole definition of an embedding map (behavioral summary) and a cost function. Crucially these choices occur in the semantic space of behaviors as opposed to parameters or visitation frequencies[2]. One of the advantages of choosing a Wasserstein geometry is that non-surjective trajectory embedding maps are allowed. This is not possible with a KL induced one (in non-surjective cases, computing the likelihood ratios in the KL definition is in general intractable). In Sections 4 and 5 we show that in order to get a handle on this geometry we can use the dual formulation of the Wasserstein distance to learn functions (Behavioral Test Functions) that can provide scores on trajectories which then can be added to the reward signal (in policy optimization) or used as a reward (in Imitation Learning).

In summary, by defining an embedding map of trajectories into a behavior embedding space equipped with a metric[3], our framework allows us to learn "reward" signals (Behavioral Test Functions) that can serve to steer policy search algorithms through the "behavior geometry" either in conjunction with a task specific reward (policy optimization) or on their own (e.g. Imitation Learning). We develop versions of on policy RL algorithms which we call Behavior Guided Policy Gradient (BGPG) and Behavior Guided Evolution Strategies (BGES) that enhance their baseline versions by the use of learned Behavioral Test Functions. Our experiments in Section 7 show this modification is useful. We also show how to use Behavioral Test Functions in Imitation Learning, where we only need access to an expert's embedding. Although our framework also has obvious applications to safety, (learning policies that avoid undesirable or dangerous behaviors) we leave this for future work. We also consider simple heuristics for the embeddings (inspired by other existing use cases), but believe future work on learned embeddings could be a significant enhancement.

---

[2]If we choose an appropriate embedding map our framework handles visitation frequencies as well.

[3]The embedding space can be discrete or continuous and the metric need not be smooth, and can be for example a simple discrete $\{0, 1\}$ valued criterion

## 3    DEFINING BEHAVIOR IN REINFORCEMENT LEARNING

A Markov Decision Process (MDP) is a tuple $(\mathcal{S}, \mathcal{A}, \mathrm{P}, \mathrm{R})$. Here $\mathcal{S}$ and $\mathcal{A}$ stand for the sets of states and actions respectively, such that for $s, s' \in \mathcal{S}$ and $a \in \mathcal{A}$: $\mathrm{P}(s'|a, s)$ is the probability that the system/agent transitions from $s$ to $s'$ given action $a$ and $\mathrm{R}(s', a, s)$ is a reward obtained by an agent transitioning from $s$ to $s'$ via $a$. A policy $\pi_\theta : \mathcal{S} \to \mathcal{A}$ is a (possibly randomized) mapping (parameterized by $\theta \in \mathbb{R}^d$) from $\mathcal{S}$ to $\mathcal{A}$. Let $\Gamma = \{\tau = s_0, a_0, r_0, \cdots s_H, a_H, r_H \text{ s.t. } s_i \in \mathcal{S}, a_i \in \mathcal{A}, r_i \in \mathbb{R}\}$ be the set of possible trajectories enriched by sequences of partial rewards under some policy $\pi$. The undiscounted reward function $\mathcal{R} : \Gamma \to \mathbb{R}$ (which expectation is to be maximized by optimizing $\theta$) satisfies $\mathcal{R}(\tau) = \sum_{i=0}^{H} r_i$, where $r_i = R(s_{i+1}, a_i, s_i)$.

### 3.1    BEHAVIORAL EMBEDDINGS

We start with a Behavioral Embeddng Space (BES) which we denote as $\mathcal{E}$ and a Behavioral Embedding Map (BEM), $\Phi : \Gamma \to \mathcal{E}$, mapping trajectories to embeddings in $\mathcal{E}$ (Fig. 1). Importantly, the mapping does not need to be surjective. We will provide examples of BESs and BEMs at the end of the section.   Given a policy $\pi$, we let $\mathbb{P}_\pi$ denote the distribution induced over the spaces of trajectories $\Gamma$ and by $\mathbb{P}_\pi^\Phi$ the corresponding pushforward distribution on $\mathcal{E}$ induced by $\Phi$. We call $P_\pi^\Phi$ the Probabilistic Policy Embedding (PPE) of a policy $\pi$. A policy $\pi$ can be fully characterized by the distribution $\mathbb{P}_\pi$.

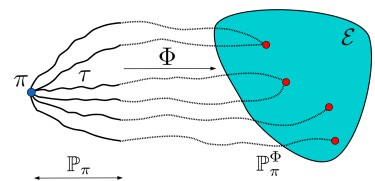

Figure 1: **Behavioral Embedding Maps (BEMs)** map trajectories to points in the behavior embedding space $\mathcal{E}$. Two trajectories may map to the same point in $\mathcal{E}$.

Additionally, we require the BES $\mathcal{E}$ to be equipped with a metric (or cost function) $C : \mathcal{E} \times \mathcal{E} \to \mathbb{R}$. Given two trajectories $\tau_1, \tau_2$ in $\Gamma$, $C(\Phi(\tau_1), \Phi(\tau_2))$ measures how different these trajectories are in the behavior space. The following are examples of BEMs (with the corresponding BESs) categorized into three main types (we will use examples from all three types in our experiments in Section 7):

1. **State-based:** the final state $\Phi_1(\tau) = s_H$, the visiting frequency of a fixed state $\Phi_2^s(\tau) = \sum_{t=0}^{H} \mathbf{1}(s_t = s)$, the frequency vector of visited states $\Phi_3(\tau) = \sum_{t=0}^{H} e_{s_t}$ (where $e_s \in \mathbb{R}^{|\mathcal{S}|}$ is the one-hot vector corresponding to state $s$); see also Section 7.2.
2. **Action-based:** the concatenation of actions $\Phi_4(\tau) = [a_0, ..., a_H]$; see also Section 7.1.
3. **Reward-based:**  the total reward $\Phi_5(\tau) = \sum_{t=0}^{H} r_t$, reward-to-go vector $\Phi_6(\tau) = \sum_{t=0}^{H} r_t \left( \sum_{i=0}^{t} e_i \right)$ (where $e_i \in \mathbb{R}^{H+1}$ is a one-hot vector corresponding to $i$ and with dimensions indexed from 0 to $H$); see also Section 7.1 and Section 7.3.

For instance, $\mathbb{P}_\pi^{\Phi_3}$ is the frequency with which different states are visited under $\pi$. Note that some of the above embeddings are only for the tabular case ($|\mathcal{S}|, |\mathcal{A}| < \infty$) while others are universal.

## 4    WASSERSTEIN DISTANCE & OPTIMAL TRANSPORT PROBLEM

Let $\mu, \nu$ be (Radon) probability measures over domains $\mathcal{X} \subseteq \mathbb{R}^m, \mathcal{Y} \subseteq \mathbb{R}^n$ and let $\mathcal{C} : \mathcal{X} \times \mathcal{Y} \to \mathbb{R}$ be a cost function. For $\gamma > 0$, a *smoothed* Wasserstein Distance is defined as:

$$\mathrm{WD}_\gamma(\mu, \nu) := \min_{\pi \in \Pi(\mu, \nu)} \int_{\mathcal{X} \times \mathcal{Y}} C(\mathbf{x}, \mathbf{y}) d\pi(\mathbf{x}, \mathbf{y}) + \gamma \mathrm{KL}(\pi|\xi), \tag{1}$$

where $\Pi(\mu, \nu)$ is the space of couplings (joint distributions) over $\mathcal{X} \times \mathcal{Y}$ with marginal distributions $\mu$ and $\nu$, $\mathrm{KL}(\cdot|\cdot)$ denotes the KL divergence between distributions $\pi$ and $\rho$ with support $\mathcal{X} \times \mathcal{Y}$ defined as: $\mathrm{KL}(\pi|\rho) = \int_{\mathcal{X} \times \mathcal{Y}} \left( \log \left( \frac{d\pi}{d\xi}(\mathbf{x}, \mathbf{y}) \right) \right) d\pi(\mathbf{x}, \mathbf{y})$ and $\xi$ is a reference measure over $\mathcal{X} \times \mathcal{Y}$. When the cost is an $\ell_p$ distance and $\gamma = 0$, $\mathrm{WD}_\gamma$ is also known as the Earth mover's distance and the corresponding optimization problem is known as the *optimal transport problem* (OTP).

### 4.1    WASSERSTEIN DISTANCE: DUAL FORMULATION

We will use smoothed WDs to derive efficient regularizers for RL algorithms. To arrive at this goal, we first need to consider the dual form of Equation 1. Under the subspace topology (Bourbaki,

---

**Algorithm 1** Random Features Wasserstein SGD

---

**Input:** kernels $\kappa, \ell$ over $\mathcal{X}, \mathcal{Y}$ respectively with corresponding random feature maps $\phi_\kappa, \phi_\ell$, smoothing parameter $\gamma$, gradient step size $\alpha$, number of optimization rounds $M$, initial dual vectors $\mathbf{p}_0^\mu, \mathbf{p}_0^\nu$.

**for** $t = 0, \cdots, M$ **do**

   1. Sample $(x_t, y_t) \sim \mu \bigotimes \nu$.

   2. Update $\begin{pmatrix} \mathbf{p}_t^\mu \\ \mathbf{p}_t^\nu \end{pmatrix} = \begin{pmatrix} \mathbf{p}_{t-1}^\mu \\ \mathbf{p}_{t-1}^\nu \end{pmatrix} + \frac{\alpha}{\sqrt{t}} \left( 1 - \exp\left( \frac{(\mathbf{p}_{t-1}^\mu)^\top \phi_\kappa(x_t) - (\mathbf{p}_{t-1}^\nu)^\top \phi_\ell(x_t) - C(x_t, y_t)}{\gamma} \right) \right) \begin{pmatrix} \phi_\kappa(x_t) \\ -\phi_\ell(y_t) \end{pmatrix}$

**Return:** $\mathbf{p}_M^\mu, \mathbf{p}_M^\nu$.

---

1966) for $\mathcal{X}$ and $\mathcal{Y}$, let $\mathcal{C}(\mathcal{X})$ denote the space of continuous functions on $\mathcal{X}$ and let $\mathcal{C}(\mathcal{Y})$ denote the space of continuous functions over $\mathcal{Y}$. The choice of the subspace topology ensures our discussion encompasses the discrete case.

Let $C : \mathcal{X} \times \mathcal{Y} \to \mathbb{R}$ be a cost function, interpreted as the "ground cost" to move a unit of mass from $x$ to $y$. Define $\mathbb{I}$ as the $(0, \infty)$ indicator function, where the value 0 denotes set membership. Using Fenchel duality, we can obtain the following dual formulation of the problem in Eq. 1:

$$\mathrm{WD}_\gamma(\mu, \nu) = \max_{\lambda_\mu \in \mathcal{C}(\mathcal{X}), \lambda_\nu \in \mathcal{C}(\mathcal{Y})} \int_\mathcal{X} \lambda_\mu(\mathbf{x}) d\mu(\mathbf{x}) - \int_\mathcal{Y} \lambda_\nu(\mathbf{y}) d\nu(\mathbf{y}) - E_C(\lambda_\mu, \lambda_\nu), \quad (2)$$

where $E_C(\lambda_\mu, \lambda_\nu)$ is defined as:

$$E_C(\lambda_\mu, \lambda_\nu) := \begin{cases} \gamma \int_{\mathcal{X} \times \mathcal{Y}} \exp\left( \frac{\lambda_\mu(\mathbf{x}) - \lambda_\nu(\mathbf{y}) - C(\mathbf{x}, \mathbf{y})}{\gamma} \right) d\xi(\mathbf{x}, \mathbf{y}) & \text{if } \gamma > 0 \\ \mathbb{I}((\lambda_\nu, \lambda_\nu) \in \{(u, v) \text{ s.t. } \forall (\mathbf{x}, \mathbf{y}) \in \mathcal{X} \times \mathcal{Y}\, u(\mathbf{x}) - v(\mathbf{y}) \le C(\mathbf{x}, \mathbf{y})\}) & \text{if } \gamma = 0. \end{cases} \quad (3)$$

We will set $d\xi(\mathbf{x}, \mathbf{y}) \propto 1$ for discrete domains and $d\xi(\mathbf{x}, \mathbf{y}) = d\mu(\mathbf{x}) d\nu(\mathbf{y})$ otherwise.

If $\lambda_\mu^*, \lambda_\nu^*$ are the functions achieving the maximum in Eq. 2, and $\gamma$ is sufficiently small then $\mathrm{WD}_\gamma(\mu, \nu) \approx \mathbb{E}_\mu\left[\lambda_\mu^*(\mathbf{x})\right] - \mathbb{E}_\nu\left[\lambda_\nu^*(\mathbf{y})\right]$, with equality when $\gamma = 0$. When for example $\gamma = 0$, $\mathcal{X} = \mathcal{Y}$, and $C(x, x) = 0$ for all $x \in \mathcal{X}$, it is easy to see $\lambda_\mu^*(x) = \lambda_\nu^*(x) = \lambda^*(x)$ for all $x \in \mathcal{X}$. In this case the difference between $\mathbb{E}_\mu\left[\lambda^*(\mathbf{x})\right]$ and $\mathbb{E}_\mu\left[\lambda^*(\mathbf{y})\right]$ equals the WD. In other words, the function $\lambda^*$ gives higher scores to regions of the space $\mathcal{X}$ where $\mu$ has more mass. This observation is key to the success of our algorithms in guiding optimization towards desired behaviors.

### 4.2 Computing $\lambda_\mu^*$ and $\lambda_\nu^*$

We combine several techniques to make the optimization of objective from Eq. 2 tractable. First, we replace $\mathcal{X}$ and $\mathcal{Y}$ with the functions from a RKHS corresponding to universal kernels (Micchelli et al., 2006). This is justified since those function classes are dense in the set of continuous functions of their ambient spaces. In this paper we choose the Gaussian kernel and approximate it using random Fourier feature maps (Rahimi & Recht, 2008) to increase efficiency. Consequently, the functions $\lambda$ learned by our algorithms have the following form: $\lambda(\mathbf{x}) = (\mathbf{p}^\lambda)^\top \phi(\mathbf{x})$, where $\phi$ is a random feature map with $m$ standing for the number of random features and $\mathbf{p}^\lambda \in \mathbb{R}^m$. For the Gaussian kernel, $\phi$ is defined as follows: $\phi(\mathbf{z}) = \frac{1}{\sqrt{m}} \cos(\mathbf{Gz} + \mathbf{b})$ for $\mathbf{z} \in \mathbb{R}^d$, where $\mathbf{G} \in \mathbb{R}^{m \times d}$ is Gaussian with iid entries taken from $\mathcal{N}(0, 1)$, $b \in \mathbb{R}^m$ with iid $b_i$s such that $b_i \sim \mathrm{Unif}[0, 2\pi]$ and the $\cos$ function acts elementwise.

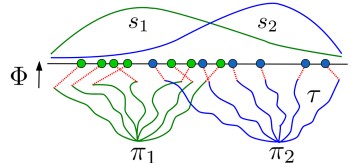

Figure 2: **Behavioral embedding functions** corresponding to two policies $\pi_1$ (green) and $\pi_2$ (blue) whose BEMs map trajectories to points in the real line.

Henceforth, when we refer to optimization over $\lambda$, we mean optimizing over corresponding dual vectors $\mathbf{p}^\lambda$ associated with $\lambda$. We can solve for the optimal dual functions by performing SGD over the dual objective in Eq. 2. Algorithm 1 is the random features equivalent of Algorithm 3 in Genevay et al. (2016) and will be a prominent subroutine of our methods. An explanation and proof of why this is the right stochastic gradient is in Lemma 10.2 in the Appendix.

If $\mathbf{p}_*^\mu, \mathbf{p}_*^\nu$ are the optimal dual vectors and $(x_1, y_1), \cdots, (x_k, y_k) \overset{\text{i.i.d}}{\sim} \mu \bigotimes \nu$, then Algorithm 1 can be used to get an estimator of $\text{WD}_\gamma(\mu, \nu)$ as follows:

$$\widehat{\text{WD}}_\gamma(\mu, \nu) = \frac{1}{k} \sum_{i=1}^k \langle \mathbf{p}_*^\mu, \phi_\kappa(x_i) \rangle - \langle \mathbf{p}_*^\nu, \phi_\ell(y_i) \rangle + \frac{1}{\gamma} \exp\left( \frac{\phi_\kappa(x_i)^\top \mathbf{p}_*^\mu - \phi_\ell(y_i)^\top \mathbf{p}_*^\nu - C(x_i, y_i)}{\gamma} \right) \quad (4)$$

## 5 BEHAVIOR-GUIDED REINFORCEMENT LEARNING

Here we introduce the framework which allows us to incorporate our behavioral approach to reinforcement learning into practical on-policy algorithms. Denote by $\pi_\theta$ a policy parameterized by $\theta \in \mathbb{R}^d$. The goal of policy optimization algorithms is to find a policy maximizing, as a function of the policy parameters, the expected total reward $\mathcal{L}(\theta) := \mathbb{E}_{\tau \sim \mathbb{P}_{\pi_\theta}} [\mathcal{R}(\tau)]$.

### 5.1 BEHAVIORAL TEST FUNCTIONS

If $C : \mathcal{E} \times \mathcal{E} \to \mathbb{R}$ is a cost function defined over behavior space $\mathcal{E}$, and $\pi_1, \pi_2$ are two policies, then:

$$\text{WD}_\gamma(\mathbb{P}_{\pi_1}^\Phi, \mathbb{P}_{\pi_2}^\Phi) \approx \mathbb{E}_{\tau \sim \mathbb{P}_{\pi_1}} [\lambda_1^*(\Phi(\tau))] - \mathbb{E}_{\tau \sim \mathbb{P}_{\pi_2}} [\lambda_2^*(\Phi(\tau))], \quad (5)$$

where $\lambda_1^*, \lambda_2^*$ are the optimal dual functions. The maps $s_1 := \lambda_1^* \circ \Phi : \Gamma \to \mathbb{R}$ and $s_2 := \lambda_2^* \circ \Phi : \Gamma \to \mathbb{R}$ define score functions over the space of trajectories. If $\gamma$ is close to zero, the score function $s_i$ gives higher scores to trajectories from $\pi_i$ whose behavioral embedding is common under $\pi_i$ but rarely appears under $\pi_j$ for $j \neq i$ (Fig. 2).

### 5.2 ALGORITHMS

We propose to solve a WD-regularized objective to tackle behavior-guided policy optimization. All of our algorithms hinge on trying to maximize an objective of the form:

$$F(\theta) = \mathcal{L}(\theta) + \beta \text{WD}_\gamma(\mathbb{P}_{\pi_\theta}^\Phi, \mathbb{P}_b^\Phi), \quad (6)$$

where $\mathbb{P}_b^\Phi$ is a base distribution over behavioral embeddings (possibly dependent on $\theta$) and $\beta \in \mathbb{R}$ could be positive or negative. Although the base distribution $\mathbb{P}_b^\Phi$ could be arbitrary, our algorithms will instantiate $\mathbb{P}_b^\Phi = \frac{1}{|\mathcal{S}|} \cup_{\pi' \in \mathcal{S}} \mathbb{P}_{\pi'}^\Phi$ for some family of policies $\mathcal{S}$ (possibly satisfying $|\mathcal{S}| = 1$) we want the optimization to attract to / repel from.

In order to compute approximate gradients for $F$, we rely on the dual formulation of the WD. After substituting the composition maps resulting from Eq. 5 into Eq. 2, we obtain:

$$F(\theta) \approx \mathbb{E}_{\tau \sim \mathbb{P}_{\pi_\theta}} [\mathcal{R}(\tau) + \beta s_1(\tau)] - \beta \mathbb{E}_{\phi \sim \mathbb{P}_b^\Phi} [\lambda_2^*(\phi)], \quad (7)$$

where $s_1 : \Gamma \to \mathbb{R}$ equals $s_1 = \lambda_1^* \circ \Phi$, the Behavioral Test Function of policy $\pi_\theta$ and $\lambda_2^*$ is the optimal dual function of embedding distribution $\mathbb{P}_b^\Phi$. Consequently $\nabla_\theta F(\theta) \approx \nabla_\theta \mathbb{E}_{\tau \sim \mathbb{P}_{\pi_\theta}} [\mathcal{R}(\tau) + \beta s_1(\tau)]$. We learn a score function $s_1$ over trajectories that can guide our optimization by favoring those trajectories that show desired global behaviors.

Eq. 7 is an approximation to the true objective from Eq. 2 whenever $\gamma > 0$. In practice, the entropy regularization requires a damping term as defined in Equation 3. If $\xi(\mathbb{P}_{\pi_\theta}^\Phi, \mathbb{P}_b^\Phi)$ is the joint distribution of choice then $F(\theta) = \mathcal{L}(\theta) + \beta V$ for

$$V = \max_{\lambda_{\pi_\theta} \in \mathcal{C}(\mathcal{E}), \lambda_b \in \mathcal{C}(\mathcal{E})} \mathbb{E}_{\tau \sim \mathbb{P}_{\pi_\theta}} [\lambda_{\pi_\theta}(\Phi(\tau))] - \mathbb{E}_{\phi \sim \mathbb{P}_b^\Phi} [\lambda_b(\phi)] + \gamma \mathbb{E}_{\phi_1, \phi_2 \sim \xi(\mathbb{P}_{\pi_\theta}^\Phi, \mathbb{P}_b^\Phi)} [\Lambda(\phi_1, \phi_2)],$$

where $\Lambda(\phi_1, \phi_2) = \exp\left( \frac{\lambda_{\pi_\theta}(\phi_1) - \lambda_b(\phi_2) - C(\phi_1, \phi_2)}{\gamma} \right)$. When the embedding space $\mathcal{E}$ is not discrete and $\mathbb{P}_b^\Phi = \mathbb{P}_\pi^\Phi$ for some policy $\pi$, we let $\xi(\mathbb{P}_{\pi_\theta}^\Phi, \mathbb{P}_b^\Phi) = \mathbb{P}_{\pi_\theta}^\Phi \bigotimes \mathbb{P}_\pi^\Phi$, otherwise $\xi(\mathbb{P}_{\pi_\theta}^\Phi, \mathbb{P}_b^\Phi) = \frac{1}{|\mathcal{E}|^2} \mathbf{1}$, a uniform distribution over $\mathcal{E} \times \mathcal{E}$.

All of our methods perform a version of alternating SGD optimization: we take certain number of SGD steps over the internal dual Wasserstein objective, followed by more SGD steps over the outer objective having fixed the current dual functions. Although in practice the different components

that make up the optimization objectives we consider here could be highly nonconvex, in the cases these functions satisfy some convexity assumptions, we can provide a sharp characterization for the convergence rates of our algorithms. Details are given in Section 10 in the Appendix.

We consider two distinct approaches to optimizing this objective, by exploring in the action space and backpropagating, as in policy gradient methods (Schulman et al., 2015; 2017), and by considering a black-box optimization problem as in Evolution Strategies (ES, Salimans et al. (2017)). These two different approaches lead to two new algorithms: Behavior-Guided Policy Gradient (BGPG) and Behavior-Guided Evolution Strategies (BGES), that we discuss next.

## 5.3 BEHAVIOR-GUIDED POLICY GRADIENT (BGPG)

Our first algorithm seeks to solve the optimization problem in Section 5.2 with policy gradients. We refer to this method as the Behavior-Guided Policy Gradient (BGPG) algorithm (see Algorithm 2 below).

---

**Algorithm 2** Behvaior-Guided Policy Gradient

**Input:** Initialize stochastic policy $\pi_0$ parametrized by $\theta_0$, $\beta < 0, \eta > 0$, $M, L \in \mathbb{N}$

**for** $t = 1, \ldots, T$ **do**

   1. Run $\pi_{t-1}$ in the environment to get advantage values $A^{\pi_{t-1}}(s, a)$ and trajectories $\{\tau_i^{(t)}\}_{i=1}^M$

   2. Update policy and test functions via several alternating gradient steps over the objective:

$$F(\theta) = \mathbb{E}_{\tau_1, \tau_2 \sim \mathbb{P}_{\pi_{t-1}} \otimes \mathbb{P}_{\pi_\theta}} \left[ \sum_{i=1}^H A^{\pi_{t-1}}(s_i, a_i) \frac{\pi_\theta(a_i|s_i)}{\pi_{t-1}(a_i|s_i)} + \beta\lambda_1(\Phi(\tau_1)) \right.$$
$$\left. - \beta\lambda_2(\Phi(\tau_2)) + \beta\gamma \exp\left( \frac{\lambda_1(\Phi(\tau_1)) - \lambda_2(\Phi(\tau_2)) - C(\Phi(\tau_1)), \Phi(\tau_2))}{\gamma} \right) \right]$$

   Where $\tau_1 = s_0, a_0, r_0, \cdots, s_H, a_H, r_H$. Let $\theta_{t-1}^{(0)} = \theta_{t-1}$.

   **for** $\ell = 1, \cdots, L$ **do**

      a. Approximate $\mathbb{P}_{\pi_{t-1}} \otimes \mathbb{P}_{\pi_\theta}$ via $\frac{1}{M}\{\tau_i^{(t)}\}_{i=1}^M \otimes \frac{1}{M}\{\tau_i^\theta\}_{i=1}^M := \hat{P}_{\pi_t, \pi_\theta}$ where $\tau_i^\theta \overset{i.i.d}{\sim} \mathbb{P}_{\pi_\theta}$

      b. Take SGA step $\theta_{t-1}^{(\ell)} = \theta_{t-1}^{(\ell-1)} + \eta\hat{\nabla}_\theta \hat{F}(\theta_{t-1}^{(\ell-1)})$ using samples from $\hat{P}_{\pi_{t-1}, \pi_\theta}$.

      c. Use samples from $\hat{P}_{\pi_{t-1}, \pi_\theta}$ and Algorithm 1 to update $\lambda_1, \lambda_2$.

Set $\theta_t = \theta_{t-1}^{(M)}$.

---

Specifically, we maintain a stochastic policy $\pi_\theta$ and compute policy gradients as in prior work (Schulman et al., 2015). To optimize the Wasserstein distance $\text{WD}_\gamma$, we approximate the gradient of this term via the random-feature Wasserstein SGD . Importantly, this stochastic gradient can be approximated by samples collected from the policy $\pi_\theta$. In its simplest form, the $\hat{\nabla}_\theta \hat{F}$ in Step b. in Algorithm 2 can be computed by the vanilla policy gradient over the advantage component and using the reinforce estimator through the components involving Behavioral Test Functions acting on trajectories from $\mathbb{P}_{\pi_\theta}$. We explain in Appendix 8.1 a lower-variance gradient estimator alternative.

BGPG can be thought of as a variant of Trust Region Policy Optimization with a Wasserstein penalty. As opposed to vanilla TRPO, the optimization path of BGPG flows through policy parameter space while encouraging it to follow a smooth trajectory through the geometry of the PPE space. We proceed to show that given the right embedding and cost function, we can prove a monotonic improvement theorem for BGPG, showing that our methods satisfy at least similar guarantees as TRPO.

For a given policy $\pi$, we denote as: $V^\pi, Q^\pi$ and $A^\pi(s, a) = Q^\pi(s, a) - V^\pi(s)$ the: value function, $Q$-function and advantage function (see Appendix: Section 10.5). Furthermore, let $V(\pi)$ be the expected reward of policy $\pi$ and $\rho_\pi(s) = \mathbb{E}_{\tau \sim \mathbb{P}_\pi}\left[ \sum_{t=0}^T \mathbf{1}(s_t = s) \right]$ be the visitation measure.

Two distinct policies $\pi$ and $\tilde{\pi}$ can be related via the equation (see: Sutton et al. (1998)) $V(\tilde{\pi}) = V(\pi) + \int_S \rho_{\tilde{\pi}}(s) \left( \int_A \tilde{\pi}(a|s)A^\pi(s, a)da \right) ds$ and the linear approximations to $V$ around $\pi$ via: $L(\tilde{\pi}) =$

$V(\pi) + \int_{\mathcal{S}} \rho_\pi(s) \left( \int_{\mathcal{A}} \tilde{\pi}(a|s) A^\pi(s,a) da \right) ds$ (see: Kakade & Langford (2002)). Let $\mathcal{S}$ be a finite set. Consider the following embedding $\Phi^s : \Gamma \to \mathbb{R}^{|\mathcal{S}|}$ defined by $(\Phi(\tau))_s = \sum_{t=0}^{T} \mathbf{1}(s_t = s)$ and related cost function defined as: $C(\mathbf{v}, \mathbf{w}) = \|\mathbf{v} - \mathbf{w}\|_1$. Then $\mathrm{WD}_0(\mathbb{P}_{\tilde{\pi}}^{\Phi^s}, \mathbb{P}_{\pi}^{\Phi^s})$ is related to visitation frequencies since $\mathrm{WD}_0(\mathbb{P}_{\tilde{\pi}}^{\Phi^s}, \mathbb{P}_{\pi}^{\Phi^s}) \geq \sum_{s \in \mathcal{S}} |\rho_\pi(s) - \rho_{\tilde{\pi}}(s)|$ (see Section 10.5 for the proof). These observations enable us to prove an analogue of Theorem 1 from Schulman et al. (2015), namely:

**Theorem 5.1.** *If* $\mathrm{WD}_0(\mathbb{P}_{\tilde{\pi}}^{\Phi^s}, \mathbb{P}_{\pi}^{\Phi^s}) \leq \delta$ *and* $\epsilon = \max_{s,a} |A^\pi(s,a)|$, *then* $V(\tilde{\pi}) \geq L(\tilde{\theta}) - \delta\epsilon$.

As in Schulman et al. (2015), Theorem 5.1 implies a policy improvement guarantee for BGPG.

## 5.4 BEHAVIOR GUIDED EVOLUTION STRATEGIES (BGES)

ES takes a black-box optimization approach to RL, by considering a rollout of a policy, parameterized by $\theta$ as a black-box function $F$. This approach has gained in popularity recently (Salimans et al., 2017; Mania et al., 2018; Choromanski et al., 2019). If we take this approach to optimizing the objective in Eq. 2, the result is a black-box optimization algorithm which seeks to maximize the reward and simultaneously maximizes or minimizes the difference in behavior from the base embedding distribution $\mathbb{P}_b^\Phi$. We call this method the Behavior-Guided Evolution Strategies (BGES) algorithm (see Algorithm 3 below).

---

**Algorithm 3** Behavior-Guided Evolution Strategies

**Input:** learning rate $\eta$, noise standard deviation $\sigma$, iterations $T$, BEM $\Phi$, $\beta$
**Initialize:** Initial policy $\pi_0$ parametrized by $\theta_0$, Behavioral Test Functions $\lambda_1, \lambda_2$. Evaluate policy $\pi_0$ to return trajectory $\tau_0$ and subsequently use the BEM to produce an initial PPE $\hat{\mathbb{P}}_{\pi_0}^\Phi$.
**for** $t = 1, \ldots, T-1$ **do**
  1. Sample $\epsilon_1, \cdots, \epsilon_n$ independently from $\mathcal{N}(0, I)$.
  2. Evaluate policies $\{\pi_t^k\}_{k=1}^n$ parameterized by $\{\theta_t + \sigma\epsilon_k\}_{k=1}^n$ to return rewards $R_k$ and trajectories $\tau_k$ for all $k$.
  3. Use BEM to map trajectories $\tau_k$ to produce empirical PPEs $\hat{\mathbb{P}}_{\pi_t^k}^\Phi$ for all $k$.
  4. Update $\lambda_1$ and $\lambda_2$ using Algorithm 1, where $\mu = \frac{1}{n} \cup_{k=1}^n \hat{\mathbb{P}}_{\pi_{t-1}^k}^\Phi$ and $\nu = \frac{1}{n} \cup_{k=1}^n \hat{\mathbb{P}}_{\pi_t^k}^\Phi$ are the uniform distribution over the set of PPEs from 3 for $t-1$ and $t$.
  5. Approximate $\widehat{\mathrm{WD}}\gamma(\mathbb{P}_{\pi_t^k}^\Phi, \mathbb{P}_{\pi_t}^\Phi)$ plugging in $\lambda_1, \lambda_2$ into Eq. 4 for each perturbed policy $\pi_k$
  6. Update Policy: $\theta_{t+1} = \theta_t + \eta\nabla_{ES}F$, where:

$$\nabla_{ES}F = \frac{1}{\sigma} \sum_{k=1}^n [(1 - \beta)(R_k - R_t) + \beta\widehat{\mathrm{WD}}\gamma(\mathbb{P}_{\pi_t^k}^\Phi, \mathbb{P}_{\pi_t}^\Phi)]\epsilon_k$$

---

When $\beta > 0$, and we take $\mathbb{P}_b^\Phi = \mathbb{P}_{\pi_{t-1}}^\Phi$, BGES resembles the NSR-ES algorithm from Conti et al. (2018), an instantiation of *novelty search* (Lehman & Stanley, 2008). The positive weight on the WD-term enforces newly constructed policies to be behaviorally different from the previous ones (improving exploration) while the $\mathcal{R}$−term drives the optimization to achieve its main objective, i.e., maximize the reward. The key difference in our approach is the probabilistic embedding map, with WD rather than Euclidean distance. We show in Section 7.2 that BGES outperforms NSR-ES for challenging exploration tasks. The approximation introduced by Step 5 bypasses the need of computing a different pair of behavioral test functions $\lambda_1, \lambda_2$ for each perturbed policy $\pi_k$.

If we take $\beta < 0$, and assume $\mathbb{P}_b^\Phi = \mathbb{P}_\pi^\Phi$ to correspond to embedded trajectories from an oracle or expert policy, we can perform imitation learning. Despite not accessing the expert's policy (just the trajectories it generates), we show in Section 7.3 that this approach dramatically improves learning.

## 6 RELATED WORK

Our work is related to research in multiple areas in neuroevolution and machine learning:

**Behavior Characterizations:** The idea of directly optimizing for behavioral diversity was introduced by Lehman & Stanley (2008) and Lehman (2012), who proposed to search directly for *novelty*,

rather than simply assuming it would naturally arise in the process of optimizing an objective function. This approach has been applied to deep RL (Conti et al., 2018) and meta-learning (Gajewski et al., 2019). In all of this work, the policy is represented via a behavioral characterization (BC), typically chosen with knowledge of the environment, for example the final (x,y) coordinate for a locomotion task. Additionally, in most cases these BCs are considered to be deterministic, with Euclidean distances used to compare BCs. In our setting, we move from deterministic BCs to stochastic PPEs, thus requiring the use of metrics capable of comparing probabilistic distributions.

**Distance Metrics:** WDs have been used in many different applications in machine learning where guarantees based on distributional similarity are required (Jiang et al., 2019; Arjovsky et al., 2017). We make use of WDs in our setting for a variety of reasons. First and foremost, the dual formulation of the WD allows us to recover Behavioral Test Functions, thus providing us with behavior-driven trajectory scores. In contrast to KL divergences, WDs are sensitive to user-defined costs between pairs of samples instead of relying only on likelihood ratios. Furthermore, as opposed to KL divergences, it is possible to take SGD steps using entropy-regularized Wasserstein objectives. Computing an estimator of the KL divergence is hard without a density model. Since in our framework multiple unknown trajectories may map to the same behavioral embedding, the likelihood ratio between two embedding distributions may be ill-defined.

**WDs for RL:** We are not the first to propose using WDs in RL. Zhang et al. (2018) have recently introduced Wasserstein Gradient Flows (WGFs) for finding efficient RL policies. This approach casts policy optimization as gradient descent flow on the manifold of corresponding probability measures, where geodesic lengths are given as second-order WDs. We note that computing WGFs is a nontrivial task. In Zhang et al. (2018) this is done via particle approximation methods. We show in Section 7 that RL algorithms using these techniques are substantially slower than our methods. The WD has also been employed to replace KL terms in standard Trust Region Policy Optimization (Richemond & Maginnis, 2017). This is a very special case of our more generic framework (cf. Section 5.2). In Richemond & Maginnis (2017) it is suggested to solve the corresponding RL problems via Fokker-Planck equations and diffusion processes, yet no empirical evidence of the feasibility of this approach is provided. We propose general practical algorithms and provide extensive empirical evaluation.

**Distributional RL** Distributional RL (DRL, Bellemare et al. (2017)) expands on traditional off-policy methods (Mnih et al., 2013) by attempting to learn a distribution of the return from a given state, rather than just the expected value. These approaches have impressive experimental results (Bellemare et al., 2017; Dabney et al., 2018), with a growing body of theory (Rowland et al., 2018; Qu et al., 2019; Bellemare et al., 2019; Rowland et al., 2019). Superficially it may seem that learning a distribution of returns is similar to our approach to PPEs, when the BEM is a distribution over rewards. Indeed, reward-driven embeddings used in DRL can be thought of as special cases of the general class of BEMs. We note two key differences: 1) DRL methods are off-policy whereas our BGES and BGPG algorithms are on-policy, and 2) DRL is typically designed for discrete domains, since Q-Learning with continuous action spaces is generally much harder. Furthermore, we note that while the WD is used in DRL, it is only for the convergence analysis of the DRL algorithm—the algorithm itself does not use WDs (Bellemare et al., 2017).

## 7 EXPERIMENTS

Here we seek to test whether our behavior-guided approach to RL translates to performance gains for simulated environments. We individually evaluate our two proposed algorithms, BGPG and BGES, versus their respective baselines for a range of benchmark tasks. While in some cases the results may not be state of the art, we believe the improvement vs. popular RL algorithms (in particular TRPO and ES) are exciting results which could stimulate future work. We also include a study of using our method for imitation learning. For each subsection we provide additional details in the Appendix.

### 7.1 BEHAVIOR-GUIDED POLICY GRADIENT

Our key question is whether BGPG can outperform baseline TRPO methods using KL divergence. In Fig. 3, we see this is clearly the case for four continuous control tasks: Pendulum from OpenAI Gym and Hopper: Stand, Hooper: Hop and Walker: Stand from the DeepMind Control Suite (Tassa et al., 2018). For the BEM, we use the concatenation-of-actions (as used already in TRPO). We also

confirm results from (Schulman et al., 2015) that a trust region greatly improves performance, as we see the black curve (without one) often fails to learn.

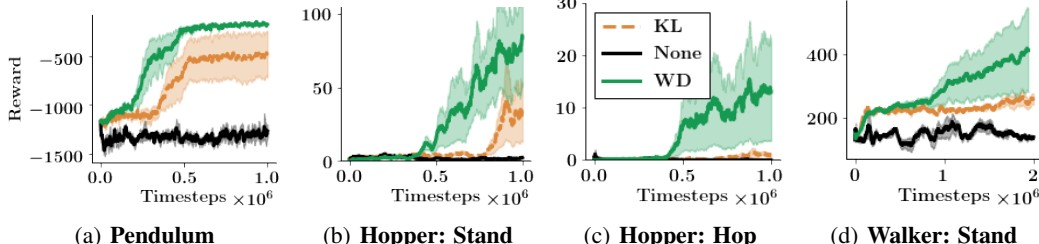

(a) **Pendulum** (b) **Hopper: Stand** (c) **Hopper: Hop** (d) **Walker: Stand**

Figure 3: **BGPG vs. TRPO:** We compare BGPG and TRPO (KL divergence) on several continuous control tasks. As a baseline we also include results without a trust region ($\beta = 0$ in Algorithm 2). Plots show the mean $\pm$ std across 5 random seeds. BGPG consistently outperforms other methods.

**Wall Clock Time:** To illustrate computational benefits of alternating optimization (AO) of WD in BGPG, we compare it to the particle approximation (PA) method introduced in Zhang et al. (2018) in Fig. 4. In practice, the WD across different state samples can be optimized in a batched manner using AO (see Appendix for details). We see that AO is substantially faster than PA.

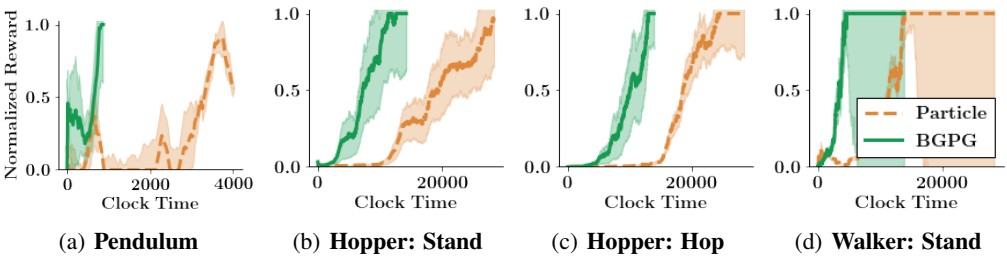

(a) **Pendulum** (b) **Hopper: Stand** (c) **Hopper: Hop** (d) **Walker: Stand**

Figure 4: The clock-time comparison (in sec) of BGPG (alternating optimization) with particle approximation.

## 7.2 BEHAVIOR-GUIDED EVOLUTION STRATEGIES

As a novelty-search method, BGES is designed to actively explore the environment by behaving differently for previous policies. With that in mind, we seek to evaluate the ability to solve two key challenges in exploration for RL: deceptive rewards and local maxima.

**Deceptive Rewards** A common challenge in model-free RL is *deceptive* rewards. These arise since agents can only learn from data gathered via exploration in the environment. To test BGES in this setting, we created two intentionally deceptive environments where agents may easily be fooled into learning suboptimal policies. In both cases the agent is penalized at each time step for being far away from a goal. The deception comes from a wall situated in the middle, which means that initially positive rewards from moving directly forward will lead to a suboptimal policy.

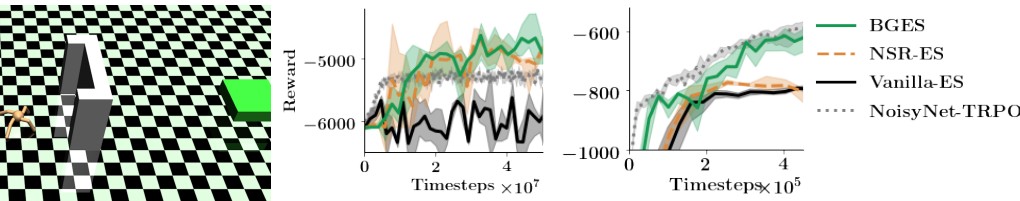

Figure 5: **Efficient Exploration.** On the left we show a visualization of the simulated environment, with the deceptive barrier between the (quadruped) agent and the goal. On the right, we show two plots with the median curve across five seeds, with the IQR shaded for the quadruped and point environment respectively.

We consider two types of agents—a two-dimensional point and a much larger quadruped. Details are provided in the Appendix (Section 9). We compare with state-of-the-art on-policy methods for efficient exploration: NSR-ES from (Conti et al., 2018), which assumes the BEM is deterministic and uses the Euclidean distance to compare policies, and NoisyNet-TRPO from Fortunato et al. (2018). Results are presented on Fig. 5. Policies avoiding the wall correspond to rewards: $R > -5000$ and $R > -800$ for the quadruped and point respectively. In the prior case an agent needs to first learn how to walk and the presence of the wall is enough to prohibit vanilla ES from even learning forward locomotion. We note that BGES is the only method that drives the agent to the goal in *both* settings. For the quadruped the BEM is the reward-to-go while for the point we used the final state.

**Escaping Local Maxima.** In Fig. 6 we compare our methods with methods using regularizers based on other distances or divergences (specifically, Hellinger, Jensen-Shannon (JS), KL and Total Variation (TV) distances), as well as vanilla ES (i.e., with no distance regularizer). Experiments were performed on a Swimmer environment from OpenAI Gym (Brockman et al., 2016), where the number of samples of the ES optimizer was drastically reduced. BGES is the only one that manages to obtain good policies which also proves that the benefits come here not just from introducing the regularizer, but from its particular form.

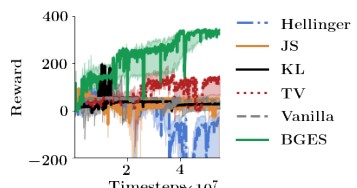

Figure 6: **Escaping Local Maxima.** A comparison of BGES with those using different distances on PPEs.

### 7.3 IMITATION LEARNING

As discussed in Section 5.3, we can also utilize the BGES algorithm for imitation learning, by setting $\beta < 0$, and using an expert's trajectories for the PPE. For this experiment we use the reward-to-go BEM (Section 5). In Fig. 7, we show that this approach significantly outperforms vanilla ES on the Swimmer task. Although conceptually simple, we believe this could be a powerful approach with potential extensions, for example in designing safer algorithms.

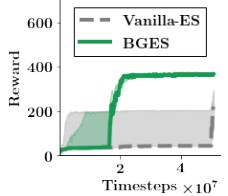

Figure 7: **Imitation Learning.**

### 7.4 HYPERPARAMETER SELECTION

Our approach includes several new hyperparameters, such as the kernel for the Behavioral Test Functions and the choice of BEM. For our experiments we did not perform any hyperparameter optimization. We only considered the rbf kernel, and only varied the BEM for BGES. For BGPG we chose the concatenation of actions, since this is the same as used in the KL divergence for TRPO. For BGES, we demonstrated several different BEMs, and we show an ablation study for the point agent in Fig. 8 where we see that both the reward-to-go (RTG) and Final State (SF) worked, but the vector of all states (SV) did not (for 5 seeds). We leave learned BEMs as exciting future work.

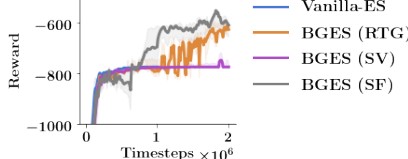

Figure 8: **Choice of BEM**

## 8 CONCLUSION AND FUTURE WORK

In this paper we proposed a new paradigm for on-policy learning in RL, where policies are embedded into expressive latent behavioral spaces and the optimization is conducted by utilizing the repelling/attraction signals in the corresponding probabilistic distribution spaces. The use of Wasserstein distances (WDs) guarantees flexibility in choosing cost funtions between embedded policy trajectories, enables stochastic gradient steps through corresponding regularized objectives (as opposed to KL divergence methods) and provides an elegant method, via their dual formulations, to quantify behaviorial difference of policies through the behavioral test functions. Furthermore, the dual formulations give rise to efficient algorithms optimizing RL objectives regularized with WDs.

We also believe the presented methods shed new light on several other challenging problems of modern RL, including: learning with safety guarantees (a repelling signal can be used to enforce behaviors away from dangerous ones) or anomaly detection for reinforcement learning agents (via the above score functions). We are also excited by the possibility of scaling this approach to a population setting, learning the behavioral embedding maps from data, or adapting the degree of repulsion/attraction during optimization (parameter $\beta$).

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

APPENDIX: BEHAVIOR-GUIDED REINFORCEMENT LEARNING

# 9 FURTHER EXPERIMENTAL DETAILS

## 9.1 BGPG

**A Lower-variance Gradient Estimator:** As explained in Section 5.2, the BGPG considers an objective which involves two parts: the conventional surrogate loss function for policy optimization (Schulman et al., 2017), and a loss function that involves the Behavior Test Functions. Though we could apply vanilla reinforced gradients on both parts, it is straightforward to notice that the second part can be optimized with reparameterized gradients (Kingma & Welling, 2013), which arguably have lower variance compared to the reinforced gradients. In particular, we note that under random feature approximation (4), as well as the action-concatenation embedding, the Wasserstein distance loss $\widehat{\mathrm{WD}}_\gamma(P_{\pi_\theta}^\Phi, P_b^\Phi)$ is a differentiable function of $\theta$. To see this more clearly, notice that under a Gaussian policy $a \sim \mathcal{N}(\mu_\theta(s), \sigma_\theta(s)^2)$ the actions $a = \mu_\theta(s) + \sigma_\theta(s) \cdot \epsilon$ are reparametrizable for $\epsilon$ being standard Gaussian noises. We can directly apply the reparametrization trick to this second objective to obtain a gradient estimator with potentially much lower variance. In our experiments, we applied this lower-variance gradient estimator.

**Trust Region Policy Optimization:** Though the original TRPO (Schulman et al., 2015) construct the trust region based on KL-divergence, we propose to construct the trust region with WD. For convenience, we adopt a dual formulation of the trust region method and aim to optimize the augmented objective $\mathbb{E}_{\tau \sim \pi_\theta}[R(\tau)] - \beta \mathrm{WD}_\gamma(\mathbb{P}_{\pi'}^\Phi, \mathbb{P}_{\pi_\theta}^\Phi)$. We apply the concatenation-of-actions embedding and random feature maps to calculate the trust region. We identify several important hyperparameters: the RKHS (for the test function) is produced by RBF kernel $k(x, y) = \exp(\|x - y\|_2^2/\sigma^2)$ with $\sigma = 0.1$; the number of random features is $D = 100$; recall the embedding is $\Phi(\tau) = [a_1, a_2...a_H]$ where $H$ is the horizon of the trajectory, here we take 10 actions per state and embed them together, this is equivalent to reducing the variance of the gradient estimator by increasing the sample size; the regularized entropy coefficient in the WD definition as $\gamma = 0.1$; the trust region trade-off constant $\beta \in \{0.1, 1, 10\}$. The alternate gradient descent is carried out with $T = 100$ alternating steps and test function coefficients $\mathbf{p} \in \mathbb{R}^D$ are updated with learning rate $\alpha_{\mathbf{p}} = 0.01$.

The baseline algorithms are: No trust region, and trust region with KL-divergence. The KL-divergence is identified by a maximum KL-divergence threshold per update, which we set to $\epsilon = 0.01$.

Across all algorithms, we adopt the open source implementation (Dhariwal et al., 2017). Hyperparameters such as number of time steps per update as well as implementation techniques such as state normalization are default in the original code base.

The additional experiment results can be found in Figure 9 where we show comparison on additional continuous control benchmarks: Tasks with DM are from DeepMind Contol Suites (Tassa et al., 2018). We see that the trust region constructed from the WD consistently outperforms other baselines (importantly, trust region methods are always better than the baseline without trust region, this confirms that trust region methods are critical in stabilizing the updates).

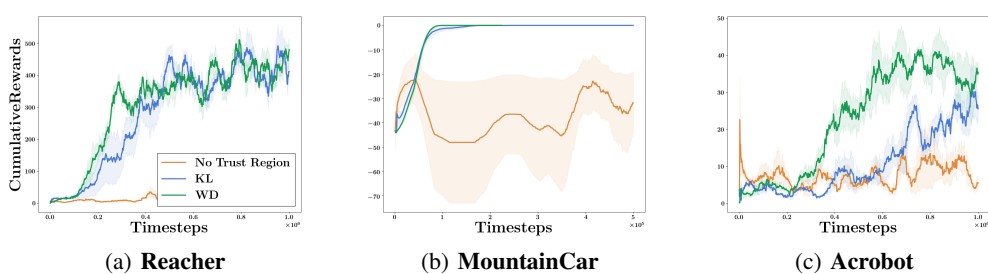

(a) **Reacher**   (b) **MountainCar**   (c) **Acrobot**

Figure 9: Additional Experiment on TRPO. We compare No Trust Region with two alternative trust region constructions: KL-divergence and Wassertein distance (ours).

**Wasserstein AO vs. Particle Approximation:** To calculate the regularized Wasserstein distance, we propose a gradient descent method that iteratively updates the test function. The alternting optimization (AO) scheme consists of updating both the test function and the distribution parameters such that the regularized Wasserstein distance of the trainable distribution against the reference distribution is minimized. Alternatively, we can also adopt a particle approximation method to calculate the Wasserstein distance and update the distribution parameters using an approximate gradient descent method (Zhang et al., 2018).

One major advantage of AO against particle approximation is its ease of parallization. In particular, when using the concatenation-of-actions embedding, the aggregate Wasserstein distance can be decomposeed into an average of a set of Wasserstein distances over states. To calculate this aggregated gradient, AO can easily leverage the matrix multiplication; on the other hand, particle approximation requires that the dual optimal variables of each subproblem be computed, which is not straightforward to parallelize.

We test both methods in the context of trust region policy search, in which we explicitly calculate the Waserstein distance of consecutive policies and enforce the constraints using a line search as in (Schulman et al., 2015). Both methods require the trust region trade-off parameter $\beta \in \{0.1, 1, 10\}$. We adopt the particle method in (Zhang et al., 2018) where for each state there are $M = 16$ particles. The gradients are derived based a RKHS where we adaptively adjust the coefficient of the RBF kernel based on the mean distance between particles. For the AO, we find that it suffices to carry out $T \in \{1, 5, 10\}$ gradient descents to approximate the regularized Wasserstein distance.

## 9.2 BGES

**Efficient Exploration:** To demonstrate the effectiveness of our method in exploring deceptive environments, we constructed two new environments using the MuJoCo simulator. For the point environment, we have a 6 dimensional state and 2 dimensional action, with the reward at each timestep calculated as the distance between the agent and the goal. We use a horizon of 50 which is sufficient to reach the goal. The quadruped environment is based on Ant from the Open AI Gym (Brockman et al., 2016), and has a similar reward structure to the point environment but a much larger state space (113) and action space (8). For the quadruped, we use a horizon length of 400.

To leverage the trivially parallelizable nature of ES algorithms, we use the ray library, and distribute the rollouts across 72 workers using AWS. Since we are sampling from an isotropic Gaussian, we are able to pass only the seed to the workers, as in Salimans et al. (2017). However we do need to return trajectory information to the master worker.

For both the point and quadruped agents, we use random features with dimensionality $m = 1000$, and 100 warm-start updates for the WD at each iteration. For point, we use the final state embedding, learning rate $\eta = 0.1$ and $\sigma = 0.01$. For the quadruped, we use the reward-to-go embedding, as we found this was needed to learn locomotion, as well as a learning rate of $\eta = 0.02$ and $\sigma = 0.02$. The hyper-parameters were the same for all ES algorithms. When computing the WD, we used the previous 2 policies, $\theta_{t-1}$ and $\theta_{t-2}$. An ablation study for the point environment for both the choice of embedding and number of prior policies is shown in Fig 12.

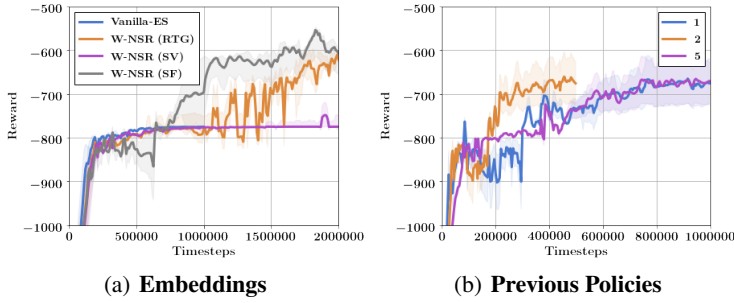

(a) **Embeddings**   (b) **Previous Policies**

Figure 10: A sensitivity analysis investigating a) the impact of the embedding and b) the number of previous policies $\theta_{t-i}, i \in 1, 2, 5$

For embeddings, we compare the reward-to-go (RTG), concatenation of states (SV) and final state (SF). In both the RTG and SF case the agent learns to navigate past the wall ($> -800$). For the number of previous policies, we use the SF embedding, and using $2$ appears to work best, but both $1$ and $5$ do learn the correct behavior.

**Escaping Local Maxima:** We also demonstrated that our method leads to faster training even in more standard settings, where exploration is not that crucial, but the optimization can be trapped in local maxima. To show it, we compared baseline ES algorithm for ES optimization from Salimans et al. (2017) with its enhancements, where regularizers using different metrics on the space of probabilistic distributions corresponding to policy embeddings were used, as in the previous paragraph. We noticed that adding Wasserstein regularizers drastically improved optimization, whereas regularizers based on other distances/divergencies, namely: Hellinger, Jensen-Shannon, KL and TV did not have any impact. We considered $\mathrm{Swimmer}$ task from $\mathrm{OpenAI}$ $\mathrm{Gym}$ and to make it challenging, reduced the number of perturbations per iteration to $80$. In that setting our method was the only one that was not trapped in local maxima and managed to learn effective policies.

## 9.3 IMITATION LEARNING:

For the Imitation Learning experiment we used the reward-to-go embedding, with learning rate $\eta = 0.1$ and $\sigma = 0.01$. We use one oracle policy, which achieves $> 360$ on the environment. The only information provided to the algorithm is the embedded trajectory, used to compute the WD. This has exciting future applications since no additional information about the oracle is required in order to significantly improve learning.

## 9.4 REPULSION LEARNING

---

**Algorithm 4** Behvaior-Guided Repulsion Learning

---

**Input:** $\beta, \eta > 0$, $M \in \mathbb{N}$
**Initialize:** Initial stochastic policies $\pi_0^{\mathbf{a}}, \pi_0^{\mathbf{b}}$, parametrized by $\theta_0^{\mathbf{a}}, \theta_0^{\mathbf{b}}$ respectively, Behavioral Test Functions $\lambda_1^{\mathbf{a}}, \lambda_2^{\mathbf{b}}$
**for** $t = 1, \ldots, T$ **do**

    1. Collect $M$ trajectories $\{\tau_i^{\mathbf{a}}\}_{i=1}^M$ from $\mathbb{P}_{\pi_{t-1}^{\mathbf{a}}}$ and $M$ trajectories $\{\tau_i^{\mathbf{b}}\}_{i=1}^M$ from $\mathbb{P}_{\pi_{t-1}^{\mathbf{b}}}$.
    Approximate $\mathbb{P}_{\pi_{t-1}^{\mathbf{a}}} \bigotimes \mathbb{P}_{\pi_{t-1}^{\mathbf{b}}}$ via $\frac{1}{M}\{\tau_i^{\mathbf{a}}\}_{i=1}^M \bigotimes \frac{1}{M}\{\tau_i^{\mathbf{b}}\}_{i=1}^M := \hat{P}_{\pi_{t-1}^{\mathbf{a}}, \pi_{t-1}^{\mathbf{b}}}$
    2. Form two distinct surrogate rewards for joint trajectories of agents $\mathbf{a}$ and $\mathbf{b}$:

$$\tilde{R}_{\mathbf{a}}(\tau_1, \tau_2) = \mathcal{R}(\tau_1) + \beta\lambda_1^{\mathbf{a}}(\Phi(\tau_1)) + \beta\gamma \exp\left(\frac{\lambda_1^{\mathbf{a}}(\Phi(\tau_1)) - \lambda_2^{\mathbf{b}}(\Phi(\tau_2)) - C(\Phi(\tau_1)), \Phi(\tau_2))}{\gamma}\right)$$

$$\tilde{R}_{\mathbf{b}}(\tau_1, \tau_2) = \mathcal{R}(\tau_2) - \beta\lambda_2^{\mathbf{b}}(\Phi(\tau_2)) + \beta\gamma \exp\left(\frac{\lambda_1^{\mathbf{a}}(\Phi(\tau_1)) - \lambda_2^{\mathbf{b}}(\Phi(\tau_2)) - C(\Phi(\tau_1)), \Phi(\tau_2))}{\gamma}\right)$$

    3. For $\mathbf{c} \in \{\mathbf{a}, \mathbf{b}\}$ use the Reinforce estimator to take gradient steps:

$$\theta_t^{\mathbf{c}} = \theta_{t-1}^{\mathbf{c}} + \eta \underset{\tau^{\mathbf{a}}, \tau^{\mathbf{b}} \sim \hat{P}_{\pi_{t-1}^{\mathbf{a}}, \pi_{t-1}^{\mathbf{b}}}}{\mathbb{E}} \left[\tilde{R}_{\mathbf{c}}(\tau^{\mathbf{a}}, \tau^{\mathbf{b}})\left(\sum_{i=0}^{H-1} \nabla_{\theta_{t-1}^{\mathbf{b}}} \log\left(\pi_{t-1}^{\mathbf{c}}(a_i^{\mathbf{c}}|s_i^{\mathbf{c}})\right)\right)\right]$$

    Where $\tau^{\mathbf{a}} = s_0^{\mathbf{a}}, a_0^{\mathbf{a}}, r_0^{\mathbf{a}}, \cdots, s_H^{\mathbf{a}}, a_H^{\mathbf{a}}, r_H^{\mathbf{a}}$ and $\tau^{\mathbf{b}} = s_0^{\mathbf{b}}, a_0^{\mathbf{b}}, r_0^{\mathbf{b}}, \cdots, s_H^{\mathbf{b}}, a_H^{\mathbf{b}}, r_H^{\mathbf{b}}$.
    5. Use samples from $\hat{P}_{\pi_{t-1}^{\mathbf{a}}, \pi_{t-1}^{\mathbf{b}}}$ and Algorithm 1 to update the Behavioral Test Functions $\lambda_1^{\mathbf{a}}, \lambda_2^{\mathbf{b}}$.

---

Although this was not discussed in the main section of the paper, it is possible to use our behavioral approach to simultaneously learn multiple policies exhibiting different behaviors all of which are able to solve the same task. This is not the main focus of the main paper, but we chose to include these results in an attempt to provide the readers with a better understanding of Behavioral Test functions.

Algorithm 4 maintains two policies $\pi^{\mathbf{a}}$ and $\pi^{\mathbf{b}}$. Each policy is optimized by taking a policy gradient step (using the Reinforce gradient estimator) in the direction optimizing surrogate rewards $\tilde{\mathcal{R}}_{\mathbf{a}}$ and $\tilde{\mathcal{R}}_{\mathbf{b}}$ that combines the signal from the task's reward function $\mathcal{R}$ and the repulsion score encoded by the behavioral test functions $\lambda^{\mathbf{a}}$ and $\lambda^{\mathbf{b}}$.

We conducted experiments testing Algorithm 4 on a simple Mujoco environment consisting of a particle that moves on the plane and whose objective is to learn a policy that allows it to reach one of two goals. Each policy outputs a velocity vector and stochasticity is achieved by adding Gaussian noise to the mean velocity encoded by a neural network with two size 5 hidden layers and ReLu activations. If an agent performs action $a$ at state $s$, it moves to state $a + s$. The reward of an agent after performing action $a$ at state $s$ equals $-\|a\|^2 * 30 - \min(d(s, \text{Goal}_1), d(s, \text{Goal}_2))^2$ where $d(x, y)$ denotes the distance between $x$ and $y$ in $\mathbb{R}^2$. The initial state is chosen by sampling a Gaussian distribution with mean $\binom{0}{0}$ and diagonal variance $0.1$. In each iteration step we sample $100$ trajectories. In the following pictures we plot the policies' behavior by plotting $100$ trajectories of each. The embedding $\Phi : \Gamma \to \mathbb{R}$ maps trajectories $\tau$ to their mean displacement in the $x-$axis. We use the squared absolute value difference as the cost function.

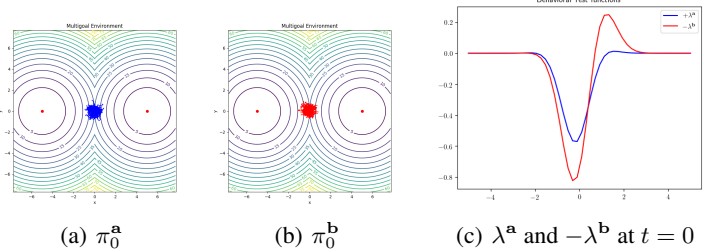

(a) $\pi_0^{\mathbf{a}}$      (b) $\pi_0^{\mathbf{b}}$      (c) $\lambda^{\mathbf{a}}$ and $-\lambda^{\mathbf{b}}$ at $t = 0$

Figure 11: Initial state of policies $\pi^{\mathbf{a}}, \pi^{\mathbf{b}}$ and Behavioral Test functions $\lambda^{\mathbf{a}}, \lambda^{\mathbf{b}}$ in the Multigoal environment.

There are two optimal policies, moving the particle to the left goal or moving it to the right goal. We now plot how the policies' behavior and evolves throughout optimization and how the Behavioral Test Functions guide the optimization by favouring the two policies to be far apart.

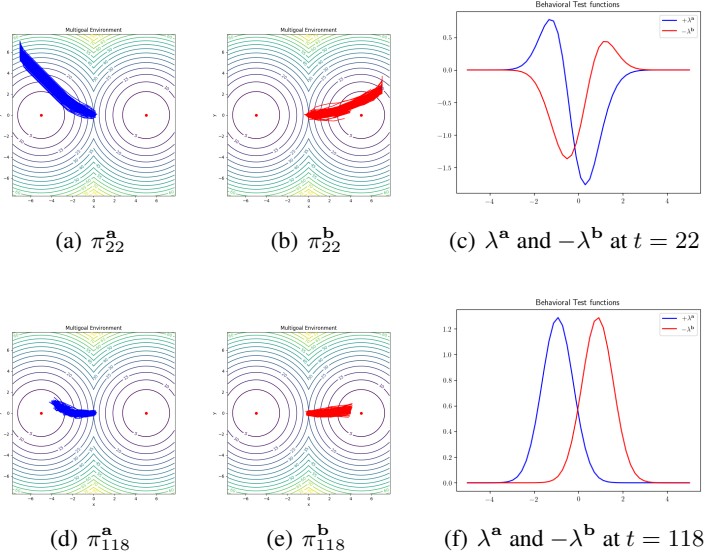

(a) $\pi_{22}^{\mathbf{a}}$      (b) $\pi_{22}^{\mathbf{b}}$      (c) $\lambda^{\mathbf{a}}$ and $-\lambda^{\mathbf{b}}$ at $t = 22$

(d) $\pi_{118}^{\mathbf{a}}$      (e) $\pi_{118}^{\mathbf{b}}$      (f) $\lambda^{\mathbf{a}}$ and $-\lambda^{\mathbf{b}}$ at $t = 118$

Figure 12: Evolution of the policies and Behavioral Test Functions throughout optimization.

Let $\mathcal{X}$ and $\mathcal{Y}$ be the domains of two measures $\mu$, and $\nu$. Recall that in case $\gamma = 0$, $\mathcal{X} = \mathcal{Y}$, and $C(x, x) = 0$ for all $x \in \mathcal{X}$, then $\lambda_\mu^*(x) = \lambda_\nu^*(x) = \lambda^*(x)$ for all $x \in \mathcal{X}$. In the case of regularized Wasserstein distances with $\gamma > 0$, this relationship may not hold true even if the cost satisfies the same diagonal assumption. For example when the regularizing measure is the product measure, and $\mu, \nu$ have disjoint supports, since the soft constraint $\gamma \exp\left(\frac{\lambda_\mu(\mathbf{x}) - \lambda_\nu(\mathbf{y}) - C(\mathbf{x}, \mathbf{y})}{\gamma}\right)$ is enforced in expectation over the product measure there may exist optimal solutions $\lambda_\mu^*, \lambda_\nu^*$ that do not satisfy $\lambda_\mu^* = \lambda_\nu^*$.

## 10 THEORETICAL RESULTS

We start by exploring some properties of the Wasserstein distance and its interaction with some simple classes of embeddings. The first lemma we show has the intention to show conditions under which two policies can be shown to be equal provided the Wasserstein distance between its trajectory embeddings is zero. This result implies that our framework is capable of capturing equality of policies when the embedding space equals the space of trajectories.

**Lemma 10.1.** *Let $\mathcal{S}$ and $\mathcal{A}$ be finite sets, the* MDP *be episodic (i.e. of finite horizon $H$), and* $\Phi(\tau) = \sum_{t=0}^H e_{s_t, a_t}$ *with $e_{s,a} \in \mathbb{R}^{|\mathcal{S}| + |\mathcal{A}|}$ the indicator vector for the state action pair $(s, a)$. Let* $C(\mathbf{v}, \mathbf{w}) = \|\mathbf{v} - \mathbf{w}\|_p^p$ *for $p \geq 1$. If $\gamma = 0$ and $\mathrm{WD}_\gamma(\mathbb{P}_\pi^\Phi, \mathbb{P}_{\pi'}^\Phi) = 0$ then $\pi = \pi'$.*

*Proof.* If $\mathrm{WD}_\gamma(\mathbb{P}_\pi^\Phi, \mathbb{P}_{\pi'}^\Phi) = 0$, there exists a coupling $\Pi$ between $\mathbb{P}_\pi^\Phi$ and $\mathbb{P}_{\pi'}^\Phi$ such that:
$$\mathbb{E}_{u,v \sim \Pi}\left[\|u - v\|_p^p\right] = 0$$
Consequently:
$$\mathbb{E}_{u,v \sim \Pi}\left[\sum_{(s,a) \in \mathcal{S} \times \mathcal{A}} |u_{s,a} - v_{s,a}|^p\right] = \sum_{(s,a) \in \mathcal{S} \times \mathcal{A}} \mathbb{E}_{u,v \sim \Pi}\left[|u_{s,a} - v_{s,a}|^p\right] = 0$$
Therefore for all $(s, a) \in \mathcal{S} \times \mathcal{A}$:
$$\left|\mathbb{E}_{u \sim \mathbb{P}_\pi^\Phi}[u_{s,a}] - \mathbb{E}_{v \sim \mathbb{P}_{\pi'}^\Phi}[v_{s,a}]\right|^p \leq \mathbb{E}_{u,v \sim \Pi}\left[|u_{s,a} - v_{s,a}|^p\right] = 0$$
Where $u_{s,a}$ and $v_{s,a}$ denote the $(s, a)$ entries of $u$ and $v$ respectively. Notice that for all $(s, a) \in \mathcal{S} \times \mathcal{A}$:
$$\mathbb{P}_\pi^\Phi(s, a) = \mathbb{P}_{\pi'}^\Phi(s, a) \tag{8}$$
Since for all $s \in \mathcal{S}$ and $p \geq 1$:
$$\left|\sum_{a \in \mathcal{A}} u_{s,a} - v_{s,a}\right|^p \leq \sum_{a \in \mathcal{A}} |u_{s,a} - v_{s,a}|^p$$
Therefore for all $s \in \mathcal{S}$:
$$\left|\mathbb{E}_{u \sim \mathbb{P}_\pi^\Phi}\left[\sum_{a \in \mathcal{A}} u_{s,a}\right] - \mathbb{E}_{v \sim \mathbb{P}_{\pi'}^\Phi}\left[\sum_{a \in \mathcal{A}} v_{s,a}\right]\right|^p \leq \mathbb{E}_{u,v \sim \Pi}\left[\sum_{a \in \mathcal{A}} |u_{s,a} - v_{s,a}|^p\right] = 0$$
Consequently $\mathbb{P}_\pi^\Phi(s) = \mathbb{P}_{\pi'}^\Phi(s)$ for all $s \in \mathcal{S}$. By Bayes rule, this plus equation 8 yields:
$$\mathbb{P}_\pi^\Phi(a|s) = \mathbb{P}_{\pi'}^\Phi(a|s)$$
And therefore: $\pi = \pi'$. $\qquad\square$

These results can be extended in the following ways:

1. In the case of a continuous state space, it is possible to define embeddings using Kernel density estimators. Under the appropriate smoothness conditions on the visitation frequencies, picking an adequate bandwidth and using the appropriate norm to compare different embeddings it is possible to derive similar results to those in Lemma 10.1 for continuous state spaces.

2. For embeddings such as $\Phi_5$ in Section 3.1 or $\Phi(\tau) = \sum_{t=0}^H e_{s_t, a_t}$, when $\gamma = 0$, if $\mathrm{WD}_\gamma(\mathbb{P}_\pi^\Phi, \mathbb{P}_{\pi'}^\Phi) \leq \epsilon$ then $|V(\pi) - V(\pi')| \leq \epsilon R$ for $R = \max_{\tau \in \Gamma} \mathcal{R}(\tau)$ thus implying that a small Wasserstein distance between $\pi$ and $\pi'$'s PPEs implies a small difference in their value functions.

## 10.1 RANDOM FEATURES STOCHASTIC GRADIENTS

Let $\phi_\kappa$ and $\phi_\ell$ be two feature maps over $\mathcal{X}$ and $\mathcal{Y}$ and corresponding to kernels $\kappa$ and $\ell$ respectively. For this and the following sections we will make use of the following expression:

$$G(\mathbf{p}^\mu, \mathbf{p}^\nu) = \beta \int_{\mathcal{X}} (\mathbf{p}^\mu)^\top \phi_\kappa(\mathbf{x}) d\mu(\mathbf{x}, \theta) - \beta \int_{\mathcal{Y}} (\mathbf{p}^\nu)^\top \phi_\ell(\mathbf{y}) d\nu(\mathbf{y}) + \tag{9}$$
$$\gamma\beta \int_{\mathcal{X}\times\mathcal{Y}} \exp\left( \frac{(\mathbf{p}^\mu)^\top \phi_\kappa(\mathbf{x}) - (\mathbf{p}^\nu)^\top \phi_\ell(\mathbf{y}) - C(\mathbf{x}, \mathbf{y})}{\gamma} \right) d\mu(\mathbf{x}) d\nu(\mathbf{y})$$

We now show how to compute gradients with respect to the random feature maps:

**Lemma 10.2.** *The gradient* $\nabla_{\binom{\mathbf{p}^\mu}{\mathbf{p}^\nu}} G(\mathbf{p}^\mu, \mathbf{p}^\nu)$ *of the objective function from Equation 9 with respect to the parameters* $\binom{\mathbf{p}^\mu}{\mathbf{p}^\nu}$ *satisfies:*

$$\nabla_{\binom{\mathbf{p}^\mu}{\mathbf{p}^\nu}} G(\mathbf{p}^\mu, \mathbf{p}^\nu) = \beta \mathbb{E}_{(\mathbf{x}, \mathbf{y}) \sim \mu \otimes \nu} \left[ \left( 1 - \exp\left( \frac{(\mathbf{p}^\mu)^\top \phi_\kappa(\mathbf{x}) - (\mathbf{p}^\nu)^\top \phi_\ell - C(\mathbf{x}, \mathbf{y})}{\gamma} \right) \right) \binom{\phi_\kappa(\mathbf{x})}{-\phi_\ell(\mathbf{y})} \right]$$

*Proof.* A simple use of the chain rule, taking the gradients inside the expectation, and the fact that $\mathbf{p}^\mu$ and $\mathbf{p}^\nu$ are vectors yields the desired result. ☐

The main consequence of this formulation is the stochastic gradients we use in Algorithm 1.

## 10.2 BGPG, BGES AND THEIR THEORETICAL GUARANTEES

Here we provide some theoretical guarantees for our algorithms. Both proposed methods BGPG and BGES follow the alternating optimization algorithmic template. We start by noting that stripped to their bare bones components our algorithms satisfy two paradigms:

1. Min-Max optimization. When $\beta < 0$ the algorithms we propose for solving the optimization objective in Equation , turn into an Alternating Min-Max optimization procedure. Henceforth whenever we refer to the problem defined by setting $\beta < 0$ we will call it the Min-Max problem.
2. Max-Max optimization. When $\beta > 0$ the algorithms we propose for solving the optimization objective in Equation turn into an alternating Max-Max procedure. Henceforth whenever we refer to the problem defined by setting $\beta < 0$ we will call it the Max-Max problem.

Our theoretical results will therefore center around providing guarantees for optimizing an objective of the form:

$$F(\theta, \mathbf{p}^{\pi_\theta}, \mathbf{p}^{\pi'}) = \mathbb{E}_{\tau_1, \tau_2 \sim \mathbb{P}_{\pi_\theta} \otimes \mathbb{P}_{\pi'}} \left[ \mathcal{R}(\pi_1) + \langle \mathbf{p}^{\pi_\theta}, \phi_\kappa(\Phi(\tau_1)) \rangle - \beta \langle \mathbf{p}^{\pi'}, \phi_\kappa(\Phi(\tau_2)) \rangle + \right. \tag{10}$$
$$\left. \gamma\beta \exp\left( \frac{\langle \mathbf{p}^{\pi_\theta}, \phi_\kappa(\Phi(\tau_1)) \rangle - \langle \phi^{\pi'}, \phi_\kappa(\Phi(\tau_2)) \rangle - C(\Phi(\tau_1), \Phi(\tau_2))}{\gamma} \right) \right]$$

Where $\pi_\theta$ is a policy parametrized by $\theta$, $\pi'$ is a target policy, and $\mathbf{p}^{\pi_\theta}$ and $\mathbf{p}^{\pi'}$ are the feature vectors corresponding to the Behavioral Test functions from $\pi_\theta$ and $\pi'$.

## 10.3 MAX-MAX PROBLEM: THEORETICAL ANALYSIS

We will analyze our AO algorithm for the Max-Max optimization problem. We show that obtained solutions converge to the local maxima of the objective function. Consider the function $F(\theta, \mathbf{p}^{\lambda_{\pi_\theta}}, \mathbf{p}^{\lambda_{\pi'}})$ as in 10. We denote by $(\theta^*, \mathbf{p}_*^{\lambda_{\pi_\theta}}, \mathbf{p}_*^{\lambda_{\pi'}})$ some of its local maxima. Define $\tilde{F}(\theta) = F(\theta, \mathbf{p}_*^{\lambda_{\pi_\theta}}, \mathbf{p}_*^{\lambda_{\pi'}})$, i.e. $\tilde{F}$ is $F$ as a function of $\theta$ for locally optimal values of $\mathbf{p}^{\lambda_{\pi_\theta}}$ and $\mathbf{p}^{\lambda_{\pi'}}$.

We will assume that $\tilde{F}$ is locally $\zeta$-strongly concave and $\delta$-smooth for some fixed $\zeta, \delta > 0$ in the neighborhood $N(\theta^*, r)$ of its optimal value $\theta^*$. We will also assume that gradient of $\tilde{F}$ is $L_2$-Lipschitz with Lipschitz coefficient $\phi$ in that neighborhood. The following convergence theorem holds:

Our goal in this section is to prove the following Theorem:

**Theorem 10.3.** *For the entropy coefficient $\gamma$, denote: $\phi_\gamma = \frac{1}{2\gamma} e^{\frac{4}{\gamma}}$, and $u_\gamma = \frac{8}{3\gamma} e^{\frac{4}{\gamma}}$. Denote $\phi_* = \max(\phi, \phi_\gamma)$ and $\xi = \min(\frac{2\delta\zeta}{\delta+\zeta}, u_\gamma)$. Let $s(t) = (\theta(t), \mathbf{p}^{\lambda_{\pi_\theta}}(t), \mathbf{p}^{\lambda_{\pi'}}(t))$ be the solution from iteration $t$ and $s_*$ the local maximum considered above. Assume that optimization starts in $\theta_0 \in N(\theta^*, r)$. If $\phi_* < \frac{2\xi}{3}$, then the error at iteration $t+1$ of the presented AO algorithm for the Max-Max problem with decaying gradient step size $\alpha^t = \frac{3/2}{[2\xi - 3\phi_*](t+2) + \frac{3}{2}\phi_*}$ satisfies:*

$$\mathbb{E}[\|s(t+1) - s_*\|_2^2] \leq \mathbb{E}[\|s(0) - s_*\|_2^2](\frac{2}{t+3})^{\frac{3}{2}} + \sigma^2 \frac{9}{[2\xi - 3\phi_*]^2(t+3)}, \tag{11}$$

*where $\sigma = \sqrt{2(1 + e^{\frac{2}{\gamma}})^2 + \sup_{N(\theta^*), r} \nabla_\theta \tilde{F}(\theta)^2}$.*

We will need several auxiliary technical results. We will use the following notation: $W_\gamma(\theta, \beta_1, \beta_2) = F(\theta, \mathbf{p}^{\lambda_{\pi_\theta}}, \mathbf{p}^{\lambda_{\pi'}})$, where $F$ is the objective function from the main body of the paper parameterised by entropy coefficient $\gamma > 0$, $\beta_1 = \mathbf{p}^{\lambda_{\pi_\theta}}$ and $\beta_2 = \mathbf{p}^{\lambda_{\pi'}}$. We will apply this notation also in the next section regarding the Min-Max Problem. For completeness, the definitions of strong concavity, smoothness and Lipschitz condition from Theorem 10.3 are given in Section 10.3.1.

We consider the dual optimization problem:

$$\max_{\theta} \max_{\lambda_1 \in \mathcal{C}(\mathcal{X}), \lambda_2 \in \mathcal{C}(\mathcal{Y})} W_\gamma(\theta, \beta_1, \beta_2)$$
$$= \max_{\theta} \max_{\lambda_1 \in \mathcal{C}(\mathcal{X}), \lambda_2 \in \mathcal{C}(\mathcal{Y})} \mathbb{E}_{(x,y,\kappa^{x,1},\ldots,\kappa^{x,D},\kappa^{y,1},\ldots,\kappa^{y,D}) \sim \mu \times \nu \times \omega \times \cdots \times \omega}$$
$$\left[ \mathcal{L}(\theta) + \lambda_1(\Phi(x)) + \lambda_2(\Phi(y)) - \gamma \exp\left( \frac{\lambda_1(\Phi(x)) + \lambda_2(\Phi(y)) - C(\Phi(x), \Phi(y))}{\gamma} \right) \right],$$

where $\gamma$ is a parameter, $\mu = \pi_\theta$, $\nu = \pi'$, $\Phi$ is a fixed trajectories' embedding and furthermore:

$$\lambda_1(z) = \beta_1^\top f(z)$$
$$\lambda_2(z) = \beta_2^\top f(z),$$

such that $f(\cdot)$ is a random feature vector. The $i^{\text{th}}$ entry of the feature vector is constructed as follows: $[f(z)]_i = \sqrt{\frac{2}{D}} \cos(z^\top w^{z,i} + b^{z,i})$, where $w^{z,i} \sim \mathcal{N}\left(0, \mathbb{I}_n \frac{1}{\rho^2}\right)$ and $b^{z,i} \sim \text{Uniform}(0, 2\pi)$. For the ease of notation we denote $\kappa^{z,i} = (w^{z,i}, b^{z,i}) \sim \omega$. We consider stochastic gradient ascent optimization strategy of the following form:

- at time $t$ we receive a *single sample* $(x_t, y_t, \kappa_t^{x_t,1}, \ldots, \kappa_t^{x_t,D}, \kappa_t^{y_t,1}, \ldots, \kappa_t^{y_t,D}) \sim \mu \times \nu \times \omega \times \cdots \times \omega$, then we form feature vectors $[f(x_t)]_i = \sqrt{\frac{2}{D}} \cos(x_t^\top w_t^{x_t,i} + b_t^{x_t,i})$ and $[f(y_t)]_i = \sqrt{\frac{2}{D}} \cos(y_t^\top w_t^{y_t,i} + b_t^{y_t,i})$, and finally update:

$$\beta_1^{t+1} \leftarrow \Pi_1 \left[ \beta_1^t + \alpha_t f(x_t) \left( 1 - \exp\left( \frac{(\beta_1^t)^\top f(x_t) + (\beta_2^t)^\top f(y_t) - C(x,y)}{\gamma} \right) \right) \right]$$

$$\beta_2^{t+1} \leftarrow \Pi_2 \left[ \beta_2^t + \alpha_t f(y_t) \left( 1 - \exp\left( \frac{(\beta_1^{t+1})^\top f(x_t) + (\beta_2^t)^\top f(y_t) - C(x,y)}{\gamma} \right) \right) \right].$$

$\Pi_d$ ($d = 1, 2$) denotes the projection onto the Euclidean ball $B_2(\frac{r_d}{2}, \beta_d^0)$ of some given radius $\frac{r_d}{2}$ centered at the initial iterate $\beta_d^0$.

Let $\{\beta_1^*, \beta_2^*\}$ denote the global optimum of $W_\gamma(\beta_1, \beta_2)$ computed on the entire data population, i.e. given access to an infinite number of samples ("oracle"). Let $B_2(r, \beta)$ denote the Euclidean ball of radius $r$ centered at $\beta$. For the sake of the theoretical analysis we assume that (a lower-bound on) the radii of convergence $r_1, r_2$ for $\beta_1, \beta_2$, respectively, is known to the algorithm (this assumption

is potentially easy to eliminate with a more careful choice of the step size in the first iterations). To be more specific, if at any point in time parameter $\beta_1$ or $\beta_2$ falls outside the ball $B_2(r_1, \beta_1^*)$ or $B_2(r_2, \beta_2^*)$, respectively, the projection is applied that pushes the parameter of interest to stay in the ball. Also, let $\nabla_{\beta_1} W_\gamma^1(\beta_1, \beta_2)$ and $\nabla_{\beta_2} W_\gamma^1(\beta_1, \beta_2)$ denote the gradients of $W_\gamma$ with respect to $\beta_1$ and $\beta_2$, respectively, computed for a single sample. Similarly, $\nabla_{\beta_1} W_\gamma(\beta_1, \beta_2)$ and $\nabla_{\beta_2} W_\gamma(\beta_1, \beta_2)$ be the gradient of $W_\gamma$ with respect to $\beta_1$ and $\beta_2$, respectively, computed for the entire data population, i.e. infinite number of samples.

Note that given any initial vector $\beta_d^0$ in the ball of radius $\frac{r_d}{2}$ centered at $\beta_d^*$, we are guaranteed that all iterates remain within an $r_d$-ball of $\beta_d^*$. This is true for all $d = 1, 2$. The projection is necessary for theoretical analysis but in practice makes little difference. The above is a two-step alternated optimization scheme.

Let the *population gradient operator*, $\mathcal{G}_d(\beta_1, \beta_2)$, where $d = 1, 2$, be defined as

$$\mathcal{G}_d(\beta_1, \beta_2) := \beta_d + \alpha \nabla_{\beta_d} W_\gamma(\beta_1, \beta_2).$$

### 10.3.1 Assumptions

Let $W_{\gamma,1}^*(\beta_1) = W_\gamma(\beta_1, \beta_2^*)$ and $W_{\gamma,2}^*(\beta_2) = W_\gamma(\beta_1^*, \beta_2)$. Let $\Omega_1, \Omega_2$ denote non-empty compact convex sets such $\beta_1 \in \Omega_1$ and $\beta_2 \in \Omega_2$. The following assumptions are made:

**Assumption 10.1** (*Strong concavity*). *For $d = 1, 2$, the function $W_{\gamma,d}^*(\beta_d)$ is $\zeta_d$-strongly concave near $\beta_d^*$, i.e. for all pairs $(a_d, b_d)$ in the neighborhood of $\beta_d^*$ the following holds*

$$W_{\gamma,d}^*(a_d) - W_{\gamma,d}^*(b_d) - \left\langle \nabla_{\beta_d} W_{\gamma,d}^*(b_d), a_d - b_d \right\rangle \leq -\frac{\zeta_d}{2} \|a_d - b_d\|_2^2,$$

*where $\zeta_d > 0$ is the strong concavity modulus.*

**Assumption 10.2** (*Smoothness*). *For $d = 1, 2$, the function $W_{\gamma,d}^*(\beta_d)$ is $\delta_d$-smooth, i.e. for all pairs $(a_d, b_d)$ the following holds*

$$W_{\gamma,d}^*(a_d) - W_{\gamma,d}^*(b_d) - \left\langle \nabla_{\beta_d} W_{\gamma,d}^*(b_d), a_d - b_d \right\rangle \geq -\frac{\delta_d}{2} \|a_d - b_d\|_2^2,$$

*where $\delta_d > 0$ is the smoothness constant.*

**Assumption 10.3** (Gradient stability (GS) / Lipschitz condition). *We assume $W_\gamma(\beta_1, \beta_2)$ satisfy GS ($\phi_d$) condition, for all $d = 1, 2$, over Euclidean balls $\beta_1 \in B_2(r_1, \beta_1^*), \beta_2 \in B_2(r_2, \beta_2^*)$ given as follows*

$$\|\nabla_{\beta_d} W_{\gamma,d}^*(\beta_d) - \nabla_{\beta_d} W_\gamma(\beta_1, \beta_2)\|_2 \leq \phi_d \|\beta_{\bar{d}} - \beta_{\bar{d}}^*\|_2,$$

*where $\phi_d > 0$ and $\bar{d} = (d \mod 2) + 1$.*

Finally, define the bound $\sigma$ that considers the expected value of the norm of gradients of our objective function as follows: $\sigma = \sqrt{\sigma_1^2 + \sigma_2^2}$, where $\sigma_d^2 = \sup\{\mathbb{E}[\|\nabla_{\beta_d} W_\gamma^1(\beta_1, \beta_2)\|_2^2] : \beta_1 \in B_2(r_1, \beta_1^*), \beta_2 \in B_2(r_2, \beta_2^*)\}$ for $d = 1, 2$.

### 10.3.2 Main theorems

**Theorem 10.4.** *Given the stochastic gradient iterates of Max-Max method with decaying step size $\{\alpha^t\}_{t=0}^\infty$ and with $\phi < \frac{2\xi}{3}$ the error at iteration $t + 1$ satisfies the recursion*

$$\mathbb{E}\left[\|\beta_1^{t+1} - \beta_1^*\|_2^2 + \|\beta_2^{t+1} - \beta_2^*\|_2^2\right] \leq (1 - q^t)\mathbb{E}\left[\|\beta_1^t - \beta_1^*\|_2^2 + \|\beta_2^t - \beta_2^*\|_2^2\right] + \frac{(\alpha^t)^2}{1 - \alpha^t \phi}\sigma^2,$$

*where $\phi = \max_{d=1,2}(\phi_d)$, $q^t = 1 - \frac{1 - 2\alpha^t \xi + 2\alpha^t \phi}{1 - \alpha^t \phi}$, and $\xi = \min_{d=1,2}\left(\frac{2\delta_d \zeta_d}{\delta_d + \zeta_d}\right)$.*

The recursion in Theorem 10.4 is expanded yielding the convergence theorem:

**Theorem 10.5.** *Given the stochastic gradient iterates of Max-Max method with decaying step size $\alpha^t = \frac{3/2}{[2\xi - 3\phi](t+2) + \frac{3}{2}\phi}$ and assuming that $\phi < \frac{2\xi}{3}$, the error at iteration $t + 1$ satisfies*

$$\mathbb{E}\left[\|\beta_1^{t+1} - \beta_1^*\|_2^2 + \|\beta_2^{t+1} - \beta_2^*\|_2^2\right] \leq \mathbb{E}\left[\|\beta_1^0 - \beta_1^*\|_2^2 + \|\beta_2^0 - \beta_2^*\|_2^2\right]\left(\frac{2}{t+3}\right)^{\frac{3}{2}} + \sigma^2 \frac{9}{[2\xi - 3\phi]^2(t+3)},$$

*where $\phi = \max_{d=1,2}(\phi_d)$ and $\xi = \min_{d=1,2}\left(\frac{2\delta_d \zeta_d}{\delta_d + \zeta_d}\right)$.*

### 10.3.3 ANALYSIS

The theoretical analysis we provide below is an extension of the analysis in Balakrishnan et al. (2017) to the two-step alternated optimization scheme.

Proof of Theorem 10.5 relies on Theorem 10.4, which in turn relies on Theorem 10.7 and Lemma 10.6, both of which are stated below. Proofs of the lemma and theorems follow in the subsequent subsections.

The next result is a standard result from convex optimization (Theorem 2.1.14 in Nesterov (2014)) and is used in the proof of Theorem 10.7 below.

**Lemma 10.6.** *The gradient operator* $\mathcal{G}_1(\beta_1, \beta_2^*)$ *under Assumption 10.1 (strong concavity) and Assumption 10.2 (smoothness) with constant step size choice* $0 < \alpha \leq \frac{2}{\delta_1 + \zeta_1}$ *is contractive, i.e.*

$$\|\mathcal{G}_1(\beta_1, \beta_2^*) - \beta_1^*\|_2 \leq \left(1 - \frac{2\alpha\delta_1\zeta_1}{\delta_1 + \zeta_1}\right)\|\beta_1 - \beta_1^*\|_2 \tag{12}$$

*for all* $\beta_1 \in B_2(r_1, \beta_1^*)$.

*Similarly, the gradient operator* $\mathcal{G}_2(\beta_1^*, \beta_2)$ *under Assumption 10.1 (strong concavity) and Assumption 10.2 (smoothness) with constant step size choice* $0 < \alpha \leq \frac{2}{\delta_2 + \zeta_2}$ *is contractive, i.e.*

$$\|\mathcal{G}_2(\beta_1^*, \beta_2) - \beta_2^*\|_2 \leq \left(1 - \frac{2\alpha\delta_2\zeta_2}{\delta_2 + \zeta_2}\right)\|\beta_2 - \beta_2^*\|_2 \tag{13}$$

*for all* $\beta_2 \in B_2(r_2, \beta_2^*)$.

The next theorem also holds for $d = 1, 2$. Let $r_1, r_2 > 0$ and $\beta_1 \in B_2(r_1, \beta_1^*), \beta_2 \in B_2(r_2, \beta_2^*)$.

**Theorem 10.7.** *For some radius* $r_d > 0$ *(d = 1, 2) and a triplet* $(\phi_d, \zeta_d, \delta_d)$ *such that* $0 \leq \phi_d < \zeta_d \leq \delta_d$, *suppose that the function* $W_{\gamma,d}^*(\beta_d)$ *is* $\zeta_d$-*strongly concave and* $\delta_d$-*smooth, and that the GS* $(\phi_d)$ *condition holds. Then the population gradient operator* $\mathcal{G}_d(\beta_1, \beta_2)$ *with step* $\alpha$ *such that* $0 < \alpha \leq \min_{d=1,2} \frac{2}{\delta_d + \zeta_d}$ *is contractive over a ball* $B_2(r_d, \beta_d^*)$, *i.e.*

$$\|\mathcal{G}_d(\beta_1, \beta_2) - \beta_d^*\|_2 \leq (1 - \xi\alpha)\|\beta_d - \beta_d^*\|_2 + \alpha\phi\|\beta_{\bar{d}} - \beta_{\bar{d}}^*\|_2 \tag{14}$$

*where* $\bar{d} = (d \mod 2) + 1$, $\phi := \max_{d=1,2} \phi_d$, *and* $\xi := \min_{d=1,2} \frac{2\delta_d\zeta_d}{\delta_d + \zeta_d}$.

*Proof.*

$$\|\mathcal{G}_d(\beta_1, \beta_2) - \beta_d^*\|_2 = \|\beta_d + \alpha\nabla_{\beta_d}W_\gamma(\beta_1, \beta_2) - \beta_d^*\|_2$$

by the triangle inequality (and with $d \neq \bar{d}$) we further get

$$\leq \|\beta_d + \alpha\nabla_{\beta_d}W_{\gamma,d}^*(\beta_d) - \beta_d^*\|_2 + \alpha\|\nabla_{\beta_d}W_\gamma(\beta_1, \beta_2) - \nabla_{\beta_d}W_{\gamma,d}^*(\beta_d)\|_2$$

by the contractivity from Lemma 10.6 and GS condition

$$\leq \left(1 - \frac{2\alpha\delta_d\zeta_d}{\delta_d + \zeta_d}\right)\|\beta_d - \beta_d^*\|_2 + \alpha\phi_d\|\beta_{\bar{d}} - \beta_{\bar{d}}^*\|_2.$$

□

### Proof of Theorem 10.4

Let $\beta_d^{t+1} = \Pi_d(\tilde{\beta}_d^{t+1})$, where $\tilde{\beta}_1^{t+1} := \beta_1^t + \alpha^t\nabla_{\beta_1}W_\gamma^1(\beta_1^t, \beta_2^t)$ and $\tilde{\beta}_2^{t+1} := \beta_2^t + \alpha^t\nabla_{\beta_2}W_\gamma^1(\beta_1^{t+1}, \beta_2^t)$, where $\nabla_{\beta_d}W_\gamma^1$ is the gradient computed with respect to a single sample, $\tilde{\beta}_1$ and $\tilde{\beta}_2$ are the updates prior to the projection onto a ball $B_2(\frac{r_d}{2}, \beta_d^0)$. Let $\Delta_d^{t+1} := \beta_d^{t+1} - \beta_d^*$ and $\tilde{\Delta}_d^{t+1} := \tilde{\beta}_d^{t+1} - \beta_d^*$. Thus

$$
\begin{aligned}
\|\Delta_d^{t+1}\|_2^2 - \|\Delta_d^t\|_2^2 &\leq \|\tilde{\Delta}_d^{t+1}\|_2^2 - \|\Delta_d^t\|_2^2 \\
&= \|\tilde{\beta}_d^{t+1} - \beta_d^*\| - \|\beta_d^t - \beta_d^*\| \\
&= \left\langle \tilde{\beta}_d^{t+1} - \beta_d^t, \tilde{\beta}_d^{t+1} + \beta_d^t - 2\beta_d^* \right\rangle.
\end{aligned}
$$

Let $\hat{Q}_1^t := \nabla_{\beta_1} W_\gamma^1(\beta_1^t, \beta_2^t)$ and $\hat{Q}_2^t := \nabla_{\beta_2} W_\gamma^1(\beta_1^{t+1}, \beta_2^t)$. Then we have that $\tilde{\beta}_d^{t+1} - \beta_d^t = \alpha^t \hat{Q}_d^t$. We combine it with Equation 17 and obtain:

$$
\begin{aligned}
&\|\Delta_d^{t+1}\|_2^2 - \|\Delta_d^t\|_2^2 \\
\leq\ & \left\langle \alpha^t \hat{Q}_d^t, \alpha^t \hat{Q}_d^t + 2(\beta_d^t - \beta_d^*) \right\rangle \\
=\ & (\alpha^t)^2 (\hat{Q}_d^t)^\top \hat{Q}_d^t + 2\alpha^t (\hat{Q}_d^t)^\top (\beta_d^t - \beta_d^*) \\
=\ & (\alpha^t)^2 \|\hat{Q}_d^t\|_2^2 + 2\alpha^t \left\langle \hat{Q}_d^t, \Delta_d^t \right\rangle.
\end{aligned}
$$

Let $Q_1^t := \nabla_{\beta_1} W_\gamma(\beta_1^t, \beta_2^t)$ and $Q_2^t := \nabla_{\beta_2} W_\gamma(\beta_1^{t+1}, \beta_2^t)$. By the properties of martingales, i.e. iterated expectations and tower property:

$$
\mathbb{E}[\|\Delta_d^{t+1}\|_2^2] \leq \mathbb{E}[\|\Delta_d^t\|_2^2] + (\alpha^t)^2 \mathbb{E}[\|\hat{Q}_d^t\|_2^2] + 2\alpha^t \mathbb{E}[\langle Q_d^t, \Delta_d^t \rangle] \tag{15}
$$

Let $Q_d^* := \nabla_{\beta_d} W_\gamma(\beta_1^*, \beta_2^*)$. By self-consistency, i.e. $\beta_d^* = \arg\max_{\beta_d \in \Omega_d} W_{\gamma,d}^*(\beta_d)$, and convexity of $\Omega_d$ we have that

$$
\left\langle Q_d^*, \Delta_d^t \right\rangle = \left\langle \nabla_{\beta_d} W_\gamma(\beta_1^*, \beta_2^*), \Delta_d^t \right\rangle = 0.
$$

Combining this with Equation 15 we have

$$
\mathbb{E}[\|\Delta_d^{t+1}\|_2^2] \leq \mathbb{E}[\|\Delta_d^t\|_2^2] + (\alpha^t)^2 \mathbb{E}[\|\hat{Q}_d^t\|_2^2] + 2\alpha^t \mathbb{E}[\langle Q_d^t - Q_d^*, \Delta_d^t \rangle].
$$

Define $\mathcal{G}_d^t := \beta_d^t + \alpha^t Q_d^t$ and $\mathcal{G}_d^{t*} := \beta_d^* + \alpha^t Q_d^*$. Thus

$$
\begin{aligned}
& \alpha^t \left\langle Q_d^t - Q_d^*, \Delta_d^t \right\rangle \\
=\ & \left\langle \mathcal{G}_d^t - \mathcal{G}_d^{t*} - (\beta_d^t - \beta_d^*), \beta_d^t - \beta_d^* \right\rangle \\
=\ & \left\langle \mathcal{G}_d^t - \mathcal{G}_d^{t*}, \beta_d^t - \beta_d^* \right\rangle - \|\beta_d^t - \beta_d^*\|_2^2
\end{aligned}
$$

by the fact that $\mathcal{G}_d^{t*} = \beta_d^* + \alpha^t Q_d^* = \beta_d^*$ (since $Q_d^* = 0$):

$$
=\ \left\langle \mathcal{G}_d^t - \beta_d^*, \beta_d^t - \beta_d^* \right\rangle - \|\beta_d^t - \beta_d^*\|_2^2
$$

by the contractivity of $\mathcal{G}^t$ from Theorem 10.7:

$$
\begin{aligned}
\leq\ & \left\{ (1 - \alpha^t \xi)\|\beta_d^t - \beta_d^*\| + \alpha^t \phi \left( \sum_{i=1}^{d-1} \|\beta_i^{t+1} - \beta_i^*\|_2 + \sum_{i=d+1}^{2} \|\beta_i^t - \beta_i^*\|_2 \right) \right\} \|\beta_d^t - \beta_d^*\|_2 - \|\beta_d^t - \beta_d^*\|_2^2 \\
\leq\ & \left\{ (1 - \alpha^t \xi)\|\Delta_d^t\|_2 + \alpha^t \phi \left( \sum_{i=1}^{d-1} \|\Delta_i^{t+1}\|_2 + \sum_{i=d+1}^{2} \|\Delta_i^t\|_2 \right) \right\} \cdot \|\Delta_d^t\|_2 - \|\Delta_d^t\|_2^2
\end{aligned}
$$

Combining this result with Equation 16 gives

$$
\begin{aligned}
\mathbb{E}[\|\Delta_d^{t+1}\|_2^2] \leq\ & \mathbb{E}[\|\Delta_d^t\|_2^2] + (\alpha^t)^2 \mathbb{E}[\|\hat{Q}_d^t\|_2^2] + 2\mathbb{E}\left[ \left\{ (1 - \alpha^t \xi)\|\Delta_d^t\|_2 + \alpha^t \phi \left( \sum_{i=1}^{d-1} \|\Delta_i^{t+1}\|_2 + \sum_{i=d+1}^{2} \|\Delta_i^t\|_2 \right) \right\} \right. \\
& \left. \cdot \|\Delta_d^t\|_2 - \|\Delta_d^t\|_2^2 \right] \\
\leq\ & \mathbb{E}[\|\Delta_d^t\|_2^2] + (\alpha^t)^2 \sigma_d^2 + 2\mathbb{E}\left[ \left\{ (1 - \alpha^t \xi)\|\Delta_d^t\|_2 + \alpha^t \phi \left( \sum_{i=1}^{d-1} \|\Delta_i^{t+1}\|_2 + \sum_{i=d+1}^{2} \|\Delta_i^t\|_2 \right) \right\} \right. \\
& \left. \cdot \|\Delta_d^t\|_2 - \|\Delta_d^t\|_2^2 \right].
\end{aligned}
$$

After re-arranging the terms we obtain

$$\mathbb{E}[\|\Delta_d^{t+1}\|_2^2] \leq (\alpha^t)^2\sigma_d^2 + (1-2\alpha^t\xi)\mathbb{E}[\|\Delta_d^t\|_2^2] + 2\alpha^t\phi\mathbb{E}\left[\left(\sum_{i=1}^{d-1}\|\Delta_i^{t+1}\|_2 + \sum_{i=d+1}^{2}\|\Delta_i^t\|_2\right)\|\Delta_d^t\|_2\right]$$

apply $2ab \leq a^2 + b^2$ :

$$\leq (\alpha^t)^2\sigma_d^2 + (1-2\alpha^t\xi)\mathbb{E}[\|\Delta_d^t\|_2^2] + \alpha^t\phi\mathbb{E}\left[\sum_{i=1}^{d-1}\left(\|\Delta_i^{t+1}\|_2^2 + \|\Delta_d^t\|_2^2\right)\right] + \alpha^t\phi\mathbb{E}\left[\sum_{i=d+1}^{2}\left(\|\Delta_i^t\|_2^2 + \|\Delta_d^t\|_2^2\right)\right]$$

$$= (\alpha^t)^2\sigma_d^2 + \mathbb{E}[\|\Delta_d^t\|_2^2]\cdot\left[1-2\alpha^t\xi + \alpha^t\phi\right] + \alpha^t\phi\mathbb{E}\left[\sum_{i=1}^{d-1}\|\Delta_i^{t+1}\|_2^2\right] + \alpha^t\phi\mathbb{E}\left[\sum_{i=d+1}^{2}\|\Delta_i^t\|_2^2\right]$$

We obtained

$$\mathbb{E}[\|\Delta_d^{t+1}\|_2^2] \leq (\alpha^t)^2\sigma_d^2 + [1-2\alpha^t\xi + \alpha^t\phi]\mathbb{E}[\|\Delta_d^t\|_2^2] + \alpha^t\phi\mathbb{E}\left[\sum_{i=1}^{d-1}\|\Delta_i^{t+1}\|_2^2\right] + \alpha^t\phi\mathbb{E}\left[\sum_{i=d+1}^{2}\|\Delta_i^t\|_2^2\right]$$

we next re-group the terms as follows

$$\mathbb{E}[\|\Delta_d^{t+1}\|_2^2] - \alpha^t\phi\mathbb{E}\left[\sum_{i=1}^{d-1}\|\Delta_i^{t+1}\|_2^2\right] \leq [1-2\alpha^t\xi + \alpha^t\phi]\mathbb{E}[\|\Delta_d^t\|_2^2] + \alpha^t\phi\mathbb{E}\left[\sum_{i=d+1}^{2}\|\Delta_i^t\|_2^2\right] + (\alpha^t)^2\sigma_d^2$$

and then sum over $d$ from 1 to 2

$$\mathbb{E}\left[\sum_{d=1}^{2}\|\Delta_d^{t+1}\|_2^2\right] - \alpha^t\phi\mathbb{E}\left[\sum_{d=1}^{2}\sum_{i=1}^{d-1}\|\Delta_i^{t+1}\|_2^2\right]$$

$$\leq [1-2\alpha^t\xi + \alpha^t\phi]\mathbb{E}\left[\sum_{d=1}^{2}\|\Delta_d^t\|_2^2\right] + \alpha^t\phi\mathbb{E}\left[\sum_{d=1}^{2}\sum_{i=d+1}^{2}\|\Delta_i^t\|_2^2\right] + (\alpha^t)^2\sum_{d=1}^{2}\sigma_d^2$$

Let $\sigma = \sqrt{\sigma_1^2 + \sigma_2^2}$. Also, note that

$$\mathbb{E}\left[\sum_{d=1}^{2}\|\Delta_d^{t+1}\|_2^2\right] - \alpha^t\phi\mathbb{E}\left[\sum_{d=1}^{2}\|\Delta_d^{t+1}\|_2^2\right] \leq \mathbb{E}\left[\sum_{d=1}^{2}\|\Delta_d^{t+1}\|_2^2\right] - \alpha^t\phi\mathbb{E}\left[\sum_{d=1}^{2}\sum_{i=1}^{d-1}\|\Delta_i^{t+1}\|_2^2\right]$$

and

$$[1-2\alpha^t\xi + \alpha^t\phi]\mathbb{E}\left[\sum_{d=1}^{2}\|\Delta_d^t\|_2^2\right] + \alpha^t\phi\mathbb{E}\left[\sum_{d=1}^{2}\sum_{i=d+1}^{2}\|\Delta_i^t\|_2^2\right] + (\alpha^t)^2\sigma^2$$

$$\leq [1-2\alpha^t\xi + \alpha^t\phi]\mathbb{E}\left[\sum_{d=1}^{2}\|\Delta_d^t\|_2^2\right] + \alpha^t\phi\mathbb{E}\left[\sum_{d=1}^{2}\|\Delta_d^t\|_2^2\right] + (\alpha^t)^2\sigma^2$$

Combining these two facts with our previous results yields:

$$[1-\alpha^t\phi]\mathbb{E}\left[\sum_{d=1}^{2}\|\Delta_d^{t+1}\|_2^2\right]$$

$$\leq [1-2\alpha^t\xi + \alpha^t\phi]\mathbb{E}\left[\sum_{d=1}^{2}\|\Delta_d^t\|_2^2\right] + \alpha^t\phi\mathbb{E}\left[\sum_{d=1}^{2}\|\Delta_d^t\|_2^2\right] + (\alpha^t)^2\sigma^2$$

$$= [1-2\alpha^t\xi + 2\alpha^t\phi]\mathbb{E}\left[\sum_{d=1}^{2}\|\Delta_d^t\|_2^2\right] + (\alpha^t)^2\sigma^2$$

Thus:

$$\mathbb{E}\left[\sum_{d=1}^{2}\|\Delta_d^{t+1}\|_2^2\right] \leq \frac{1 - 2\alpha^t\xi + 2\alpha^t\phi}{1 - \alpha^t\phi}\mathbb{E}\left[\sum_{d=1}^{2}\|\Delta_d^t\|_2^2\right]$$
$$+ \frac{(\alpha^t)^2}{1 - \alpha^t\phi}\sigma^2.$$

Since $\phi < \frac{2\xi}{3}$, $\frac{1 - 2\alpha^t\xi + 2\alpha^t\phi}{1 - \alpha^t\phi} < 1$.

**Proof of Theorem 10.5**

To obtain the final theorem we need to expand the recursion from Theorem 10.4. We obtained

$$\mathbb{E}\left[\sum_{d=1}^{2}\|\Delta_d^{t+1}\|_2^2\right]$$
$$\leq \frac{1 - 2\alpha^t[\xi - \phi]}{1 - \alpha^t\phi}\mathbb{E}\left[\sum_{d=1}^{2}\|\Delta_d^t\|_2^2\right] + \frac{(\alpha^t)^2}{1 - \alpha^t\phi}\sigma^2$$
$$= \left(1 - \frac{\alpha^t[2\xi - 3\phi]}{1 - \alpha^t\phi}\right)\mathbb{E}\left[\sum_{d=1}^{2}\|\Delta_d^t\|_2^2\right] + \frac{(\alpha^t)^2}{1 - \alpha^t\phi}\sigma^2$$

Recall that we defined $q^t$ in Theorem 10.4 as

$$q^t = 1 - \frac{1 - 2\alpha^t\xi + 2\alpha^t\phi}{1 - \alpha^t\phi} = \frac{\alpha^t[2\xi - 3\phi]}{1 - \alpha^t\phi}$$

and denote

$$f^t = \frac{(\alpha^t)^2}{1 - \alpha^t\phi}.$$

Thus we have

$$\mathbb{E}\left[\sum_{d=1}^{2}\|\Delta_d^{t+1}\|_2^2\right] \leq (1 - q^t)\mathbb{E}\left[\sum_{d=1}^{2}\|\Delta_d^t\|_2^2\right] + f^t\sigma^2$$

$$\leq (1 - q^t)\left\{(1 - q^{t-1})\mathbb{E}\left[\sum_{d=1}^{2}\|\Delta_d^{t-1}\|_2^2\right] + f^{t-1}\sigma^2\right\} + f^t\sigma^2$$

$$= (1 - q^t)(1 - q^{t-1})\mathbb{E}\left[\sum_{d=1}^{2}\|\Delta_d^{t-1}\|_2^2\right] + (1 - q^t)f^{t-1}\sigma^2 + f^t\sigma^2$$

$$\leq (1 - q^t)(1 - q^{t-1})\left\{(1 - q^{t-2})\mathbb{E}\left[\sum_{d=1}^{2}\|\Delta_d^{t-2}\|_2^2\right] + f^{t-2}\sigma^2\right\} + (1 - q^t)f^{t-1}\sigma^2 + f^t\sigma^2$$

$$= (1 - q^t)(1 - q^{t-1})(1 - q^{t-2})\mathbb{E}\left[\sum_{d=1}^{2}\|\Delta_d^{t-2}\|_2^2\right]$$
$$+ (1 - q^t)(1 - q^{t-1})f^{t-2}\sigma^2 + (1 - q^t)f^{t-1}\sigma^2 + f^t\sigma^2$$

We end-up with the following

$$\mathbb{E}\left[\sum_{d=1}^{2}\|\Delta_d^{t+1}\|_2^2\right] \leq \mathbb{E}\left[\sum_{d=1}^{2}\|\Delta_d^0\|_2^2\right]\prod_{i=0}^{t}(1 - q^i) + \sigma^2\sum_{i=0}^{t-1}f^i\prod_{j=i+1}^{t}(1 - q^j) + f^t\sigma^2.$$

Set $q^t = \frac{\frac{3}{2}}{t+2}$ and

$$
\begin{aligned}
\alpha^t &= \frac{q^t}{2\xi - 3\phi + q^t\phi} \\
&= \frac{\frac{3}{2}}{[2\xi - 3\phi](t+2) + \frac{3}{2}\phi}.
\end{aligned}
$$

Denote $A = 2\xi - 3\phi$ and $B = \frac{3}{2}\phi$. Thus

$$
\alpha^t = \frac{\frac{3}{2}}{A(t+2) + B}
$$

and

$$
f^t = \frac{(\alpha^t)^2}{1 - \frac{2}{3}B\alpha^t} = \frac{\frac{9}{4}}{A(t+2)[A(t+2) + B]}.
$$

$$
\mathbb{E}\left[\sum_{d=1}^{2} \|\Delta_d^{t+1}\|_2^2\right]
$$

$$
\leq \mathbb{E}\left[\sum_{d=1}^{2} \|\Delta_d^0\|_2^2\right] \prod_{i=0}^{t}\left(1 - \frac{\frac{3}{2}}{i+2}\right) + \sigma^2 \sum_{i=0}^{t-1} \frac{\frac{9}{4}}{A(i+2)[A(i+2) + B]} \prod_{j=i+1}^{t}\left(1 - \frac{\frac{3}{2}}{j+2}\right)
$$

$$
+ \sigma^2 \frac{\frac{9}{4}}{A(t+2)[A(t+2) + B]}
$$

$$
= \mathbb{E}\left[\sum_{d=1}^{2} \|\Delta_d^0\|_2^2\right] \prod_{i=2}^{t+2}\left(1 - \frac{\frac{3}{2}}{i}\right) + \sigma^2 \sum_{i=2}^{t+1} \frac{\frac{9}{4}}{Ai[Ai + B]} \prod_{j=i+1}^{t+2}\left(1 - \frac{\frac{3}{2}}{j}\right) + \sigma^2 \frac{\frac{9}{4}}{A(t+2)[A(t+2) + B]}
$$

Since $A > 0$ and $B > 0$ thus

$$
\mathbb{E}\left[\sum_{d=1}^{2} \|\Delta_d^{t+1}\|_2^2\right]
$$

$$
\leq \mathbb{E}\left[\sum_{d=1}^{2} \|\Delta_d^0\|_2^2\right] \prod_{i=2}^{t+2}\left(1 - \frac{\frac{3}{2}}{i}\right) + \sigma^2 \sum_{i=2}^{t+1} \frac{\frac{9}{4}}{Ai[Ai + B]} \prod_{j=i+1}^{t+2}\left(1 - \frac{\frac{3}{2}}{j}\right) + \sigma^2 \frac{\frac{9}{4}}{A(t+2)[A(t+2) + B]}
$$

$$
\leq \mathbb{E}\left[\sum_{d=1}^{2} \|\Delta_d^0\|_2^2\right] \prod_{i=2}^{t+2}\left(1 - \frac{\frac{3}{2}}{i}\right) + \sigma^2 \sum_{i=2}^{t+1} \frac{\frac{9}{4}}{(Ai)^2} \prod_{j=i+1}^{t+2}\left(1 - \frac{\frac{3}{2}}{j}\right) + \sigma^2 \frac{\frac{9}{4}}{[A(t+2)]^2}
$$

We can next use the fact that for any $a \in (1, 2)$:

$$
\prod_{i=\tau+1}^{t+2}\left(1 - \frac{a}{i}\right) \leq \left(\frac{\tau+1}{t+3}\right)^a.
$$

The bound then becomes

$$
\mathbb{E}\left[\sum_{d=1}^{2} \|\Delta_d^{t+1}\|_2^2\right]
$$

$$
\leq \mathbb{E}\left[\sum_{d=1}^{2} \|\Delta_d^0\|_2^2\right] \prod_{i=2}^{t+2}\left(1 - \frac{\frac{3}{2}}{i}\right) + \sigma^2 \sum_{i=2}^{t+1} \frac{\frac{9}{4}}{(Ai)^2} \prod_{j=i+1}^{t+2}\left(1 - \frac{\frac{3}{2}}{j}\right) + \sigma^2 \frac{\frac{9}{4}}{[A(t+2)]^2}
$$

$$
\leq \mathbb{E}\left[\sum_{d=1}^{2} \|\Delta_d^0\|_2^2\right] \left(\frac{2}{t+3}\right)^{\frac{3}{2}} + \sigma^2 \sum_{i=2}^{t+1} \frac{\frac{9}{4}}{(Ai)^2} \left(\frac{i+1}{t+3}\right)^{\frac{3}{2}} + \sigma^2 \frac{\frac{9}{4}}{[A(t+2)]^2}
$$

$$
= \mathbb{E}\left[\sum_{d=1}^{2} \|\Delta_d^0\|_2^2\right] \left(\frac{2}{t+3}\right)^{\frac{3}{2}} + \sigma^2 \sum_{i=2}^{t+2} \frac{\frac{9}{4}}{(Ai)^2} \left(\frac{i+1}{t+3}\right)^{\frac{3}{2}}
$$

Note that $(i+1)^{\frac{3}{2}} \leq 2i$ for $i = 2, 3, \ldots$, thus

$$\mathbb{E}\left[\sum_{d=1}^{2} \|\Delta_d^{t+1}\|_2^2\right]$$

$$\leq \mathbb{E}\left[\sum_{d=1}^{2} \|\Delta_d^0\|_2^2\right]\left(\frac{2}{t+3}\right)^{\frac{3}{2}} + \sigma^2 \frac{\frac{9}{4}}{A^2(t+3)^{\frac{3}{2}}} \sum_{i=2}^{t+2} \frac{(i+1)^{\frac{3}{2}}}{i^2}$$

$$\leq \mathbb{E}\left[\sum_{d=1}^{2} \|\Delta_d^0\|_2^2\right]\left(\frac{2}{t+3}\right)^{\frac{3}{2}} + \sigma^2 \frac{\frac{9}{2}}{A^2(t+3)^{\frac{3}{2}}} \sum_{i=2}^{t+2} \frac{1}{i^{\frac{1}{2}}}$$

finally note that $\sum_{i=2}^{t+2} \frac{1}{i^{\frac{1}{2}}} \leq \int_1^{t+2} \frac{1}{x^{\frac{1}{2}}} dx \leq 2(t+3)^{\frac{1}{2}}$. Thus

$$\leq \mathbb{E}\left[\sum_{d=1}^{2} \|\Delta_d^0\|_2^2\right]\left(\frac{2}{t+3}\right)^{\frac{3}{2}} + \sigma^2 \frac{9}{A^2(t+3)}$$

substituting $A = 2\xi - 3\phi$ gives

$$= \mathbb{E}\left[\sum_{d=1}^{2} \|\Delta_d^0\|_2^2\right]\left(\frac{2}{t+3}\right)^{\frac{3}{2}} + \sigma^2 \frac{9}{[2\xi - 3\phi]^2(t+3)}$$

This leads us to the final theorem.

**Proof of Theorem 10.3**

In order to prove Theorem 10.3 it suffices to apply Theorem 10.5 and notice that:

- function $h_{\mathbf{v},C} : \mathbb{R}^d \to \mathbb{R}$ defined as follows: $h_{\mathbf{v},C}(\mathbf{w}) = \mathbf{w}^\top \mathbf{v} - A e^{\frac{\mathbf{w}^\top \mathbf{v}}{\lambda}}$ for $A = \gamma e^{-\frac{C}{\gamma}}$ is $\frac{2}{\gamma} e^{\frac{4}{\gamma}}$-smooth, $\frac{2}{\gamma}$-strongly concave and its gradient is Lipschitz with Lipschitz coefficient $\frac{1}{2\gamma} e^{\frac{4}{\gamma}}$ (with respect to $L_2$-norm) for $C > 0$ and $\mathbf{v}$ satisfying: $|\mathbf{w}^\top \mathbf{v}| \leq 1$,

- $\|\nabla h_{\mathbf{v},C}(\mathbf{v})\|_2^2 \leq 2(1 + e^{\frac{2}{\gamma}})^2$ under above conditions.

## 10.4 MinMax Problem: theoretical analysis

In this section we aim to obtain similar results for Min-Max Problem as for Max-Max problem. We will use the same notation as in the main body of the paper. We prove the following results:

**Theorem 10.8.** *Denote* $\phi_* = \max(\phi, \phi_\gamma)$ *and* $\xi = \min(\frac{2\delta\zeta}{\delta+\zeta}, u_\gamma)$, *where* $\phi_\gamma$ *and* $u_\gamma$ *are as in Theorem 10.3. Let* $s(t) = (\theta(t), \mathbf{p}^{\lambda_{\pi_\theta}}(t), \mathbf{p}^{\lambda_{\pi'}}(t))$ *be the solution obtained in iteration* $t$ *and* $s_*$ *the local optimum. Assume that optimization starts in* $\theta_0 \in N(\theta^*, r)$. *If* $\phi_* < \frac{\xi}{3}$, *then the error at iteration* $t+1$ *of the alternating optimization algorithm for the Min-Max problem with decaying gradient step size* $\alpha^t = \frac{3/2}{[2\xi - 6\phi_*](t+2) + 3\phi_*}$ *satisfies:*

$$\mathbb{E}[\|s(t+1) - s_*\|_2^2] \leq \mathbb{E}[\|s(0) - s_*\|_2^2]\left(\frac{2}{t+3}\right)^{\frac{3}{2}} + \sigma^2 \frac{9}{[2\xi - 6\phi_*]^2(t+3)}, \quad (16)$$

*where* $\sigma = \sqrt{2(1 + e^{\frac{2}{\gamma}})^2 + \sup_{N(\theta^*),r)} \nabla_\theta \tilde{F}(\theta)^2}$.

We use the same technical notation as in the previous section, with the only exception that this time we denote: $W_\gamma(\theta, \beta_1, \beta_2) = -F(\theta, \mathbf{p}^{\lambda_{\pi_\theta}}, \mathbf{p}^{\lambda_{\pi'}})$. We consider the MinMax problem of the following form

$$\min_\theta \max_{\lambda_1 \in \mathcal{C}(\mathcal{X}), \lambda_2 \in \mathcal{C}(\mathcal{Y})} W_\gamma(\theta, \beta_1, \beta_2)$$

$$= \min_\theta \max_{\lambda_1 \in \mathcal{C}(\mathcal{X}), \lambda_2 \in \mathcal{C}(\mathcal{Y})} \mathbb{E}_{(x, y, \kappa^{x,1}, \ldots, \kappa^{x,D}, \kappa^{y,1}, \ldots, \kappa^{y,D}) \sim \mu \times \nu \times \omega \times \cdots \times \omega}$$

$$\left[-\mathcal{L}(\theta) + \lambda_1(\Phi(x)) + \lambda_2(\Phi(y)) - \gamma \exp\left(\frac{\lambda_1(\Phi(x)) + \lambda_2(\Phi(y)) - C(\Phi(x), \Phi(y))}{\gamma}\right)\right],$$

where $\gamma$ is a parameter and the remaining notation is analogous to the Max-Max case.

We consider mixed stochastic gradient descent/ascend optimization strategy of the following form:

- at time $t$ we receive a *single sample* $(x_t, y_t, \kappa_t^{x_t,1}, \ldots, \kappa_t^{x_t,D}, \kappa_t^{y_t,1}, \ldots, \kappa_t^{y_t,D}) \sim \pi_{\theta^t} \times \nu \times \omega \times \cdots \times \omega$, then we form feature vectors $[f(x_t)]_i = \sqrt{\frac{2}{D}} \cos(x_t^\top w_t^{x_t,i} + b_t^{x_t,i})$ and $[f(y_t)]_i = \sqrt{\frac{2}{D}} \cos(y_t^\top w_t^{y_t,i} + b_t^{y_t,i})$, and finally update:

$$\theta^{t+1} \leftarrow \Pi_1 \left[ \theta^t - \alpha_t \nabla_{\theta=\theta^t} W_\gamma(\theta, \beta_1^t, \beta_2^t) \right]^4$$

$$\beta_1^{t+1} \leftarrow \Pi_2 \left[ \beta_1^t + \alpha_t f(x_t) \left( 1 - \exp \left( \frac{(\beta_1^t)^\top f(x_t) + (\beta_2^t)^\top f(y_t) - C(x,y)}{\gamma} \right) \right) \right]$$

$$\beta_2^{t+1} \leftarrow \Pi_3 \left[ \beta_2^t + \alpha_t f(y_t) \left( 1 - \exp \left( \frac{(\beta_1^{t+1})^\top f(x_t) + (\beta_2^t)^\top f(y_t) - C(x,y)}{\gamma} \right) \right) \right].$$

$\Pi_1$ denotes the projection onto the Euclidean ball $B_2(\frac{r_1}{2}, \theta^0)$ and $\Pi_d$ $(d = 1, 2)$ denotes the projection onto the Euclidean ball $B_2(\frac{r_1}{2}, \beta_d^0)$.

Let $\{\theta^*, \beta_1^*, \beta_2^*\}$ denote the the global optimal solution of the saddle point problem $\min_\theta \max_{\beta_1, \beta_2} W_\gamma(\theta, \beta_1, \beta_2)$ computed on the entire data population, i.e. given access to an infinite number of samples ("oracle"). As before, we assume that (a lower-bound on) the radii of convergence $r_1, r_2, r_3$ for $\theta, \beta_1, \beta_2$, respectively, is known to the algorithm and thus the projection is applied to control $\theta, \beta_1, \beta_2$ to stay in their respective balls. Also, let $\nabla_\theta W_\gamma^1(\theta, \beta_1, \beta_2), \nabla_{\beta_1} W_\gamma^1(\theta, \beta_1, \beta_2)$ and $\nabla_{\beta_2} W_\gamma^1(\theta, \beta_1, \beta_2)$ denote the gradients of $W_\gamma$ with respect to $\theta, \beta_1$ and $\beta_2$, respectively, computed for a single sample. Similarly, $\nabla_\theta W_\gamma(\theta, \beta_1, \beta_2), \nabla_{\beta_1} W_\gamma(\theta, \beta_1, \beta_2)$ and $\nabla_{\beta_2} W_\gamma(\theta, \beta_1, \beta_2)$ be the gradient of $W_\gamma$ with respect to $\theta, \beta_1$ and $\beta_2$, respectively, computed for the entire data population, i.e. infinite number of samples.

Note that given any initial vector $\theta^0$ in the ball of radius $\frac{r_1}{2}$ centered at $\theta^*$, we are guaranteed that all iterates remain within an $r_1$-ball of $\theta^*$ and given any initial vector $\beta_d^0$ $(d = 1, 2)$ in the ball of radius $\frac{r_d}{2}$ centered at $\beta_d^*$, we are guaranteed that all iterates remain within an $r_d$-ball of $\beta_d^*$. The projection is necessary for theoretical analysis but in practice makes little difference. The above is a three-step alternated optimization scheme.

Let the *population gradient operator*, $\mathcal{G}_d(\theta, \beta_1, \beta_2)$, where $d = 1, 2, 3$, be defined as

$$\mathcal{G}_1(\theta, \beta_1, \beta_2) := \theta - \alpha \nabla_\theta W_\gamma(\theta, \beta_1, \beta_2)$$

and

$$\mathcal{G}_d(\theta, \beta_1, \beta_2) := \beta_d + \alpha \nabla_{\beta_i} W_\gamma(\theta, \beta_1, \beta_2) \quad \text{for } d = 2, 3.$$

### 10.4.1 Assumptions

Let $W_{\gamma,1}^*(\theta) = W_\gamma(\theta, \beta_1^*, \beta_2^*)$, $W_{\gamma,2}^*(\beta_1) = W_\gamma(\theta^*, \beta_1, \beta_2^*)$ and $W_{\gamma,3}^*(\beta_2) = W_\gamma(\theta^*, \beta_1^*, \beta_2)$. Let $\Omega_1, \Omega_2, \Omega_3$ denote non-empty compact convex sets such $\theta \in \Omega_1, \beta_1 \in \Omega_2$ and $\beta_2 \in \Omega_3$. The following assumptions are made:

**Assumption 10.4** (*Strong convexity/concavity*). *The function $W_{\gamma,1}^*(\theta)$ is $\zeta_1$-strongly convex near $\theta^*$ and the functions $W_{\gamma,2}^*(\beta_1)$ and $W_{\gamma,3}^*(\beta_2)$ are $\zeta_2$- and $\zeta_3$-strongly concave, respectively, near $\beta_1^*$ and $\beta_2^*$, respectively, where $\zeta_1, \zeta_2, \zeta_3 > 0$.*

**Assumption 10.5** (*Smoothness*). *The functions $W_{\gamma,1}^*(\theta), W_{\gamma,2}^*(\beta_1)$, and $W_{\gamma,3}^*(\beta_2)$ are $\delta_1$-, $\delta_2$-, and $\delta_3$-smooth, respectively, where $\delta_1, \delta_2, \delta_3 > 0$ are the smoothness constants.*

**Assumption 10.6** (Gradient stability (GS) / Lipschitz condition). *We assume $W_\gamma(\theta, \beta_1, \beta_2)$ satisfy GS ($\phi_d$) condition, for all $d = 1, 2, 3$, over Euclidean balls $\theta \in B_2(r_1, \theta^*), \beta_1 \in B_2(r_2, \beta_1^*), \beta_2 \in B_2(r_3, \beta_2^*)$ given as follows*

$$\|\nabla_\theta W_{\gamma,1}^*(\theta) - \nabla_\theta W_\gamma(\theta, \beta_1, \beta_2)\|_2 \le \phi_1 \sum_{d=1}^2 \|\beta_d - \beta_d^*\|_2,$$

---

[4]later we also use alternative notation for gradient $\nabla_{\theta=\theta^t} W_\gamma(\theta, \beta_1^t, \beta_2^t)$ as $\nabla_\theta W_\gamma(\theta^t, \beta_1^t, \beta_2^t)$

*and for $d = 1, 2$*

$$\|\nabla_{\beta_d} W^*_{\gamma, d+1}(\beta_d) - \nabla_{\beta_d} W_\gamma(\theta, \beta_1, \beta_2)\|_2 \leq \phi_d(\|\theta - \theta^*\|_2 + \|\beta_{\bar{d}} - \beta^*_{\bar{d}}\|_2),$$

*where $\phi_d > 0$ and $\bar{d} = (d \mod 2) + 1$.*

Finally, as before, define the bound $\sigma$ that considers the expected value of the norm of gradients of our objective function as follows: $\sigma = \sqrt{\sigma_1^2 + \sigma_2^2 + \sigma_3^2}$, where $\sigma_1^2 = \sup\{\mathbb{E}[\|\nabla_\theta W^1_\gamma(\theta, \beta_1, \beta_2)\|_2^2] : \theta \in B_2(r_1, \theta^*), \beta_1 \in B_2(r_2, \beta_1^*), \beta_2 \in B_2(r_3, \beta_2^*)\}$ and for $d = 1, 2$ $\sigma_{d+1}^2 = \sup\{\mathbb{E}[\|\nabla_{\beta_d} W^1_\gamma(\theta, \beta_1, \beta_2)\|_2^2] : \theta \in B_2(r_1, \theta^*), \beta_1 \in B_2(r_2, \beta_1^*), \beta_2 \in B_2(r_3, \beta_2^*)\}$.

### 10.4.2 MAIN THEOREMS

**Theorem 10.9.** *Given the stochastic gradient iterates of MinMax method with decaying step size $\{\alpha^t\}_{t=0}^\infty$ and with $\phi < \frac{\xi}{3}$ the error at iteration $t + 1$ satisfies the recursion*

$$\mathbb{E}\left[\|\theta^{t+1} - \theta^*\|_2^2 + \|\beta_1^{t+1} - \beta_1^*\|_2^2 + \|\beta_2^{t+1} - \beta_2^*\|_2^2\right]$$

$$\leq (1 - q^t)\mathbb{E}\left[\|\theta^t - \theta^*\|_2^2 + \|\beta_1^t - \beta_1^*\|_2^2 + \|\beta_2^t - \beta_2^*\|_2^2\right] + \frac{(\alpha^t)^2}{1 - 2\alpha^t\phi}\sigma^2,$$

*where $\phi = \max_{d=1,2,3}(\phi_d)$, $q^t = 1 - \frac{1 - 2\alpha^t\xi + 4\alpha^t\phi}{1 - 2\alpha^t\phi}$, and $\xi = \min_{d=1,2,3}\left(\frac{2\delta_d\zeta_d}{\delta_d + \zeta_d}\right)$.*

The recursion in Theorem 10.4 is expanded yielding the convergence theorem:

**Theorem 10.10.** *Given the stochastic gradient iterates of MinMax method with decaying step size $\alpha^t = \frac{3/2}{[2\xi - 6\phi](t+2) + 3\phi}$ and assuming that $\phi < \frac{\xi}{3}$, the error at iteration $t + 1$ satisfies*

$$\mathbb{E}\left[\|\theta^{t+1} - \theta^*\|_2^2 + \|\beta_1^{t+1} - \beta_1^*\|_2^2 + \|\beta_2^{t+1} - \beta_2^*\|_2^2\right]$$

$$\leq \mathbb{E}\left[\|\theta^0 - \theta^*\|_2^2 + \|\beta_1^0 - \beta_1^*\|_2^2 + \|\beta_2^0 - \beta_2^*\|_2^2\right]\left(\frac{2}{t+3}\right)^{\frac{3}{2}} + \sigma^2\frac{9}{[2\xi - 6\phi]^2(t+3)},$$

*where $\phi = \max_{d=1,2,3}(\phi_d)$ and $\xi = \min_{d=1,2,3}\left(\frac{2\delta_d\zeta_d}{\delta_d + \zeta_d}\right)$.*

Proof of Theorem 10.10 relies on Theorem 10.9, which in turn relies on Theorem 10.12 and Lemma 10.11, both of which are stated below. Proofs of the lemma and theorems follow in the subsequent subsections.

### 10.4.3 ANALYSIS

The next result is a standard result from convex optimization (Theorem 2.1.14 in Nesterov (2014)) and is used in the proof of Theorem 10.12 below.

**Lemma 10.11.** *The gradient operator $\mathcal{G}_1(\theta, \beta_1^*, \beta_2^*)$ under strong convexity and smoothness assumptions with constant step size choice $0 < \alpha \leq \frac{2}{\delta_1 + \zeta_1}$ is contractive, i.e.*

$$\|\mathcal{G}_1(\theta, \beta_1^*, \beta_2^*) - \theta^*\|_2 \leq \left(1 - \frac{2\alpha\delta_1\zeta_1}{\delta_1 + \zeta_1}\right)\|\theta - \theta^*\|_2$$

*for all $\theta \in B_2(r_1, \theta^*)$.*

*Similarly, the gradient operator $\mathcal{G}_1(\theta^*, \beta_1, \beta_2^*)$ under strong concavity and smoothness assumptions with constant step size choice $0 < \alpha \leq \frac{2}{\delta_2 + \zeta_2}$ is contractive, i.e.*

$$\|\mathcal{G}_1(\theta^*, \beta_1, \beta_2^*) - \beta_1^*\|_2 \leq \left(1 - \frac{2\alpha\delta_2\zeta_2}{\delta_2 + \zeta_2}\right)\|\beta_1 - \beta_1^*\|_2$$

*for all $\beta_1 \in B_2(r_2, \beta_1^*)$.*

*And similarly, the gradient operator $\mathcal{G}_2(\theta^*, \beta_1^*, \beta_2)$ under strong concavity and smoothness assumptions with constant step size choice $0 < \alpha \leq \frac{2}{\delta_3 + \zeta_3}$ is contractive, i.e.*

$$\|\mathcal{G}_2(\theta^*, \beta_1^*, \beta_2) - \beta_2^*\|_2 \leq \left(1 - \frac{2\alpha\delta_3\zeta_3}{\delta_3 + \zeta_3}\right)\|\beta_2 - \beta_2^*\|_2$$

*for all $\beta_2 \in B_2(r_3, \beta_2^*)$.*

The next theorem holds for $d = 1, 2, 3$. Let $r_1, r_2, r_3 > 0$ and $\theta \in B_2(r_1, \theta^*)$, $\beta_1 \in B_2(r_2, \beta_1^*)$, $\beta_2 \in B_2(r_3, \beta_2^*)$.

**Theorem 10.12.** *For some radius $r_1$ and a triplet $(\phi_1, \zeta_1, \delta_1)$ such that $0 \le \phi_1 < \zeta_1 \le \delta_1$, suppose that the function $W_{\gamma,1}^*(\theta)$ is $\zeta_1$-strongly convex and $\delta_1$-smooth and that the GS $(\phi_1)$ condition holds. Then the population gradient operator $\mathcal{G}_1(\theta, \beta_1, \beta_2)$ with step $\alpha$ such that $0 < \alpha \le \min_{d=1,2,3} \frac{2}{\delta_d + \zeta_d}$ is contractive over a ball $B_2(r_1, \theta^*)$, i.e.*

$$\|\mathcal{G}_1(\theta, \beta_1, \beta_2) - \theta^*\|_2 \le (1 - \xi\alpha)\|\theta - \theta^*\|_2 + \alpha\phi \sum_{d=1}^{2} \|\beta_d - \beta_d^*\|_2$$

*where $\phi := \max_{d=1,2,3} \phi_d$ and $\xi := \min_{d=1,2,3} \frac{2\delta_d\zeta_d}{\delta_d + \zeta_d}$.*

*For some radius $r_d$ ($d = 2, 3$) and a triplet $(\phi_d, \zeta_d, \delta_d)$ such that $0 \le \phi_d < \zeta_d \le \delta_d$, suppose that the function $W_{\gamma,d}^*(\beta_{d-1})$ is $\zeta_d$-strongly concave and $\delta_d$-smooth and that the GS $(\phi_d)$ condition holds. Then the population gradient operator $\mathcal{G}_d(\theta, \beta_1, \beta_2)$ with step $\alpha$ such that $0 < \alpha \le \min_{d=1,2,3} \frac{2}{\delta_d + \zeta_d}$ is contractive over a ball $B_2(r_d, \beta_d^*)$, i.e.*

$$\|\mathcal{G}_d(\theta, \beta_1, \beta_2) - \beta_d^*\|_2 \le (1 - \xi\alpha)\|\beta_d - \beta_d^*\|_2 + \alpha\phi(\|\beta_{\bar{d}} - \beta_{\bar{d}}^*\|_2 + \|\theta - \theta^*\|_2).$$

*where $\bar{d} = ((d-1) \mod 2) + 1$, $\phi := \max_{d=1,2,3} \phi_d$, and $\xi := \min_{d=1,2,3} \frac{2\delta_d\zeta_d}{\delta_d + \zeta_d}$.*

*Proof.*

$$\|\mathcal{G}_1(\theta, \beta_1, \beta_2) - \theta^*\|_2 = \|\theta - \alpha\nabla_\theta W_\gamma(\theta, \beta_1, \beta_2) - \theta^*\|_2$$

by the triangle inequality we further get

$$\le \|\theta - \alpha\nabla_\theta W_{\gamma,1}^* - \theta^*\|_2 + \alpha\|\nabla_\theta W_\gamma(\theta, \beta_1, \beta_2) - \nabla_\theta W_{\gamma,1}^*\|_2$$

by the contractivity from Lemma 10.11 and GS condition

$$\le \left(1 - \frac{2\alpha\delta_1\zeta_1}{\delta_1 + \zeta_1}\right)\|\theta - \theta^*\|_2 + \alpha\phi_1 \sum_{d=1}^{2} \|\beta_d - \beta_d^*\|_2.$$

The proof of the rest of the theorem is analogous to the proof of Theorem 10.7.

$\square$

**Proof of Theorem 10.9**

Let $\theta_1 = \theta$, $\theta_2 = \beta_1$, and $\theta_3 = \beta_2$.

Let $\theta_d^{t+1} = \Pi_d(\tilde{\theta}_d^{t+1})$, where $\tilde{\theta}_1^{t+1} := \theta_1^t - \alpha^t \nabla_{\theta_1} W_\gamma^1(\theta_1^t, \theta_2^t, \theta_3^t)$, $\tilde{\theta}_2^{t+1} := \theta_2^t + \alpha^t \nabla_{\theta_2} W_\gamma^1(\theta_1^{t+1}, \theta_2^t, \theta_3^t)$, and $\tilde{\theta}_3^{t+1} := \theta_3^t + \alpha^t \nabla_{\theta_3} W_\gamma^1(\theta_1^{t+1}, \theta_2^{t+1}, \theta_3^t)$, where $\nabla_{\theta_d} W_\gamma^1$ is the gradient computed with respect to a single sample, $\tilde{\theta}_1, \tilde{\theta}_2$, and $\tilde{\theta}_3$ are the updates prior to the projection. Let $\Delta_1^{t+1} := -\theta_1^{t+1} + \theta_1^*$ and for $d = 2, 3$, $\tilde{\Delta}_d^{t+1} := \tilde{\theta}_d^{t+1} - \theta_d^*$. Thus

$$\begin{aligned}
\|\Delta_d^{t+1}\|_2^2 - \|\Delta_d^t\|_2^2 &\le \|\tilde{\Delta}_d^{t+1}\|_2^2 - \|\Delta_d^t\|_2^2 \\
&= \|\tilde{\theta}_d^{t+1} - \theta_d^*\| - \|\theta_d^t - \theta_d^*\| \\
&= \left\langle \tilde{\theta}_d^{t+1} - \theta_d^t, \tilde{\theta}_d^{t+1} + \theta_d^t - 2\theta_d^* \right\rangle.
\end{aligned}$$

Let $\hat{Q}_1^t := \nabla_{\theta_1} W_\gamma^1(\theta_1^t, \theta_2^t, \theta_3^t)$, $\hat{Q}_2^t := \nabla_{\theta_2} W_\gamma^1(\theta_1^{t+1}, \theta_2^t, \theta_3^t)$, and $\hat{Q}_3^t := \nabla_{\theta_3} W_\gamma^1(\theta_1^{t+1}, \theta_2^{t+1}, \theta_3^t)$. Thus:

$$\begin{aligned}
\|\Delta_d^{t+1}\|_2^2 - \|\Delta_d^t\|_2^2 & \\
&\le \left\langle \alpha^t \hat{Q}_d^t, \alpha^t \hat{Q}_d^t + 2(\theta_d^t - \theta_d^*) \right\rangle \\
&= (\alpha^t)^2 (\hat{Q}_d^t)^\top \hat{Q}_d^t + 2\alpha^t (\hat{Q}_d^t)^\top (\theta_d^t - \theta_d^*) \\
&= (\alpha^t)^2 \|\hat{Q}_d^t\|_2^2 + 2\alpha^t \left\langle \hat{Q}_d^t, \Delta_d^t \right\rangle.
\end{aligned}$$

Let $Q_1^t := \nabla_{\theta_1} W_\gamma(\theta_1^t, \theta_2^t, \theta_3^t)$, $Q_2^t := \nabla_{\theta_2} W_\gamma(\theta_1^{t+1}, \theta_2^t, \theta_3^t)$, and $Q_3^t := \nabla_{\theta_3} W_\gamma(\theta_1^{t+1}, \theta_2^{t+1}, \theta_3^t)$. By the properties of martingales, i.e. iterated expectations and tower property:

$$\mathbb{E}[\|\Delta_d^{t+1}\|_2^2] \quad \leq \quad \mathbb{E}[\|\Delta_d^t\|_2^2] + (\alpha^t)^2 \mathbb{E}[\|\hat{Q}_d^t\|_2^2] + 2\alpha^t \mathbb{E}[\langle Q_d^t, \Delta_d^t \rangle]$$

Let $Q_d^* := \nabla_{\theta_d} W_\gamma(\theta_1^*, \theta_2^*, \theta_3^*)$. By self-consistency, i.e. $\theta_d^* = \arg\max_{\theta_d \in \Omega_d} W_{\gamma,d}^*(\theta_d)$, and convexity of $\Omega_d$ we have that

$$\langle Q_d^*, \Delta_d^t \rangle = \langle \nabla_{\theta_d} W_\gamma(\theta_1^*, \theta_2^*, \theta_3^*), \Delta_d^t \rangle = 0.$$

Combining this with the above inequality yields

$$\mathbb{E}[\|\Delta_d^{t+1}\|_2^2] \quad \leq \quad \mathbb{E}[\|\Delta_d^t\|_2^2] + (\alpha^t)^2 \mathbb{E}[\|\hat{Q}_d^t\|_2^2] + 2\alpha^t \mathbb{E}[\langle Q_d^t - Q_d^*, \Delta_d^t \rangle].$$

Define $\mathcal{G}_1^t := \theta_1^t - \alpha^t Q_1^t$ and $\mathcal{G}_1^{t*} := \theta_1^* - \alpha^t Q_1^*$. Also, for $d = 2, 3$ define $\mathcal{G}_d^t := \theta_d^t + \alpha^t Q_d^t$ and $\mathcal{G}_d^{t*} := \theta_d^* + \alpha^t Q_d^*$. Thus

$$\alpha^t \langle Q_d^t - Q_d^*, \Delta_d^t \rangle$$
$$= \quad \langle \mathcal{G}_d^t - \mathcal{G}_d^{t*} - (\theta_d^t - \theta_d^*), \theta_d^t - \theta_d^* \rangle$$
$$= \quad \langle \mathcal{G}_d^t - \mathcal{G}_d^{t*}, \theta_d^t - \theta_d^* \rangle - \|\theta_d^t - \theta_d^*\|_2^2$$

by the fact that $\mathcal{G}_d^{t*} = \theta_d^* + \alpha^t Q_d^* = \theta_d^*$ (since $Q_d^* = 0$):
$$= \quad \langle \mathcal{G}_d^t - \theta_d^*, \theta_d^t - \theta_d^* \rangle - \|\theta_d^t - \theta_d^*\|_2^2$$

by the contractivity of $\mathcal{G}^t$ from Theorem 10.7:

$$\leq \quad \left\{ (1 - \alpha^t \xi) \|\theta_d^t - \theta_d^*\| + \alpha^t \phi \left( \sum_{i=1}^{d-1} \|\theta_i^{t+1} - \theta_i^*\|_2 + \sum_{i=d+1}^{3} \|\theta_i^t - \theta_i^*\|_2 \right) \right\} \|\theta_d^t - \theta_d^*\|_2 - \|\theta_d^t - \theta_d^*\|_2^2$$

$$\leq \quad \left\{ (1 - \alpha^t \xi) \|\Delta_d^t\|_2 + \alpha^t \phi \left( \sum_{i=1}^{d-1} \|\Delta_i^{t+1}\|_2 + \sum_{i=d+1}^{3} \|\Delta_i^t\|_2 \right) \right\} \cdot \|\Delta_d^t\|_2 - \|\Delta_d^t\|_2^2$$

Thus

$$\mathbb{E}[\|\Delta_d^{t+1}\|_2^2] \quad \leq \quad \mathbb{E}[\|\Delta_d^t\|_2^2] + (\alpha^t)^2 \mathbb{E}[\|\hat{Q}_d^t\|_2^2] + 2\alpha^t \mathbb{E}[\langle Q_d^t - Q_d^*, \Delta_d^t \rangle]$$

$$= \quad \mathbb{E}[\|\Delta_d^t\|_2^2] + (\alpha^t)^2 \mathbb{E}[\|\hat{Q}_d^t\|_2^2] + 2\mathbb{E}\left[ \left\{ (1 - \alpha^t \xi) \|\Delta_d^t\|_2 + \alpha^t \phi \left( \sum_{i=1}^{d-1} \|\Delta_i^{t+1}\|_2 + \sum_{i=d+1}^{3} \|\Delta_i^t\|_2 \right) \right\} \right.$$
$$\left. \cdot \|\Delta_d^t\|_2 - \|\Delta_d^t\|_2^2 \right]$$

$$\leq \quad \mathbb{E}[\|\Delta_d^t\|_2^2] + (\alpha^t)^2 \sigma_d^2 + 2\mathbb{E}\left[ \left\{ (1 - \alpha^t \xi) \|\Delta_d^t\|_2 + \alpha^t \phi \left( \sum_{i=1}^{d-1} \|\Delta_i^{t+1}\|_2 + \sum_{i=d+1}^{3} \|\Delta_i^t\|_2 \right) \right\} \right.$$
$$\left. \cdot \|\Delta_d^t\|_2 - \|\Delta_d^t\|_2^2 \right].$$

After re-arranging the terms we obtain

$$\mathbb{E}[\|\Delta_d^{t+1}\|_2^2] \leq (\alpha^t)^2 \sigma_d^2 + (1 - 2\alpha^t \xi) \mathbb{E}[\|\Delta_d^t\|_2^2] + 2\alpha^t \phi \mathbb{E}\left[ \left( \sum_{i=1}^{d-1} \|\Delta_i^{t+1}\|_2 + \sum_{i=d+1}^{3} \|\Delta_i^t\|_2 \right) \|\Delta_d^t\|_2 \right]$$

apply $2ab \leq a^2 + b^2$ :

$$\leq \quad (\alpha^t)^2 \sigma_d^2 + (1 - 2\alpha^t \xi) \mathbb{E}[\|\Delta_d^t\|_2^2] + \alpha^t \phi \mathbb{E}\left[ \sum_{i=1}^{d-1} \left( \|\Delta_i^{t+1}\|_2^2 + \|\Delta_d^t\|_2^2 \right) \right] + \alpha^t \phi \mathbb{E}\left[ \sum_{i=d+1}^{3} \left( \|\Delta_i^t\|_2^2 + \|\Delta_d^t\|_2^2 \right) \right]$$

$$= \quad (\alpha^t)^2 \sigma_d^2 + \mathbb{E}[\|\Delta_d^t\|_2^2] \cdot \left[ 1 - 2\alpha^t \xi + 2\alpha^t \phi \right] + \alpha^t \phi \mathbb{E}\left[ \sum_{i=1}^{d-1} \|\Delta_i^{t+1}\|_2^2 \right] + \alpha^t \phi \mathbb{E}\left[ \sum_{i=d+1}^{3} \|\Delta_i^t\|_2^2 \right]$$

We obtained

$$\mathbb{E}[\|\Delta_d^{t+1}\|_2^2] \le (\alpha^t)^2 \sigma_d^2 + [1 - 2\alpha^t \xi + 2\alpha^t \phi]\mathbb{E}[\|\Delta_d^t\|_2^2] + \alpha^t \phi \mathbb{E}\left[\sum_{i=1}^{d-1} \|\Delta_i^{t+1}\|_2^2\right] + \alpha^t \phi \mathbb{E}\left[\sum_{i=d+1}^{3} \|\Delta_i^t\|_2^2\right]$$

we next re-group the terms as follows

$$\mathbb{E}[\|\Delta_d^{t+1}\|_2^2] - \alpha^t \phi \mathbb{E}\left[\sum_{i=1}^{d-1} \|\Delta_i^{t+1}\|_2^2\right] \le [1 - 2\alpha^t \xi + 2\alpha^t \phi]\mathbb{E}[\|\Delta_d^t\|_2^2] + \alpha^t \phi \mathbb{E}\left[\sum_{i=d+1}^{3} \|\Delta_i^t\|_2^2\right] + (\alpha^t)^2 \sigma_d^2$$

and then sum over $d$ from 1 to 3

$$\mathbb{E}\left[\sum_{d=1}^{3} \|\Delta_d^{t+1}\|_2^2\right] - \alpha^t \phi \mathbb{E}\left[\sum_{d=1}^{3}\sum_{i=1}^{d-1} \|\Delta_i^{t+1}\|_2^2\right]$$

$$\le [1 - 2\alpha^t \xi + 2\alpha^t \phi]\mathbb{E}\left[\sum_{d=1}^{3} \|\Delta_d^t\|_2^2\right] + \alpha^t \phi \mathbb{E}\left[\sum_{d=1}^{3}\sum_{i=d+1}^{3} \|\Delta_i^t\|_2^2\right] + 2(\alpha^t)^2 \sum_{d=1}^{3} \sigma_d^2$$

Note that

$$\mathbb{E}\left[\sum_{d=1}^{3} \|\Delta_d^{t+1}\|_2^2\right] - 2\alpha^t \phi \mathbb{E}\left[\sum_{d=1}^{3} \|\Delta_d^{t+1}\|_2^2\right] \le \mathbb{E}\left[\sum_{d=1}^{3} \|\Delta_d^{t+1}\|_2^2\right] - \alpha^t \phi \mathbb{E}\left[\sum_{d=1}^{3}\sum_{i=1}^{d-1} \|\Delta_i^{t+1}\|_2^2\right]$$

and

$$[1 - 2\alpha^t \xi + 2\alpha^t \phi]\mathbb{E}\left[\sum_{d=1}^{3} \|\Delta_d^t\|_2^2\right] + \alpha^t \phi \mathbb{E}\left[\sum_{d=1}^{3}\sum_{i=d+1}^{3} \|\Delta_i^t\|_2^2\right] + (\alpha^t)^2 \sigma^2$$

$$\le [1 - 2\alpha^t \xi + 2\alpha^t \phi]\mathbb{E}\left[\sum_{d=1}^{3} \|\Delta_d^t\|_2^2\right] + 2\alpha^t \phi \mathbb{E}\left[\sum_{d=1}^{3} \|\Delta_d^t\|_2^2\right] + (\alpha^t)^2 \sigma^2$$

Combining these two facts with our previous results yields:

$$[1 - 2\alpha^t \phi]\mathbb{E}\left[\sum_{d=1}^{3} \|\Delta_d^{t+1}\|_2^2\right]$$

$$\le [1 - 2\alpha^t \xi + 2\alpha^t \phi]\mathbb{E}\left[\sum_{d=1}^{3} \|\Delta_d^t\|_2^2\right] + 2\alpha^t \phi \mathbb{E}\left[\sum_{d=1}^{3} \|\Delta_d^t\|_2^2\right] + (\alpha^t)^2 \sigma^2$$

$$= [1 - 2\alpha^t \xi + 2\alpha^t \phi]\mathbb{E}\left[\sum_{d=1}^{3} \|\Delta_d^t\|_2^2\right] + (\alpha^t)^2 \sigma^2$$

Thus:

$$\mathbb{E}\left[\sum_{d=1}^{3} \|\Delta_d^{t+1}\|_2^2\right] \le \frac{1 - 2\alpha^t \xi + 4\alpha^t \phi}{1 - 2\alpha^t \phi} \mathbb{E}\left[\sum_{d=1}^{3} \|\Delta_d^t\|_2^2\right]$$

$$+ \frac{(\alpha^t)^2}{1 - 2\alpha^t \phi}\sigma^2.$$

Since $\phi < \frac{\xi}{3}$, $\frac{1 - 2\alpha^t \xi + 4\alpha^t \phi}{1 - 2\alpha^t \phi} < 1$.

**Proof of Theorem 10.10**

To obtain the final theorem we need to expand the recursion from Theorem 10.4. We obtained

$$\mathbb{E}\left[\sum_{d=1}^{3}\|\Delta_d^{t+1}\|_2^2\right]$$

$$\leq \frac{1-2\alpha^t[\xi-2\phi]}{1-2\alpha^t\phi}\mathbb{E}\left[\sum_{d=1}^{3}\|\Delta_d^t\|_2^2\right] + \frac{(\alpha^t)^2}{1-2\alpha^t\phi}\sigma^2$$

$$= \left(1-\frac{\alpha^t[2\xi-6\phi]}{1-2\alpha^t\phi}\right)\mathbb{E}\left[\sum_{d=1}^{3}\|\Delta_d^t\|_2^2\right] + \frac{(\alpha^t)^2}{1-2\alpha^t\phi}\sigma^2$$

Recall that we defined $q^t$ in Theorem 10.4 as

$$q^t = 1 - \frac{1-2\alpha^t\xi+4\alpha^t\phi}{1-2\alpha^t\phi} = \frac{\alpha^t[2\xi-6\phi]}{1-2\alpha^t\phi}$$

and denote

$$f^t = \frac{(\alpha^t)^2}{1-2\alpha^t\phi}.$$

Thus we have

$$\mathbb{E}\left[\sum_{d=1}^{3}\|\Delta_d^{t+1}\|_2^2\right] \leq (1-q^t)\mathbb{E}\left[\sum_{d=1}^{3}\|\Delta_d^t\|_2^2\right] + f^t\sigma^2$$

$$\leq (1-q^t)\left\{(1-q^{t-1})\mathbb{E}\left[\sum_{d=1}^{3}\|\Delta_d^{t-1}\|_2^2\right] + f^{t-1}\sigma^2\right\} + f^t\sigma^2$$

$$= (1-q^t)(1-q^{t-1})\mathbb{E}\left[\sum_{d=1}^{3}\|\Delta_d^{t-1}\|_2^2\right] + (1-q^t)f^{t-1}\sigma^2 + f^t\sigma^2$$

$$\leq (1-q^t)(1-q^{t-1})\left\{(1-q^{t-2})\mathbb{E}\left[\sum_{d=1}^{3}\|\Delta_d^{t-2}\|_2^2\right] + f^{t-2}\sigma^2\right\} + (1-q^t)f^{t-1}\sigma^2 + f^t\sigma^2$$

$$= (1-q^t)(1-q^{t-1})(1-q^{t-2})\mathbb{E}\left[\sum_{d=1}^{3}\|\Delta_d^{t-2}\|_2^2\right]$$
$$+ (1-q^t)(1-q^{t-1})f^{t-2}\sigma^2 + (1-q^t)f^{t-1}\sigma^2 + f^t\sigma^2$$

We end-up with the following

$$\mathbb{E}\left[\sum_{d=1}^{3}\|\Delta_d^{t+1}\|_2^2\right] \leq \mathbb{E}\left[\sum_{d=1}^{3}\|\Delta_d^0\|_2^2\right]\prod_{i=0}^{t}(1-q^i) + \sigma^2\sum_{i=0}^{t-1}f^i\prod_{j=i+1}^{t}(1-q^j) + f^t\sigma^2.$$

Set $q^t = \frac{\frac{3}{2}}{t+2}$ and

$$\begin{aligned}\alpha^t &= \frac{q^t}{2\xi-6\phi+2q^t\phi}\\ &= \frac{\frac{3}{2}}{[2\xi-6\phi](t+2)+3\phi}.\end{aligned}$$

Denote $A = 2\xi - 6\phi$ and $B = 3\phi$. Thus

$$\alpha^t = \frac{\frac{3}{2}}{A(t+2)+B}$$

and

$$f^t = \frac{(\alpha^t)^2}{1-\frac{2}{3}B\alpha^t} = \frac{\frac{9}{4}}{A(t+2)[A(t+2)+B]}.$$

$$\mathbb{E}\left[\sum_{d=1}^{2}\|\Delta_d^{t+1}\|_2^2\right]$$

$$\leq \mathbb{E}\left[\sum_{d=1}^{2}\|\Delta_d^0\|_2^2\right]\prod_{i=0}^{t}\left(1-\frac{\frac{3}{2}}{i+2}\right)+\sigma^2\sum_{i=0}^{t-1}\frac{\frac{9}{4}}{A(i+2)[A(i+2)+B]}\prod_{j=i+1}^{t}\left(1-\frac{\frac{3}{2}}{j+2}\right)$$

$$+\sigma^2\frac{\frac{9}{4}}{A(t+2)[A(t+2)+B]}$$

$$= \mathbb{E}\left[\sum_{d=1}^{2}\|\Delta_d^0\|_2^2\right]\prod_{i=2}^{t+2}\left(1-\frac{\frac{3}{2}}{i}\right)+\sigma^2\sum_{i=2}^{t+1}\frac{\frac{9}{4}}{Ai[Ai+B]}\prod_{j=i+1}^{t+2}\left(1-\frac{\frac{3}{2}}{j}\right)+\sigma^2\frac{\frac{9}{4}}{A(t+2)[A(t+2)+B]}$$

Since $A > 0$ and $B > 0$ thus

$$\mathbb{E}\left[\sum_{d=1}^{3}\|\Delta_d^{t+1}\|_2^2\right]$$

$$\leq \mathbb{E}\left[\sum_{d=1}^{3}\|\Delta_d^0\|_2^2\right]\prod_{i=2}^{t+2}\left(1-\frac{\frac{3}{2}}{i}\right)+\sigma^2\sum_{i=2}^{t+1}\frac{\frac{9}{4}}{Ai[Ai+B]}\prod_{j=i+1}^{t+2}\left(1-\frac{\frac{3}{2}}{j}\right)+\sigma^2\frac{\frac{9}{4}}{A(t+2)[A(t+2)+B]}$$

$$\leq \mathbb{E}\left[\sum_{d=1}^{3}\|\Delta_d^0\|_2^2\right]\prod_{i=2}^{t+2}\left(1-\frac{\frac{3}{2}}{i}\right)+\sigma^2\sum_{i=2}^{t+1}\frac{\frac{9}{4}}{(Ai)^2}\prod_{j=i+1}^{t+2}\left(1-\frac{\frac{3}{2}}{j}\right)+\sigma^2\frac{\frac{9}{4}}{[A(t+2)]^2}$$

We can next use the fact that for any $a \in (1, 2)$:

$$\prod_{i=\tau+1}^{t+2}\left(1-\frac{a}{i}\right)\leq\left(\frac{\tau+1}{t+3}\right)^a.$$

The bound then becomes

$$\mathbb{E}\left[\sum_{d=1}^{3}\|\Delta_d^{t+1}\|_2^2\right]$$

$$\leq \mathbb{E}\left[\sum_{d=1}^{3}\|\Delta_d^0\|_2^2\right]\prod_{i=2}^{t+2}\left(1-\frac{\frac{3}{2}}{i}\right)+\sigma^2\sum_{i=2}^{t+1}\frac{\frac{9}{4}}{(Ai)^2}\prod_{j=i+1}^{t+2}\left(1-\frac{\frac{3}{2}}{j}\right)+\sigma^2\frac{\frac{9}{4}}{[A(t+2)]^2}$$

$$\leq \mathbb{E}\left[\sum_{d=1}^{3}\|\Delta_d^0\|_2^2\right]\left(\frac{2}{t+3}\right)^{\frac{3}{2}}+\sigma^2\sum_{i=2}^{t+1}\frac{\frac{9}{4}}{(Ai)^2}\left(\frac{i+1}{t+3}\right)^{\frac{3}{2}}+\sigma^2\frac{\frac{9}{4}}{[A(t+2)]^2}$$

$$= \mathbb{E}\left[\sum_{d=1}^{3}\|\Delta_d^0\|_2^2\right]\left(\frac{2}{t+3}\right)^{\frac{3}{2}}+\sigma^2\sum_{i=2}^{t+2}\frac{\frac{9}{4}}{(Ai)^2}\left(\frac{i+1}{t+3}\right)^{\frac{3}{2}}$$

Note that $(i+1)^{\frac{3}{2}} \leq 2i$ for $i = 2, 3, \ldots$, thus

$$\mathbb{E}\left[\sum_{d=1}^{3} \|\Delta_d^{t+1}\|_2^2\right]$$

$$\leq \mathbb{E}\left[\sum_{d=1}^{2} \|\Delta_d^0\|_2^2\right]\left(\frac{2}{t+3}\right)^{\frac{3}{2}} + \sigma^2 \frac{\frac{9}{4}}{A^2(t+3)^{\frac{3}{2}}} \sum_{i=2}^{t+2} \frac{(i+1)^{\frac{3}{2}}}{i^2}$$

$$\leq \mathbb{E}\left[\sum_{d=1}^{3} \|\Delta_d^0\|_2^2\right]\left(\frac{2}{t+3}\right)^{\frac{3}{2}} + \sigma^2 \frac{\frac{9}{2}}{A^2(t+3)^{\frac{3}{2}}} \sum_{i=2}^{t+2} \frac{1}{i^{\frac{1}{2}}}$$

finally note that $\sum_{i=2}^{t+2} \frac{1}{i^{\frac{1}{2}}} \leq \int_1^{t+2} \frac{1}{x^{\frac{1}{2}}} dx \leq 2(t+3)^{\frac{1}{2}}$. Thus

$$\leq \mathbb{E}\left[\sum_{d=1}^{3} \|\Delta_d^0\|_2^2\right]\left(\frac{2}{t+3}\right)^{\frac{3}{2}} + \sigma^2 \frac{9}{A^2(t+3)}$$

substituting $A = 2\xi - 6\phi$ gives

$$= \mathbb{E}\left[\sum_{d=1}^{3} \|\Delta_d^0\|_2^2\right]\left(\frac{2}{t+3}\right)^{\frac{3}{2}} + \sigma^2 \frac{9}{[2\xi - 6\phi]^2(t+3)}$$

This leads us to the final theorem. To obtain Theorem 10.8, we proceed in an analogous way as form Theorem 10.3, but this time applying Theorem 10.10 that we have just proved.

## 10.5 BEHAVIOR GUIDED POLICY GRADIENT AND WASSERSTEIN TRUST REGION

The chief goal of this section is to prove Theorem 5.1. We restate the section's definitions here for the reader's convenience: To ease the discussion we make the following assumptions:

1. Finite horizon $T$.
2. Undiscounted MDP.
3. States are time indexed. In other words, states visited at time $t$ can't be visited at any other time.
4. $\mathcal{S}$ and $\mathcal{A}$ are finite sets.

The third assumption is solely to avoid having to define a time indexed Value function. It can be completely avoided. We chose not to do this in the spirit of notational simplicity. These assumptions can be relaxed, most notably we can show similar results for the discounted and infinite horizon case. We chose to present the finite horizon proof because of the nature of our experimental results.

Let $\Phi = \mathrm{id}$ be the identity embedding so that $\mathcal{E} = \Gamma$. In this case $\mathbb{P}_\pi^\Phi$ denotes the distribution of trajectories corresponding to policy $\pi$. We define the value function $V^\pi : \mathcal{S} \to \mathbb{R}$ as

$$V^\pi(s_t = s) = \mathbb{E}_{\tau \sim \mathbb{P}_\pi^{\mathrm{id}}}\left[\sum_{\ell=t}^{T} R(s_{\ell+1}, a_\ell, s_\ell) | s_t = s\right]$$

The Q-function $Q^\pi : \mathcal{S} \times \mathcal{A} \to \mathbb{R}$ as:

$$Q^\pi(s_t, a_t = a) = \mathbb{E}_{\tau \sim \mathbb{P}_\pi^{\mathrm{id}}}\left[\sum_{\ell=t}^{T} R(s_{\ell+1}, a_\ell, s_\ell)\right]$$

Similarly, the advantage function is defined as:

$$A^\pi(s, a) = Q^\pi(s, a) - V^\pi(s)$$

We denote by $V(\pi) = \mathbb{E}_{\tau \sim \mathbb{P}_\pi^{\mathrm{id}}}\left[\sum_{t=0}^{T} R(s_{t+1}, a_t, s_t)\right]$ the expected reward of policy $\pi$ and define the visitation frequency as:

$$\rho_\pi(s) = \mathbb{E}_{\tau \sim \mathbb{P}_\pi^{\mathrm{id}}}\left[\sum_{t=0}^{T} \mathbf{1}(s_t = s)\right]$$

The first observation in this section is the following lemma:

**Lemma 10.13.** *two distinct policies $\pi$ and $\tilde{\pi}$ can be related via the following equation :*

$$V(\tilde{\pi}) = V(\pi) + \sum_{s \in \mathcal{S}} \left( \rho_{\tilde{\pi}}(s) \left( \sum_{a \in \mathcal{A}} \tilde{\pi}(a|s) A^{\pi}(s,a) \right) \right)$$

*Proof.* Notice that $A^{\pi}(s,a) = \mathbb{E}_{s' \sim P(s'|a,s)} [R(s',a,s) + V^{\pi}(s') - V^{\pi}(s)]$. Therefore:

$$\mathbb{E}_{\tau \sim \mathbb{P}_{\tilde{\pi}}^{\text{id}}} \left[ \sum_{t=0}^{T} A_{\pi}(s_t, a_t) \right] = \mathbb{E}_{\tau \sim \mathbb{P}_{\tilde{\pi}}^{\text{id}}} \left[ \sum_{t=0}^{T} R(s_{t+1}, a_t, s_t) + V^{\pi}(s_{t+1}) - V^{\pi}(s_t) \right]$$

$$= \mathbb{E}_{\tau \sim \mathbb{P}_{\tilde{\pi}}^{\text{id}}} \left[ \sum_{t=0}^{T} R(s_{t+1}, a_t, s_t) \right] - \mathbb{E}_{s_0} [V^{\pi}(s_0)]$$

$$= -V(\pi) + V(\tilde{\pi})$$

The result follows. $\square$

See Sutton et al. (1998) for an alternative proof. We also consider the following linear approximation to $V$ around policy $\pi$ (see: Kakade & Langford (2002)):

$$L(\tilde{\pi}) = V(\pi) + \sum_{s \in \mathcal{S}} \left( \rho_{\pi}(s) \left( \sum_{a \in \mathcal{A}} \tilde{\pi}(a|s) A^{\pi}(s,a) \right) \right)$$

Where the only difference is that $\rho_{\tilde{\pi}}$ was substituted by $\rho_{\pi}$. Consider the following embedding $\Phi^s : \Gamma \to \mathbb{R}^{|\mathcal{S}|}$ defined by $(\Phi(\tau))_s = \sum_{t=0}^{T} \mathbf{1}(s_t = s)$, and related cost function defined as: $C(\mathbf{v}, \mathbf{w}) = \|\mathbf{v} - \mathbf{w}\|_1$.

**Lemma 10.14.** *The Wasserstein distance $\mathrm{WD}_0(\mathbb{P}_{\tilde{\pi}}^{\Phi^s}, \mathbb{P}_{\pi}^{\Phi^s})$ is related to visit frequencies since:*

$$\mathrm{WD}_0(\mathbb{P}_{\tilde{\pi}}^{\Phi^s}, \mathbb{P}_{\pi}^{\Phi^s}) \geq \sum_{s \in \mathcal{S}} |\rho_{\pi}(s) - \rho_{\tilde{\pi}}(s)|$$

*Proof.* Let $\Pi$ be the optimal coupling between $\mathbb{P}_{\tilde{\pi}}^{\Phi^s}$ and $\mathbb{P}_{\pi}^{\Phi^s}$. Then:

$$\mathrm{WD}_0(\mathbb{P}_{\tilde{\pi}}^{\Phi^s}, \mathbb{P}_{\pi}^{\Phi^s}) = \mathbb{E}_{u,v \sim \Pi} [\|u - v\|_1]$$

$$= \sum_{s \in \mathcal{S}} \mathbb{E}_{u,v \sim \Pi} [|u_s - v_s|]$$

Where $u_s$ and $v_s$ denote the $s \in \mathcal{S}$ indexed entry of the $u$ and $v$ vectors respectively. Notice that for all $s \in \mathcal{S}$ the following is true:

$$\left| \underbrace{\mathbb{E}_{u \sim \mathbb{P}_{\pi}^{\Phi^s}} [u_s]}_{\rho_{\pi}(s)} - \underbrace{\mathbb{E}_{v \sim \mathbb{P}_{\pi}^{\Phi^s}} [v_s]}_{\rho_{\pi'}(s)} \right| \leq \mathbb{E}_{u,v \sim \Pi} [|u_s - v_s|]$$

The result follows.

$\square$

These observations enable us to prove an analogue of Theorem 1 from Schulman et al. (2015), namely:

**Theorem 10.15.** *If $\mathrm{WD}_0(\mathbb{P}_{\tilde{\pi}}^{\Phi^s}, \mathbb{P}_{\pi}^{\Phi^s}) \leq \delta$ and $\epsilon = \max_{s,a} |A^{\pi}(s,a)|$, then $V(\tilde{\pi}) \geq L(\tilde{\theta}) - \delta\epsilon$.*

As in Schulman et al. (2015), Theorem 5.1 implies a policy improvement guarantee for BGPG from Section 5.3.

*Proof.* Notice that:

$$V(\tilde{\pi}) - L(\tilde{\pi}) = \sum_{s \in \mathcal{S}} \left( (\rho_{\tilde{\pi}}(s) - \rho_{\pi}(s)) \left( \sum_{a \in \mathcal{A}} \tilde{\pi}(a|s) A^{\pi}(s, a) \right) \right)$$

Therefore by Holder inequality:

$$|V(\tilde{\pi}) - L(\tilde{\pi})| \leq \underbrace{\left( \sum_{s \in \mathcal{S}} |\rho_{\pi}(s) - \rho_{\tilde{\pi}}(s)| \right)}_{\leq \mathrm{WD}_0(\mathbb{P}_{\tilde{\pi}}^{\Phi^s}, \mathbb{P}_{\pi}^{\Phi^s}) \leq \delta} \underbrace{\left( \sup_{s \in \mathcal{S}} \left| \sum_{a \in \mathcal{A}} \tilde{\pi}(a|s) A^{\pi}(s, a) \right| \right)}_{\leq \epsilon}$$

The result follows. $\qquad\square$

We can leverage the results of Theorem 10.15 to show wasserstein trust regions methods with embedding $\Phi^s$ give a monotonically improving sequence of policies. The proof can be concluded by following the logic of Section 3 in Schulman et al. (2015).

