# OpenReview forum: "Behavior-Guided Reinforcement Learning"
_ICLR.cc/2020/Conference — Reject_

### Official Review · AnonReviewer2 · 2019-10-19
**Official Blind Review #2**

**Rating:** 1

**Review:**

Summary

This paper proposes using Wasserstein Distances to measure the difference between higher-level functions of policies, which this paper terms as "behaviors". For example, one such behavior could be the distribution over final states given the policy, or the distribution over returns given policy. Through the lens of these behavioral embeddings, this paper recovers a few important special cases that are well-known in the literature including WD-based TRPO and distributional RL. This paper shows that the dual formulation of the Wasserstein Distance gives the ability to score individual policies based on a given "behavioral mapping".

Review

The idea of generalizing the trust-region of TRPO by using the WD measure and behavior maps is intriguing. The paper introduces many choices of behavior maps that could lead towards interesting algorithms that merit additional study. I think the connections between the proposed family of algorithms and WD-based TRPO and distributional RL is highly motivating.

There are, however, a several key issues with the empirical study of the proposed methods that make it challenging to assess the value of introducing another partially understood deep policy gradient algorithm. The results show only behavioral studies of the algorithm, not investigating the effects of the WD based regularizer or the effects of the choice of behavioral map. The results are also significantly limited in their statistical significance, making distinguishing between algorithms difficult in most cases. And the comparison of wall-clock time is particularly difficult to assess due to the uncountably many possible sources of noise when comparing wall-clock time of highly complex algorithms. In the following paragraphs, I will expand upon each of these points.

One of the key contributions of this paper is the ability to define regularizers based on the definition of the behavior map. The paper introduces many such behavior maps, those over state visitation, actions, and returns; however, the paper does not investigate the effects this novel choice has on the results. It is unclear if introducing each of these behavior maps leads towards different results or if they each induce roughly the same final performance of the agent. It would be highly valuable for me to see a more careful study of the effect of choosing each of these behavior maps on a single simple environment, clarifying that this formulation leads to a family of useful algorithms. As it stands, BGPG may only beat TRPO on these domains because it had more meta-parameters to choose between (BGPG introduces many new meta-parameters: choice of behavior map, kernel for produce RKHS, the meta-parameters of that kernel, the meta-parameters of the behavior map like trajectory length, the entropy regularization term in the WD, etc.). Without a careful study, or intuitive explanation of any of these parameter choices, it is unclear if BGPG won simply through overfitting to the problem.

The statistical significance of the proposed algorithm is impossible to assess in the given form. The comparisons are made using only five random seeds and the standard error bars are frequently quite large. I refer to Henderson et al. 2017 for further explanation as to why five random seeds is simply too few to provide a meaningful comparison between algorithms. Instead, I'll discuss a few of the results in particular. In figure 3a, I notice that the proposed method has high variance until it plateaus around -300 reward; why? In each of the remaining plots of Figure 3, I notice that the propose method is significantly higher variance than any of its competitors. This greatly leads me to suspect that the proposed method would have much lower average performance if studied across a greater number of random seeds. In the walker domain (Figure 3d), why do BGPG and TRPO both plateau at the same point for many timesteps, then eventually BGPG starts improving again? From Figure 5, the paper somewhat misleadingly states that "BGES is the only method that drives the agent to the goal in both settings." However Figure 5 on the left clearly shows that BGES and NSR-ES have indistinguishable performance, and the error bars indicate that neither significantly outperform NoisyNet-TRPO. In fact, due to the high variance of BGES it is unclear if it would drive the agent towards the goal on average if run over more random seeds. On the right, NoisyNet-TRPO noticeably outperforms BGES and is significantly lower variance. The language used to describe Figure 6 is strong, stating that Figure 6 "proves that the benefits come here not just from introducing the regularizer, but from its particular form." However, I tend to disagree that Figure 6 reliably proves anything. Although it appears that BGES is significantly outperforming other methods, the number of meta-parameters over which it gets to optimize makes it difficult to say if the performance gain is from overfitting to the problem or the form of the regularizer. Additionally, good performance demonstrated across a single problem is hardly proof especially using only five random seeds (Henderson et al. 2017 Figure 5).

Although this plays comparatively little role in my scoring of the paper, I feel it is necessary to discuss briefly. Because it is impossible to determine the source of the speed differences between the particle approximator and BGPG, the results in Figure 4 are extremely difficult to interpret. It simply could be that BGPG uses slightly more optimized code than the particle approximator, or it could be that there is in fact a significant performance difference between the algorithms. The only true way to discuss performance differences between algorithms generally is through computational complexity for these reasons. In the case that these statistical algorithms share the same computational complexity, then further analysis including convergence rates would be able to shed light on the speed difference. However, wall-clock time vs. performance has so many confounding factors, it is a fairly meaningless unit of measure. For an empirical investigation of speed, I would suggest giving each algorithm similar number of learning steps or similar number of updates to their weights and compare performance in this way.

Additional Comments (do not affect score)

I'm curious if this work could be extended to the off-policy case. It does seem like a minor disadvantage that distributional RL is well-defined in both the on-policy and off-policy cases, but the proposed family of methods is not. From my reading of the paper, it appears that there is little preventing this extension. Is this true?

--------
Edit after reading other reviews and rebuttals.

I appreciate the response noting that the meta-parameters of this algorithm were not tuned. I still feel that understanding the sensitivity to these parameters is important to understand how the choice of these parameters impacts the performance. While it is certainly likely that extensive tuning of these parameters could yield even more improved results, it is also quite possible that the chosen parameters have biased the results. If a more intuitive discussion of the choice of meta-parameters could be included in the paper, I would be more willing to concede this point.

I recognize that 3 or 5 random seeds are a standard in the deep RL community, but I think Henderson et al. 2017 make this point better than I could: this is not a standard that we should hold to. I think that using small domains and smaller networks would allow running across more random seeds and providing a much more careful scientific study. This paper does not convincingly show state of the art performance, a metric that is nearly impossibly to define in a field that moves as quickly as ours, so should not strive to follow the same demonstration study design that SOTA papers follow. A paper with strong theoretical motivations (such as this) should compliment with strong empirical understanding of how that theory translates to practice. Instead, having strong theoretical motivations backed by a demonstration that the algorithm still works is much less convincing (to me). I think a more convincing set of experiments that match well with the intended purpose of the paper (theoretical contribution of behavioral embeddings) would be an investigation to _how_ the optimization affects the learned policies or representations. Instead of performance benchmarks, this would provide understanding for the various effects of the new framework.

After the rebuttal phase, I now have an additional new concern about the clarity of the writing and the completeness of the empirical details. Many of the results do not mention most of the important details necessary to replicate or even interpret the results. I had assumed that details were consistent across figures (e.g. figure 3 uses 5 random seeds, so I assumed the same to be true for all other figures that did not specify), but the author response made clear that this is not true. In additional, little to no details are provided in the paper referring to the choices of meta-parameters for any algorithm or the methodology used to chose these meta-parameters. After these details were given in the rebuttal, my concerns were greatly alleviated, but not completely removed. It is unclear if the authors intend to update the paper before the final version, but based on the language of the rebuttal (e.g. indicating it is purely a misunderstanding of the paper), I am forced to assume that the paper will not include these important details.

To summarize. I think the theoretical underpinnings of these paper are extremely motivating and interesting. I do not want to see these innovations lost. However, the current state of the empirical section of the paper leaves too much open for me to be able to recommend an accept at this time. I do not believe a paper should solve everything in one pass (e.g. careful parameter studies, SOTA demonstrations, real-world demonstrations, etc.), but I do not believe this paper demonstrates any one of these convincingly.

**Experience Assessment:**

I have published one or two papers in this area.

**Review Assessment: Checking Correctness Of Derivations And Theory:**

I assessed the sensibility of the derivations and theory.

**Review Assessment: Checking Correctness Of Experiments:**

I carefully checked the experiments.

**Review Assessment: Thoroughness In Paper Reading:**

I read the paper thoroughly.

---

> ### Author Response · Authors · 2019-11-07
> **Author response to AnonReviewer2 - Part 1**
>
> Thank you for your detailed review of our paper. Please see below our responses to some of your concerns/comments (since we have detailed comments to each of the reviewer's points we split our response in three):
>
> -> “The results show only behavioral studies of the algorithm, not investigating the effects of the WD based regularizer or the effects of the choice of behavioral map... The paper introduces many such behavior maps...however, the paper does not investigate the effects this novel choice has on the results. It is unclear if introducing each of these behavior maps leads towards different results or if they each induce roughly the same final performance of the agent. “
>
> Thank you for the comment. We emphasize in the paper that *it is not our goal* to conduct detailed studies of the quality of different embeddings types (indeed, in the  experimental section we successfully use all three main types), but rather to propose a learning paradigm (which is novel on its own), where all these classes can be utilized. The main contribution of the paper is that paradigm, where policies are translated to probabilistic behavioral embeddings that are then leveraged via corresponding regularizers to improve large spectrum of RL algorithms ranging from imitation learning to ES. Furthermore, this paradigm is built on rigorous mathematical foundations of Wasserstein metrics, which itself constitutes a significant technical contribution. We leave the analysis of the relationship between the structure of the RL problem and optimal embedding to be applied to future work, agreeing that it is an exciting question to be studied.
>
> In addition, please see results on repulsion learning in the appendix that further show how can this paradigm be useful in applications that go beyond policy optimization.
>
> Specifically: The BEM we use for BGPG is actually \pi(a|s_t) which is the same as what is used in the KL constraint for TRPO. Thus, the only difference between BGPG and TRPO is the use of the Wasserstein Distance. However, our method does allow different embeddings, which as we discussed will be left for future work, and thus we believe our new perspective can . The novelty comes from the differing distance metric + the means to calculate it (AO vs. particle), as well as the framework, of which the embedding from TRPO is one case. For BGES, we do indeed include an ablation study for the simple point environment. This was in the appendix but we have moved it to the main body of the paper, as we understand the reviewer’s concerns and believe others may feel the same way. We hope this adds clarification!
>
>
> “As it stands, BGPG may only beat TRPO on these domains because it had more meta-parameters to choose between (BGPG introduces many new meta-parameters: choice of behavior map, kernel for produce RKHS, the meta-parameters of that kernel, the meta-parameters of the behavior map like trajectory length, the entropy regularization term in the WD, etc.). Without a careful study, or intuitive explanation of any of these parameter choices, it is unclear if BGPG won simply through overfitting to the problem.”
>
> To clarify, we did not tune these parameters, especially for TRPO. See comments on each below:
> -choice of behavior map: discussed above, we used the same as regular TRPO
> -kernel: we only considered the rbf kernel, widely used in machine learning. In other comments we discussed the potentially more flexible representations from other kernels, but we did not want to consider this as an additional degree of freedom for this work.
> -meta-params of kernel: we only used one set of meta-parameters for all experiments
> -behavior map trajectory length: this is the default length for openai gym/deepmind control suite tasks
>
> We only ran one setting of each of these, and in all cases there was an obvious choice which we did not deviate from. We were satisfied that our results showed promise, could make people think about policy optimization in a different way, and encourage several future directions for both ourselves and other members of the community.

---

> > ### Author Response · Authors · 2019-11-07
> > **Author response to AnonReviewer2 - Part 2**
> >
> > Part 2:
> >
> > -> Experimental results:
> >
> > “The results are also significantly limited in their statistical significance, making distinguishing between algorithms difficult in most cases.”
> >
> > See detailed answers regarding that  point below:
> >
> > Metaparameters of the algorithm: We think there is some fundamental misunderstanding regarding how our algorithm performs. We *do not tune* any hyperparameters of the algorithm, there is no any pre-learning phase. We fix a priori Gaussian kernel to define RKHS functions, use standard random feature map mechanism and choose beta = ½ to equally weight the regularizer and standard objective. No effort was made to optimize these choices. In fact we believe presented results can be substantially improved if more careful metaparameter tuning is applied. However we did not do that to show that the algorithm *does not overfit* to tediously tuned hyperparameters. Therefore we believe the main concern about overfitting to metaparameters that leads to down-weighting the statistical significance of the obtained results is irrelevant.
> >
> > “And the comparison of wall-clock time is particularly difficult to assess due to the uncountably many possible sources of noise when comparing wall-clock time of highly complex algorithms.”
> >
> > Though the wall-clock time of a given algorithm is influenced by multiple factors, we control all the other variations in the algorithm and compare the only changing part. It is then reasonable to assume that the difference in the wall-clock time is due to the changing factor, which in this case is the way to carry out the optimization on the Wasserstein distance.
> >
> > “The comparisons are made using only five random seeds and the standard error bars are frequently quite large.”
> >
> > In some cases we could have run our methods for longer, but we lack the computational budget to run all baselines for 2x the amount of compute, and wanted to ensure fair comparison across all methods. We felt that five seeds (rather than three which is often presented in RL papers), was enough to convince us that our methods worked well. Several state of the art papers in the field use five random seeds and thus our paper does not differ in that sense from previous works.
> >
> > “In figure 3a, I notice that the proposed method has high variance until it plateaus around -300 reward; why?”
> >
> > “In each of the remaining plots of Figure 3, I notice that the propose method is significantly higher variance than any of its competitors. This greatly leads me to suspect that the proposed method would have much lower average performance if studied across a greater number of random seeds.”
> >
> > Why is this? The mean -1sd of our approach is often still better than the mean +1sd of the next best method. Presumably increasing the number of seeds will reduce the sd in both cases and only increase this gap.
> >
> > “In the walker domain (Figure 3d), why do BGPG and TRPO both plateau at the same point for many timesteps, then eventually BGPG starts improving again?”
> >
> > This is hard to explain qualitatively, as we did not analyze the learned gaits at different stages of optimization.
> >
> > “From Figure 5, the paper somewhat misleadingly states that "BGES is the only method that drives the agent to the goal in both settings." However Figure 5 on the left clearly shows that BGES and NSR-ES have indistinguishable performance, and the error bars indicate that neither significantly outperform NoisyNet-TRPO. In fact, due to the high variance of BGES it is unclear if it would drive the agent towards the goal on average if run over more random seeds. On the right, NoisyNet-TRPO noticeably outperforms BGES and is significantly lower variance.”
> >
> > I think the reviewer misunderstood our comment. We specifically said that BGES is the only method that achieves good performance on *both*. In each case, another method did in fact also perform well. NoisyNet TRPO did achieve locomotion for the quadruped, but did not pass the wall, hence plateauing at <-5000. NSR-ES failed to pass the wall for the point environment. Hence we believe our claim is reasonable.

---

> > > ### Author Response · Authors · 2019-11-07
> > > **Author response to AnonReviewer2 - Part 3**
> > >
> > > Part 3:
> > >
> > > “The language used to describe Figure 6 is strong...However, I tend to disagree that Figure 6 reliably proves anything. Although it appears that BGES is significantly outperforming other methods, the number of meta-parameters over which it gets to optimize makes it difficult to say if the performance gain is from overfitting to the problem or the form of the regularizer. Additionally, good performance demonstrated across a single problem is hardly proof especially using only five random seeds...”
> > >
> > > See our discussion on hyperparameters of the algorithm, also he additional ablation study added to the paper. Furthermore, we never said that these results were over *five random seeds*.  These results we obtained over many more random seeds, we will clarify it in the final version. Thus we believe that the comment on the number of random seeds here is irrelevant. We run our experiments for Swimmer since it is one of the few OpenAI Gym problems where ES optimization is known to suffer from being trapped in local maxima. Thus several previous papers used that environment to demonstrate the performance of improved versions of ES baselines. Therefore we believe this is a perfect example proving our claim. Notice also that a setting in which we consider Swimmer environment is just one from a variety of different settings, where we consistently obtain gains by using our behavioral Wasserstein regularizers. In addition, this experiment serves as an ablation study, where we included four additional distance metrics not previously used for ES.
> > >
> > >
> > > Extensions to off-policy case:
> > >
> > > This is a great question! It would be possible to use an importance sampling estimator to make use of off policy data which would require the use of stochastic policies. It is less clear how to adapt a Q learning type of algorithm to fit this framework. Our guess would be that it may be possible in case when the embedding maps are markovian (in other words, decompose into a sum or concatenation of smaller embedding maps for triplets (s, a, r) along a trajectory). Additionally, this question points in the direction of a very recent paper “If MaxEnt is the answer, what is the Question?”. The authors of this work pose the question, when is it possible to decompose into a markovian reward a “score function” over a trajectory and provide (in some cases) conditions under which this is possible. A similar story may be true for our methods, whenever the Behavioral Test functions can be decomposed into markovian reward signals then it is possible to use a Q learning type of algorithm. Again, we think this is another exciting, non-trivial direction which our paper could inspire.

---

### Official Review · AnonReviewer3 · 2019-10-23
**Official Blind Review #3**

**Rating:** 3

**Review:**

This work explores two uses of Wasserstein distances (WD) within reinforcement learning: the first is a variant of policy gradient, where WD is used to guide the policy search (instead of alternative such as Trust-region used in TRPO); the second is a variant of evolutionary search where WD is used again to guide the policy updates.

One of the strengths of the work is to clarify the notion of Behavior embeddings (Sec.3), which I expect can have several uses in RL.   In this paper, the behavioral embeddings are assumed to be given; it would be interesting to discuss/explore learning these embeddings.

Section 4 of the paper reviews key concepts related to WD.  This is much harder to follow for an RL researcher, and would be improved by adding some intuition relating the material presented to the concepts of Sec.3.  Furthermore, this confusion carries out in Sec.5.  For example, what is the best way to think of \lambda_1 and \lambda_2?  And the maps s_1 and s_2?  What are necessary/desirable properties of P^\phi_b?   There are also many steps packed in Alg.2 & Alg.3, which are difficult to unpack.  For example, what are the \epsilon (step 1., Alg.3), scalars or vectors, how are they sampled?  It would be helpful to have a discussion of the complexity (both data & compute) of both algorithms.

Section 6 presents empirical results for each proposed algorithm.  Corresponding baselines are presented, but I would be interested to see a wider set of baseline methods. The literature is rich with methods in these classes, both variants of TRPO and ES.  It’s necessary to at least pick a representative sample to show and compare (e.g. GAE, SAC).  I am also puzzled by the actual results presented, for example the Hopper reward shown in Fig.3 seems much worse (by orders of magnitude) compared to that reported in the SAC paper (Haarnoja et al. 2018).




**Experience Assessment:**

I have published in this field for several years.

**Review Assessment: Checking Correctness Of Derivations And Theory:**

I assessed the sensibility of the derivations and theory.

**Review Assessment: Checking Correctness Of Experiments:**

I carefully checked the experiments.

**Review Assessment: Thoroughness In Paper Reading:**

I read the paper at least twice and used my best judgement in assessing the paper.

---

> ### Author Response · Authors · 2019-11-07
> **Author response to AnonReviewer3**
>
> Thank you for your review and detailed comments.
>
> -> Clarity of the presentation:
>
> To address this key issue, we added a whole new section (section 2) which we hope makes the paper more digestible. Please let us know if this can be further improved. We understand that Wasserstein distances come with a great deal of notation. Unfortunately this is hard to avoid - but we think the improved version is much easier to read (and we hope you think so too). We also think that successfully incorporating such techniques as Wasserstein metrics, which are not typically used in RL, should be considered a strength rather than a weakness of the presented approach. In addition, it may be illustrative to see the section on repulsion learning in the appendix for the reviewer to understand the meaning of lambda_i.
>
> Given the technical nature of our contribution, we think it is important to maintain details in the Algorithm boxes, so that anyone can reproduce them. For a simple, intuitive demo of BGES, we strongly encourage all reviewers to see our open source implementation (https://github.com/behaviorguidedRL/BGRL) which has the full algorithm. It only requires numpy and a MuJoCo license, and runs on a single CPU in minutes.
>
>
> -> Section 6:
>
> The goal of the paper is *not to* propose a completely new algorithm for policy learning that can be competitive or outperform state-of-the-art variants of algorithms from TRPO, ES classes, etc. We rather propose a *new paradigm* for policy learning via behavioral embeddings and show that we can enrich a large class of different state-of-the-art RL algorithms (ranging from ES to imitation learning, and repulsion learning (see appendix for repulsion learning)) by adding behavioral regularizers proposed by that paradigm. That is why the goal was not to compare with several different versions of the TRPO algorithm, but rather to show that such versions can be improved by using our techniques. Given the diversity of the considered settings, we believe we managed to show that. We also hope we motivated this better (as we discussed above), so that the reviewers share our belief that this approach can be extended in material ways, thus presents a strong contribution.
>
> To clarify, in the experiments  we implemented the TPRO algorithm where the KL-divergence constraint is applied as a penalty in the optimization objective. This formulation is in fact closer to the PPO [Schulman et al, 2017] formulation of the trust-region policy search problem. This formulation allows us to naturally replace the KL-divergence penalty by the Wasserstein distance penalty, and leads to alternating optimization. Also, we used the most recent state-of-the-art ES algorithms applying state and reward normalization methods (as in state-of-the-art class of augmented random search algorithms, see: https://arxiv.org/pdf/1803.07055.pdf) as well as  compact neural network architectures (for more effective training) and quasi Monte Carlo sampling for better quality ES gradient estimations (see: https://arxiv.org/pdf/1804.02395.pdf).
>
> -> Comparison vs. SAC for Hopper
>
> The first thing we note is that we used the DeepMind Control Suite version of these tasks, which have a different scale for reward. For some of these environments it may be the case that results for SAC are stronger than our methods. However, we emphasize that SAC is an off policy method, so not a direct comparison (see discussion in Reviewer 2’s comments). We feel that although on-policy algorithms are not necessarily SOTA for every task, they still warrant further research. What we are mainly showing is improvement over a baseline (in this case TRPO - which is still widely used in the community), plus a framework for future research which could lead to a vast array of future works.

---

### Official Review · AnonReviewer1 · 2019-10-25
**Official Blind Review #2**

**Rating:** 6

**Review:**


Summary: this paper proposes a new regularized policy optimization (PO) method which is based on Wasserstein distances. Its idea is to use SGD to optimize the dual form of the WD, then used in two different policy search approaches TRPO and Evolution Strategies. The evaluations are carried out on a variety of control tasks from OpenAI Gym.


Overall, the paper studies an interesting problem in RL. The idea is somewhat interesting. The experiment results also look promising. However, the writing sometimes has unclear descriptions, probably because there are many things packed in this paper. I have some following concerns about it.

- The idea of using behavior embedding is new in PO. The description in section 3 is a bit unclear. There are lacks of motivation why in the paper there needs BES/BEM, the interplay of policies, trajectories, and embedding maps, and why non-surjective of BEM is still fine in this case, etc.. In addition, the definition of state-based, action-based, reward-based assume only discrete domains? Besides, they seem not to reappear in other places in the paper.

- Does the choice of the functions \lambda in 4.2 as a function in RKHS, especially when approximated as a linear function with random Gaussian features, limit the representation power of the embedding space?

- What is the effect of \beta when positive vs. negative?

- Experiments: The proposed methods show a lot of potentials, but the description in this section is sometimes unclear. Is TRPO with KL divergence the standard algorithm defined without using BEM or with BEM? Then one can wonder how more ablations can be added, e.g. BGPG with KL divergence, TRPO with WD distances are compared with the proposed algorithm. In addition, how BGPG is compared with other related work that also uses skill/policy embedding, instead of a flat PO approach like TRPO. Similar questions are also applied to the experiments for BGES.

- It would also be more self-contained if the paper includes experiment settings.

- The paper also comes with theoretical results. The author could also consider mentioning one key result in the main paper, instead, everything is put in the appendix.


* Minor comments:
- The KL definition on page 4 use rho instead of \xi


- are u and v in Eq. 3 functions in \cal C(X) and \cal C(Y), respectively?

**Experience Assessment:**

I have read many papers in this area.

**Review Assessment: Checking Correctness Of Derivations And Theory:**

I assessed the sensibility of the derivations and theory.

**Review Assessment: Checking Correctness Of Experiments:**

I assessed the sensibility of the experiments.

**Review Assessment: Thoroughness In Paper Reading:**

I read the paper at least twice and used my best judgement in assessing the paper.

---

> ### Author Response · Authors · 2019-11-07
> **Author response to AnonReviewer1**
>
> Thank you very much for your review!
>
> Below we clarify some of your questions/comments:
>
> “In addition, the definition of state-based, action-based, reward-based assume only discrete domains...”
>
> This works also for continuous domains. In particular we used all three types of embeddings in our experiments with continuous domains, as we emphasize in the paper. For instance, reward-based embeddings maps a trajectory to a final reward and action-based embedding just concatenates actions (that one we used in particular in TRPO experiments). In fact, TRPO already does this, but presents it in a different way. In that sense, the only difference between BGPG and TRPO is the WD vs. KL distance - however - we introduce the flexibility to try new (potentially learned) representations, which we hypothesize could lead to significant performance improvements. In all these settings there is no need for a finite set of states or actions.
>
> “Does the choice of the functions \lambda in 4.2 as a function in RKHS, especially when approximated as a linear function with random Gaussian features, limit the representation power of the embedding space ?”
>
> Thank you for the question. In practice it does not (yet makes the problem tractable) since considered RKHS functions are dense in the family of continuous functions. It may be possible that other kernels lead to better representations. In fact, it may even be possible to learn the kernel, one of many directions we are currently considering on the back of this paper.
>
> “What is the effect of \beta when positive vs. negative?”
>
> Positive \beta encourages being diverse from the previous policies, negative \beta has an effect of attracting to the behavior of the desired policy. We chose simple equal weighted values (beta = 0.5 for BGES), as in NSR-ES (Conti ‘18). We did this in the hope to make our comparisons fair. However, we are excited about this as a potential future research direction, and noted as such in the Conclusion. We think that it may be desirable to adaptively repel/attract from policies at different stages of optimization, possibly in a population setting. However, we feel this is beyond the scope of a paper where we already introduce several new concepts. For now, the hard coded values (which were not tuned) represent a (convincing) proof of concept.
>
> We did our best to fix all listed typos. Thank you for pointing them out!

---

### Author Response · Authors · 2019-11-07
**Meta response for all reviewers**

Thank you all for your time in reviewing our paper - we appreciate the detailed comments and believe all questions have helped us improve the paper.

We have just resubmitted an updated version, which we hope addresses many of your concerns. In particular, we:
Include a new section [MOTIVATING BEHAVIOR-GUIDED REINFORCEMENT LEARNING], where we introduce the framework in an intuitive way, which should make it easier to follow during the technical sections
Include additional clarification of our experiments, including a new ablation study of differing BEMs for BGES.

Additionally, we want to emphasize that the purpose of this paper is to introduce the “theory” of behavioral guided Reinforcement Learning, our methods use the Wasserstein distance to obtain signals that help existing algorithms guide their optimization towards or away from desired behaviors. Our techniques have application on (at least) policy optimization, imitation learning, repulsion learning, exploration and even safety (although we did not pursue that angle in our paper). As such, the purpose of our paper is not to beat every single benchmark but to show the promise of this framework for different downstream tasks.

That being said - for five seeds, in a variety of environments, we believe there is evidence that our methods do work. We are incredibly excited about several future directions beyond this, and we hope to work on many ourselves.

We also encourage all reviewers (/members of the public on this forum) to check our open source demo. It is very simple, and runs in just a few minutes on a single CPU. We hope this leads to future collaborations/inspires others.

---

### Decision · Program_Chairs · 2019-12-19

**Decision:**

Reject

**Comment:**

The authors introduce the idea of using Wasserstein distances over latent "behavioral spaces" to measure the similarity between two polices, for use in RL algorithms.  Depending on the choice of behavioral embedding, this method produces different regularizers for policy optimization, in some cases recovering known algorithms such as TRPO.  This approach generalizes ideas of similarity used in many common algorithms like TRPO, making these ideas widely applicable to many policy optimization approaches.  The reviewers all agree that the core idea is interesting and would likely be useful to the community.  However, a primary concern that was not sufficiently resolved during the rebuttal period was the experimental evaluation -- both the ability of the experiments to be replicated, as well as whether they provide sufficient insight into how/why the algorithm performs.  Thus, I recommend rejection of this paper at this time.